# Control of mitophagy initiation and progression by the TBK1 adaptors NAP1 and SINTBAD

Elias Adriaenssens [1,2,3,10] ✉, Thanh Ngoc Nguyen [3,4,5,6,10],
Justyna Sawa-Makarska [1,2,3], Grace Khuu [3,4,5,6], Martina Schuschnig[1,2],
Stephen Shoebridge[1,2,3], Marvin Skulsuppaisarn[4,5,6], Emily Maria Watts[4,5],
Kitti Dora Csalyi [7], Benjamin Scott Padman [5,8,9], Michael Lazarou [3,4,5,6] ✉ &
Sascha Martens [1,2,3] ✉

Mitophagy preserves overall mitochondrial fitness by selectively targeting damaged mitochondria for degradation. The regulatory mechanisms that prevent PTEN-induced putative kinase 1 (PINK1) and E3 ubiquitin ligase Parkin (PINK1/Parkin)-dependent mitophagy and other selective autophagy pathways from overreacting while ensuring swift progression once initiated are largely elusive. Here, we demonstrate how the TBK1 (TANK-binding kinase 1) adaptors NAP1 (NAK-associated protein 1) and SINTBAD (similar to NAP1 TBK1 adaptor) restrict the initiation of OPTN (optineurin)-driven mitophagy by competing with OPTN for TBK1. Conversely, they promote the progression of nuclear dot protein 52 (NDP52)-driven mitophagy by recruiting TBK1 to NDP52 and stabilizing its interaction with FIP200. Notably, OPTN emerges as the primary recruiter of TBK1 during mitophagy initiation, which in return boosts NDP52-mediated mitophagy. Our results thus define NAP1 and SINTBAD as cargo receptor rheostats, elevating the threshold for mitophagy initiation by OPTN while promoting the progression of the pathway once set in motion by supporting NDP52. These findings shed light on the cellular strategy to prevent pathway hyperactivity while still ensuring efficient progression.

Mitochondria are dynamic and multifunctional organelles that fuel energy production through oxidative phosphorylation and have pivotal roles in cell signaling, biosynthetic pathways and programmed cell death[1–3]. They are susceptible to damage from reactive oxygen species and other stressors, necessitating stringent quality control mechanisms[4–6]. The selective removal of damaged mitochondria through autophagy is termed mitophagy and has emerged as essential for maintaining a healthy mitochondrial network[7–12]. Impaired mitophagy is linked to diverse human disorders, including neurodegenerative diseases, cancer, metabolic syndromes and aging[13].

PINK1 and Parkin are key components in mitophagy[14,15], and mutations in these genes underlie early-onset Parkinson's disease[16–18].

Under basal conditions, PINK1 is continuously degraded by the proteasome[19–21]. However, upon mitochondrial damage, PINK1 accumulates at the outer mitochondrial membrane, recruiting and activating Parkin[22–29]. Parkin marks damaged mitochondria with ubiquitin for recognition by the cargo receptors (also known as cargo adaptors) OPTN and NDP52 (also called CALCOCO2)[30–38]. Autophagosome formation is initiated by the cargo receptors, directly on the surface of the cargo, leading to the engulfment and degradation of the damaged organelle.

TBK1 is a master kinase in mitophagy and other selective autophagy pathways, phosphorylating cargo receptors such as OPTN and NDP52 to increase their affinities for ubiquitin and LC3/GABARAP proteins[32,33]. However, OPTN and NDP52 use TBK1 in different ways. Although TBK1

is essential for OPTN-mediated mitophagy initiation[30,39,40], NDP52 can redundantly use either TBK1 or ULK1 as the mitophagy-initiating kinase[40]. These mechanistic differences between OPTN and NDP52 suggest that TBK1 regulatory factors could have significant roles during mitophagy initiation, especially given that OPTN can directly bind TBK1 whereas NDP52 does not[30,32,33,39,41–44].

NAP1 (also known as AZI2) and SINTBAD are TBK1 adaptors (hereafter referred to as NAP1/SINTBAD). Through binding to the SKICH domain of NDP52, NAP1/SINTBAD bridge NDP52 and TBK1 in xenophagy, a selective autophagy pathway designed to protect the cytosol against bacterial invasion[42,43,45,46]. NAP1/SINTBAD were found to support NDP52-mediated degradation of *Salmonella enterica* serovar Typhimurium by interacting with TBK1 and the core autophagy factor FIP200[42,43,46,47]. However, NAP1/SINTBAD share the same TBK1 binding site as OPTN[44], prompting questions about their potential roles in mitophagy and how their seemingly opposing interactions with OPTN and NDP52 might impact mitophagy dynamics.

We therefore investigated the roles of NAP1/SINTBAD in PINK1/Parkin-dependent mitophagy and discovered their overall inhibitory role in this pathway. Although they support NDP52-mediated mitophagy, they negatively regulate TBK1 recruitment and activation by OPTN. This competition for TBK1 binding prevents OPTN from fulfilling one of its primary functions during mitophagy initiation. Our findings highlight a multilayer regulation of mitophagy initiation by NAP1/SINTBAD, acting as cargo receptor rheostats that increase the threshold for mitophagy initiation but promote the progression of the pathway once set in motion. As such, NAP1/SINTBAD provide insight into the cellular strategy that prevents selective autophagy pathways from overreacting while ensuring swift progression once initiated.

## Results

### NAP1/SINTBAD are recruited and co-degraded during mitophagy

To understand whether the TBK1 adaptors NAP1/SINTBAD have a function in PINK1/Parkin mitophagy, we investigated whether NAP1/SINTBAD are recruited to mitochondria during this process. To this end, we stably expressed HA-NAP1 or HA-SINTBAD in wild-type HeLa cells that also expressed YFP–Parkin and assessed their subcellular localization. Under basal conditions, NAP1/SINTBAD were dispersed throughout the cytosol (Fig. 1a). However, upon induction of mitophagy using a combination of oligomycin A and antimycin A1 (O/A), agents targeting the mitochondrial ATP synthase and complex III, respectively, both NAP1 and SINTBAD notably accumulated on depolarized mitochondria (Fig. 1a). We then performed co-staining with WIPI2, a marker for early cup-shaped membrane structures known as phagophores, the precursors to autophagosomes. This demonstrated colocalization between NAP1/SINTBAD and WIPI2 (Fig. 1b), indicating that both NAP1 and SINTBAD were recruited to sites of autophagosome formation.

To test whether NAP1/SINTBAD are degraded along with damaged mitochondria during mitophagy, we assessed the protein levels of NAP1/SINTBAD and found a decrease in NAP1/SINTBAD levels upon mitophagy induction, which was partially mitigated when lysosomal degradation was inhibited by bafilomycin A1 (Fig. 1c). This indicates that NAP1/SINTBAD are not only recruited to sites of autophagosome formation but also a portion of NAP1/SINTBAD undergo autophagy-dependent degradation alongside damaged mitochondria, implying a potential role for them in the PINK1/Parkin mitophagy pathway.

### NAP1/SINTBAD are mitophagy inhibitors

To explore the involvement of NAP1/SINTBAD in PINK1/Parkin mitophagy, we generated knockout HeLa cells for both factors and assessed mitophagy flux. Depletion of either NAP1 or SINTBAD alone did not impact the mitophagy rate in a statistically significant manner, as shown by the mitochondrial-targeted mKeima (mt-mKeima) assay (Fig. 2a,b)[48]. Recognizing their structural similarities, which might

facilitate compensation for each other, we also generated NAP1/SINTBAD double knockout (DKO) cells. To our surprise, we observed an enhancement in mitophagy flux in NAP1/SINTBAD DKO cells (Fig. 2c), contrasting their supporting role in NDP52-mediated xenophagy[46]. This finding was validated by assessing mitochondrial protein COXII levels by western blotting, confirming accelerated mitochondrial degradation in NAP1/SINTBAD DKO cells (Fig. 2d).

To substantiate their inhibitory role, we investigated whether NAP1 overexpression could inhibit mitophagy. Our analysis indeed revealed that NAP1 overexpression led to reduced COXII degradation (Fig. 2e). Thus, NAP1/SINTBAD serve as mitophagy inhibitors, counteracting PINK1/Parkin-mediated mitophagy.

To explore whether NAP1/SINTBAD also regulate nonselective bulk autophagy, we evaluated p62 degradation in starved cells. Our findings indicated no discernible changes in p62 degradation in single or DKO cell lines when compared to control wild-type cells (Extended Data Fig. 1). Therefore, NAP1/SINTBAD are involved in the regulation of selective forms of autophagy, such as mitophagy, but not in nonselective bulk autophagy.

### NAP1/SINTBAD support NDP52-mediated mitophagy

To investigate the mechanisms underlying NAP1/SINTBAD's inhibition of mitophagy, we first focused on their functional interaction with NDP52, as they were previously implicated in an NDP52-dependent selective autophagy pathway, albeit in a stimulatory manner[46].

To explore their interplay with NDP52, we generated CRISPR–Cas9 DKO clones for NAP1/SINTBAD in the pentaKO background, which lacks five key cargo receptors: OPTN, NDP52, TAX1BP1, p62 and NBR1 (ref. 30). This allowed us to reintroduce NDP52 into these cells and to assess NDP52-driven mitophagy rates in the presence or absence of NAP1/SINTBAD, eliminating the confounding effects from other cargo receptors, including OPTN. Surprisingly, contrary to our previous observations (Fig. 2), deleting NAP1/SINTBAD in these cells resulted in reduced mitophagy. This was evident from reduced degradation of the mitochondrial marker COXII (Fig. 3a), decreased mt-mKeima conversion (Fig. 3b and Extended Data Fig. 2a) and impaired TBK1 activation (Fig. 3c). Moreover, the deletion of NAP1/SINTBAD may weaken the NDP52–FIP200 interaction in cells, given that in vitro reconstitution revealed that FIP200 was more robustly retained on NDP52-coated beads in presence of SINTBAD. This was particularly evident when the protein–protein interactions were weakened with increased salt concentrations (Fig. 3d). In cells, the absence of NAP1/SINTBAD resulted in a reduction of ATG13 recruitment to mitochondria during NDP52-mediated mitophagy (Extended Data Fig. 2b), underscoring the importance of NAP1/SINTBAD in this critical early step of mitophagy initiation.

Despite the significant contribution of NAP1/SINTBAD to these important first steps of NDP52-mediated mitophagy initiation, the overall reduction in mitophagy flux was relatively modest. However, considering that NDP52 can drive mitophagy through either ULK1/2 or TBK1 (ref. 40), we knocked out ULK1/2 in NAP1/SINTBAD DKO/pentaKO cells to elucidate the necessity of NAP1/SINTBAD when NDP52 engages in mitophagy solely through the TBK1 pathway. In the absence of ULK1/2, NAP1/SINTBAD emerged as crucial factors for NDP52-mediated mitophagy as evidenced by significantly reduced COXII turnover (Fig. 3e) and a reduction in WIPI2 recruitment to mitochondria upon O/A treatment (Fig. 3f). The impact on WIPI2 recruitment was comparable to the inhibition of TBK1 using the small molecule BX795.

In summary, these findings reveal important roles for NAP1/SINTBAD in supporting NDP52-mediated mitophagy through the recruitment of TBK1 and stabilization of the NDP52–FIP200 complex.

### Recruitment of NAP1/SINTBAD is sufficient to induce mitophagy

From our experiments above (Fig. 3), it becomes evident that NAP1/SINTBAD exhibit traits of cargo receptors, including their ability to bind

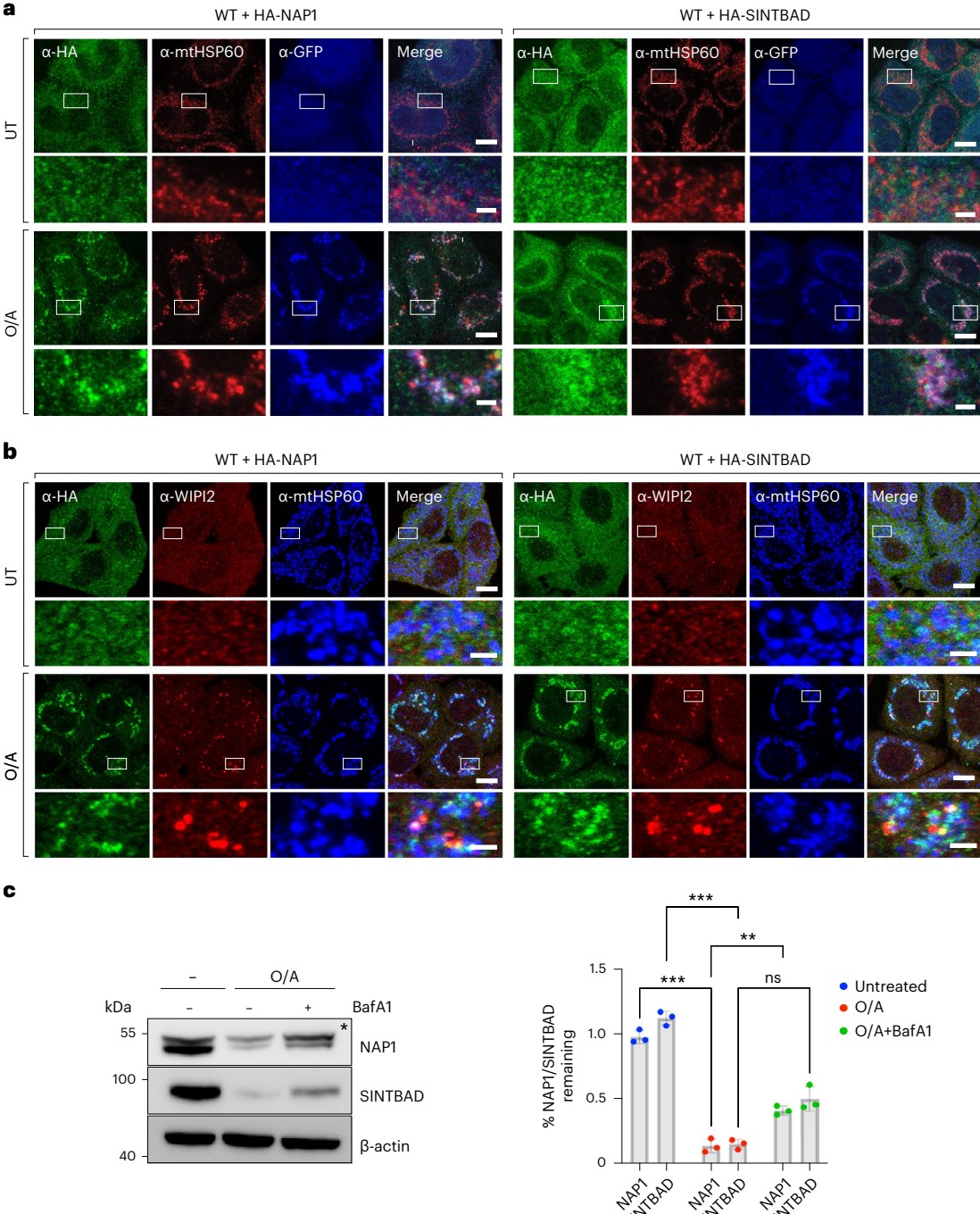

**Fig. 1 | NAP1 and SINTBAD are recruited and co-degraded during mitophagy.**
**a**,**b**, Wild-type (WT) HeLa cells stably expressing YFP–Parkin (**a**) or BFP–Parkin (**b**) and HA-NAP1 or HA-SINTBAD, left untreated (UT) or treated with O/A for 2 h and immunostained with indicated antibodies. The blue channel represents immunostaining in the far-red channel, represented as pseudocolored for blue. Scale bars: overviews, 10 μm; insets: 2 μm. (**c**) WT HeLa cells treated with O/A or O/A and bafilomycin A1 (BafA1) for 24 h and analyzed by immunoblotting. Asterisks indicate a nonspecific band. Densitometric analysis was performed for NAP1 and SINTBAD (mean ± s.d.) (*n* = 3 biologically independent experiments). Two-way ANOVA with Sidak's multiple comparison test was performed. *$P < 0.05$; **$P < 0.005$; ***$P < 0.001$; ns, not significant. Source numerical data, including exact *P* values, and unprocessed blots are available in source data.

FIP200 and TBK1, albeit lacking the ubiquitin-binding capabilities of cargo receptors. However, ubiquitin chains are critical in marking damaged organelles for autophagic degradation. We therefore reasoned that bypassing this ubiquitin-dependent recruitment by artificially tethering NAP1 to the outer mitochondrial membrane might be sufficient to initiate autophagosome biogenesis.

To test this hypothesis, we used a chemically induced dimerization assay, wherein FRB and FKBP can be dimerized upon rapalog addition[49,50]. By positioning FRB on the mitochondrial outer membrane through fusion with the transmembrane domain of Fis1 and attaching NDP52 or NAP1 to FKBP, we gained the ability to redirect NAP1 or NDP52 to the outer mitochondrial membrane upon rapalog treatment (Fig. 4a).

We first confirmed that NDP52 induced mitophagy upon rapalog addition (Fig. 4b), as previously demonstrated[30,51]. We then evaluated whether FKBP–NAP1 could similarly initiate mitophagy. Intriguingly,

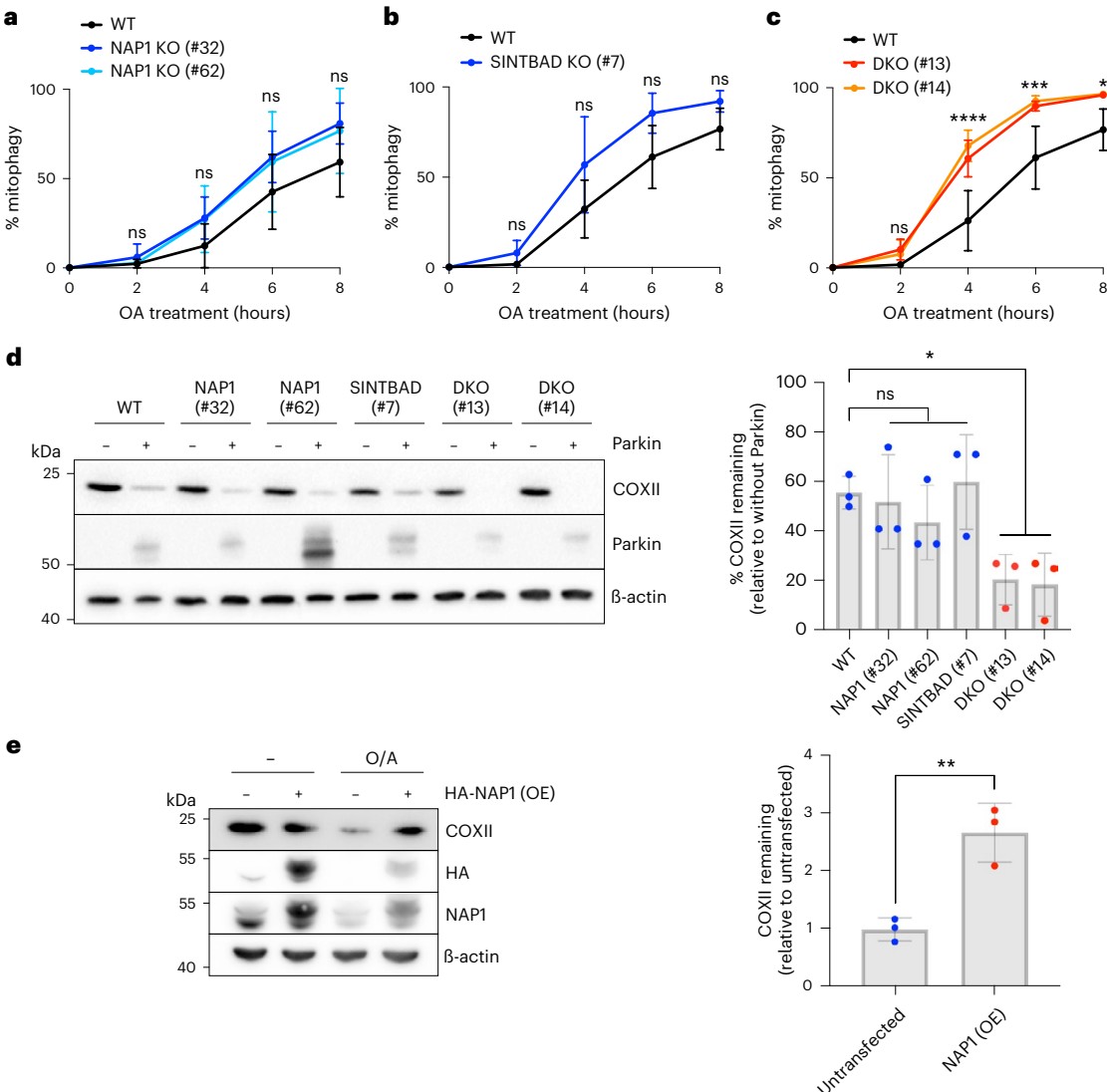

**Fig. 2 | NAP1 and SINTBAD are negative regulators of mitophagy.**
**a–c**, Mitophagy flux was measured by flow cytometry in indicated HeLa cell lines expressing YFP–Parkin and mt-mKeima, untreated or treated with O/A for indicated times: WT versus NAP1 knockout (KO) (**a**), SINTBAD KO (**b**) or NAP1 and SINTBAD DKO cells (**c**) (mean ± s.d.) (*n* = 3 biologically independent experiments). Two-way ANOVA with Tukey's multiple comparisons test was performed in **a** and **c**, and with Šídák's multiple comparison test in **b**. (**d**) Immunoblotting of COXII levels in various HeLa cell lines treated with O/A for 18 h. PINK1/Parkin-dependent versus PINK1/Parkin-independent mitophagy was compared by overexpression of YFP–Parkin. Densitometric analysis was performed for the percentage of COXII remaining (mean ± s.d.) (*n* = 3 biologically independent experiments). One-way ANOVA with Dunnett's multiple comparison test was performed. (**e**) Immunoblotting of COXII levels in HeLa cells overexpressing (OE) HA-NAP1 and treated with O/A for 16 h. The proportion of COXII remaining after O/A relative to the untransfected sample was quantified. Densitometric analysis was performed for COXII (mean ± s.d.) (*n* = 3 biologically independent experiments). A two-tailed unpaired Student's *t*-test was performed. *$P < 0.05$; **$P < 0.005$; ***$P < 0.001$; ****$P < 0.0001$; ns, not significant. Source numerical data, including exact *P* values, and unprocessed blots are available in source data.

artificial tethering of NAP1 to the mitochondrial surface resulted in comparable levels of mitophagy induction upon rapalog treatment compared to NDP52 (Fig. 4b). To rule out that this effect stemmed from the indirect recruitment of NDP52 by NAP1, we repeated the experiment in pentaKO cells. This confirmed that NAP1 could autonomously induce mitophagy, independently of NDP52 (Fig. 4b). Moreover, blocking autophagosome formation with a VPS34 inhibitor or impeding autophagosome degradation with bafilomycin A1 validated that the mitochondrial turnover was mediated by autophagy (Fig. 4c).

Using the rapalog-induced tethering assay, we further dissected the mechanism of NAP1-induced mitophagy. Specifically, we used NAP1 mutants deficient in NDP52-binding, FIP200-binding or TBK1-binding (Extended Data Fig. 3a,b). The mutants lacking NDP52-binding or FIP200-binding abilities retained their capacity to induce mitophagy upon rapalog treatment (Fig. 4d). However, the TBK1-binding deficient mutant lost its ability to initiate mitophagy, underscoring the critical role of the NAP1–TBK1 interaction in mitophagy. Consistently, inhibition of TBK1 with the small molecule GSK8612 prevented ectopically tethered NAP1 from inducing mitophagy (Fig. 4e). We then repeated the experiments with FKBP–SINTBAD and observed that SINTBAD can also induce mitophagy in a TBK1-dependent manner (Extended Data Fig. 3c,d).

As TBK1 is known to phosphorylate cargo receptors like OPTN and NDP52 to stimulate their LC3/GABARAP-binding capacity[33,41], we tested whether NAP1 would also be phosphorylated by TBK1. An in vitro kinase assay revealed that in the presence of ATP/MgCl₂, a smearing pattern for NAP1 was much more prominent when incubated with TBK1 than with ULK1 (Fig. 4f). To verify whether the smearing pattern

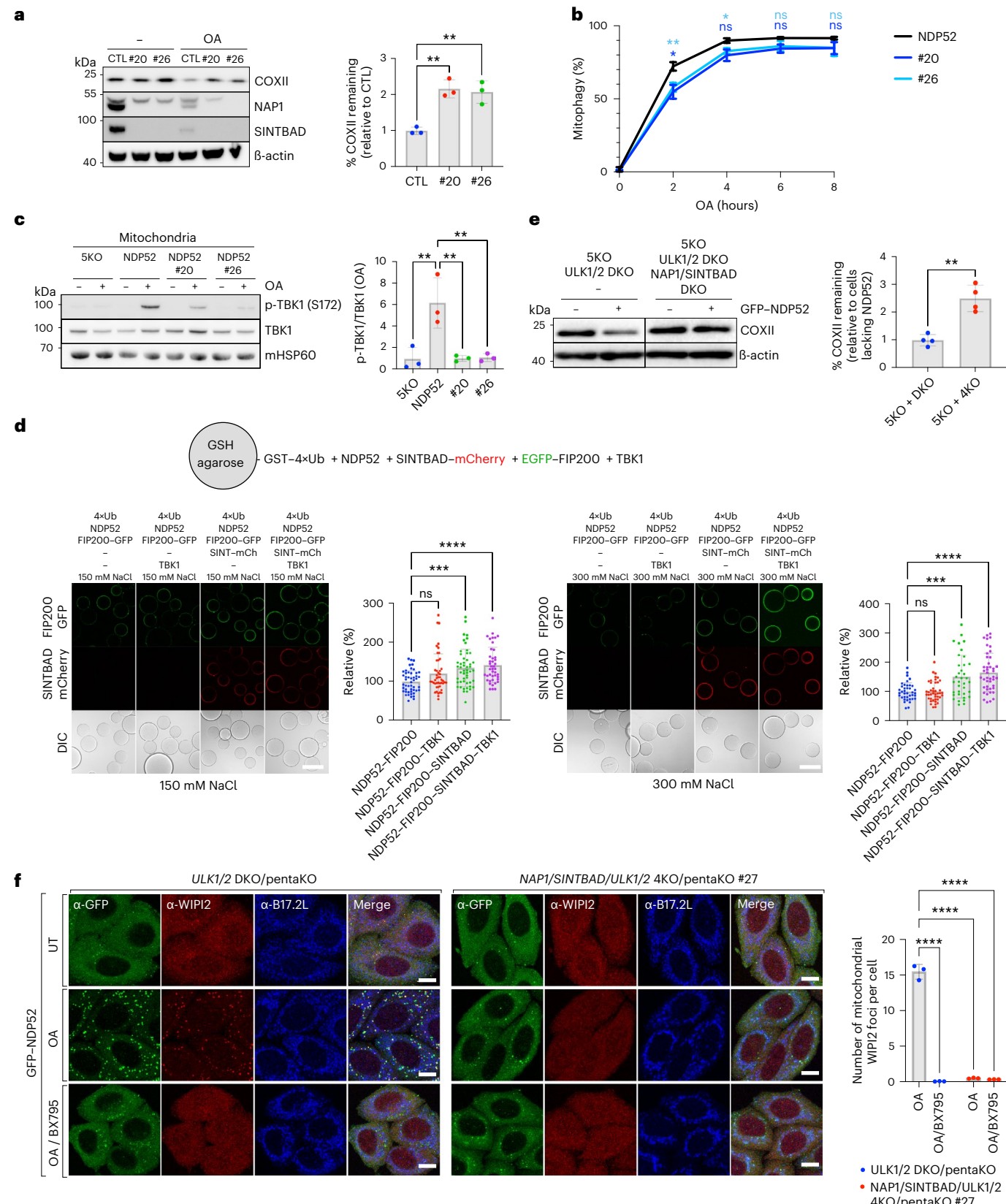

represented phosphorylation of NAP1 by TBK1, we analyzed these samples by mass spectrometry. This process confirmed that NAP1 was phosphorylated by TBK1 at several residues. We then tested whether NAP1 was also phosphorylated in cells, using FIP200 knockout cells in which TBK1 is hyperactivated[52,53], and confirmed that NAP1 is also a substrate of TBK1 phosphorylation in cells (Extended Data Fig. 4a,b). Moreover, many of the residues identified in vitro were also detected by mass spectrometry in FIP200 knockout cells (Fig. 4g and Extended Data Fig. 4c) and appeared as highly conserved residues (Extended Data Fig. 4d). Alanine substitution of the six residues identified both

**Fig. 3 | NAP1 and SINTBAD support NDP52-mediated mitophagy by stabilizing interactions with the autophagy machinery. a**, PentaKO (parental control, CTL) and NAP1/SINTBAD DKO/pentaKO (clones #20 and #26) expressing BFP–Parkin and GFP–NDP52 were treated with O/A for 16 h and analyzed by immunoblotting. Densitometric analysis was performed for COXII (mean ± s.d.) (*n* = 3 biologically independent experiments). One-way ANOVA with Dunnett's multiple comparison test was performed. **b**, Indicated cell lines expressing BFP–Parkin and mt-mKeima were treated with O/A for indicated times. Mitochondrial flux was measured by flow cytometry (mean ± s.d.) (*n* = 3 biologically independent experiments). Representative FACS plots are provided in Extended Data Fig. 2. Two-way ANOVA with Tukey's multiple comparisons test was performed. **c**, Crude mitochondria were isolated from pentaKO (5KO) and NAP1/SINTBAD DKO/pentaKO (clones #20 and #26) expressing BFP–Parkin and GFP–NDP52 untreated or treated with O/A for 1 h and analyzed via immunoblotting with indicated antibodies. The fraction of p-TBK1 over total TBK1 was quantified by densitometric analysis (mean ± s.d.) (*n* = 3 biologically independent experiments). One-way ANOVA with Dunnett's multiple comparison test was performed. **d**, Biochemical reconstitution of mitophagy initiation by NDP52. Glutathione sepharose beads coated with GST-tagged linear ubiquitin chains (GST–4×Ub) were incubated with NDP52, SINTBAD–mCherry, TBK1 and FIP200–GFP, as indicated, in bead assay buffer containing either 150 mM or 300 mM NaCl

and supplemented with ATP/MgCl₂. Samples were analyzed by confocal imaging. The mean signal intensity for an individual bead was quantified and plotted as individual data points (mean ± s.d.) (*n* = 35 beads or more per condition examined over three independent experiments). One-way ANOVA with Dunnett's multiple comparison test was performed. Scale bar, 100 µm. **e**, PentaKO with ULK1/2 DKO and pentaKO with ULK1/2/NAP1/SINTBAD 4KO (clones #13 and #27) expressing BFP–Parkin and GFP–NDP52 were treated with O/A for 16 h and analyzed by immunoblotting. Densitometric analysis was performed for COXII (mean ± s.d.) (*n* = 4 biologically independent experiments). Two-tailed unpaired Student's *t*-test was performed. **f**, PentaKO with ULK1/2 DKO and pentaKO with ULK1/2/NAP1/SINTBAD 4KO HeLa cells stably expressing BFP–Parkin were left untreated or treated with O/A or O/A plus TBK1 inhibitor (BX795) for 1 h, and immunostained with indicated antibodies. Note the defect in GFP–NDP52 recruitment in #27 owing to failure of recruiting downstream ATG8 molecules, which feedback and stabilize NDP52 (ref. 79). Scale bars, 10 µm. Number of mitochondrial WIPI2 foci per cell was quantified (mean ± s.d.) (*n* = 3 biologically independent experiments). Two-way ANOVA with Tukey's multiple comparisons test was performed. *$P < 0.05$; **$P < 0.005$; ***$P < 0.001$; ****$P < 0.0001$; ns, not significant. Source numerical data, including exact *P* values, and unprocessed blots are available in source data.

in vitro and in cells (S34, S82, S83, S120, S318, S343) eliminated the smearing of NAP1 in the in vitro kinase assay, indicating that these are the primary sites for TBK1 phosphorylation (Extended Data Fig. 4e). We then tested whether NAP1 is also phosphorylated during mitophagy and observed a similar smearing pattern, which was absent in cells lacking Parkin or treated with TBK1-inhibitor (Fig. 4h). Finally, we determined whether this TBK1-mediated phosphorylation activated the ATG8-binding capacity of NAP1 and SINTBAD. Indeed, phosphorylation by TBK1 boosted their capacity to interact with the ATG8 family protein LC3B (Fig. 4i). We then tested whether this phosphorylation forms an essential step during mitophagy by rescuing the cells from Fig. 3e, where NAP1 and SINTBAD are critical for NDP52-driven mitophagy, with the 6×Ala NAP1 variant. This revealed that the phospho-deficient mutant could rescue mitophagy to a similar extent as wild-type NAP1 (Extended Data Fig. 4f). In line with this finding, artificial tethering of FKBP–NAP1 to the mitochondrial surface also still induced mitophagy for the phospho-deficient mutant (Extended Data Fig. 4g). The NAP1 phosphorylation by TBK1 may therefore promote mitophagy without being strictly required, similar to how cargo receptors like OPTN and NDP52 are further activated by TBK1 phosphorylation.

Collectively, these findings highlight the resemblance of NAP1/SINTBAD to cargo receptors, with the exception of ubiquitin binding. By artificially tethering NAP1/SINTBAD to the mitochondrial surface, we demonstrated their competency as an autophagy cargo receptor in a TBK1-dependent manner. Direct phosphorylation by TBK1 promotes

their mitophagy-supporting activity. Based on these insights, we propose the term 'cargo co-receptors' for NAP1/SINTBAD, emphasizing their ability to facilitate selective autophagy through interactions with cargo receptors like NDP52.

## NAP1/SINTBAD compete with OPTN for TBK1 binding

Although the findings outlined above underscore the importance of NAP1/SINTBAD for NDP52-driven selective autophagy pathways, these results do not explain our earlier observations in NAP1/SINTBAD DKO cells, in which their overall effect on mitophagy was inhibitory rather than stimulatory. This suggests that the roles of NAP1/SINTBAD in mitophagy might be cargo-receptor-specific, considering that NAP1/SINTBAD DKO cells express all five cargo receptors, whereas experiments in the pentaKO background were conducted in cells expressing only NDP52. Given that NAP1/SINTBAD bind to TBK1 at the same binding site as OPTN[44], we reasoned that their inhibitory impact on mitophagy might arise from direct or indirect regulation of OPTN, the other major cargo receptor in PINK1/Parkin-dependent mitophagy.

To test whether NAP1/SINTBAD could inhibit mitophagy by competing with OPTN for TBK1 binding, we reconstituted the initiation of OPTN-driven mitophagy in vitro using purified components. Agarose beads coated with linear 4× ubiquitin, mimicking the surface of ubiquitin-marked damaged mitochondria, were co-incubated with mCherry-tagged OPTN, EGFP-tagged TBK1 and increasing concentrations of NAP1 (Fig. 5a). This experiment revealed that OPTN was

**Fig. 4 | NAP1 can drive mitophagy when artificially tethered to the mitochondrial surface. a**, Diagram of the experimental setup and the effect of rapalog treatment, resulting in the tethering of NDP52 or NAP1 to the outer mitochondrial membrane. IMS, intermembrane space; OMM, outer mitochondrial membrane. **b**, Mitophagy flux was measured by flow cytometry in WT or 5KO HeLa cells expressing BFP–Parkin and mt-mKeima, not induced or induced for 24 h by rapalog treatment. The percentage of mitophagy-induced cells (upper left) is quantified (mean ± s.d.) (*n* = 4 biologically independent experiments). **c**, As in the pentaKO cells from **b** but with and without the addition of autophagy inhibitors: PI3K inhibitor (VPS34 inh) and BafA1. The percentage of mitophagy-induced cells is quantified (mean ± s.d.) (*n* = 3 biologically independent experiments). **d**, Different NAP1 variants deficient in binding NDP52, FIP200 or TBK1 were ectopically tethered to the outer mitochondrial membrane and compared to wild-type NAP1 in 5KO + NAP1/SINTBAD DKO cells. The percentage of mitophagy-induced cells is quantified (mean ± s.d.) (*n* = 3 biologically independent experiments). **e**, As in **b**, with the pentaKO background, but with and without the addition of the TBK1 inhibitor (GSK8612)

(mean ± s.d.) (*n* = 3 biologically independent experiments). **f**, In vitro kinase assay analyzed by SDS–PAGE and western blotting. The recombinant proteins were mixed as indicated for varying times in the presence or absence of ATP/MgCl₂. **g**, Schematic of NAP1 residues phosphorylated by TBK1, as identified by mass spectrometry. Only those residues that were identified in both the in vitro kinase samples and in cellulo from FIP200 knockout cells are displayed in the yellow circles. **h**, Wild-type HeLa cells, with or without BFP–Parkin, were treated for 1 h with O/A alone or O/A plus TBK1-inhibitor (GSK8612) and analyzed by immunoblotting. **i**, Microscopy-based bead assay to verify the interaction between NAP1–mCherry or SINTBAD–GFP and GST–LC3B coated beads, mixed with recombinant TBK1 and incubated for 30 min in the absence or presence of ATP/MgCl₂. Samples were analyzed by confocal imaging. One of three representative experiments is shown. Scale bar, 100 µm. Representative FACS plots are shown for **b–e** and a two-way ANOVA with Tukey's multiple comparisons test was performed for **b–e**. *$P < 0.05$; **$P < 0.005$; ***$P < 0.001$; ****$P < 0.0001$; ns, not significant. Source numerical data, including exact *P* values, and unprocessed blots are available in source data.

recruited to the ubiquitin-coated beads, subsequently recruiting TBK1 (Fig. 5b). However, increasing NAP1 levels led to TBK1 displacement from the OPTN-bound beads, indicating that OPTN and NAP1 compete for the same binding site. This competition was further validated through conventional pull-down experiments (Extended Data Fig. 5).

To assess whether NAP1/SINTBAD also competed with OPTN for TBK1 binding in cells, we used the NAP1/SINTBAD DKOs in the pentaKO background, in which OPTN was reintroduced. This setup allowed us to distinguish the effects of NAP1/SINTBAD on OPTN-mediated mitophagy from those on NDP52-mediated

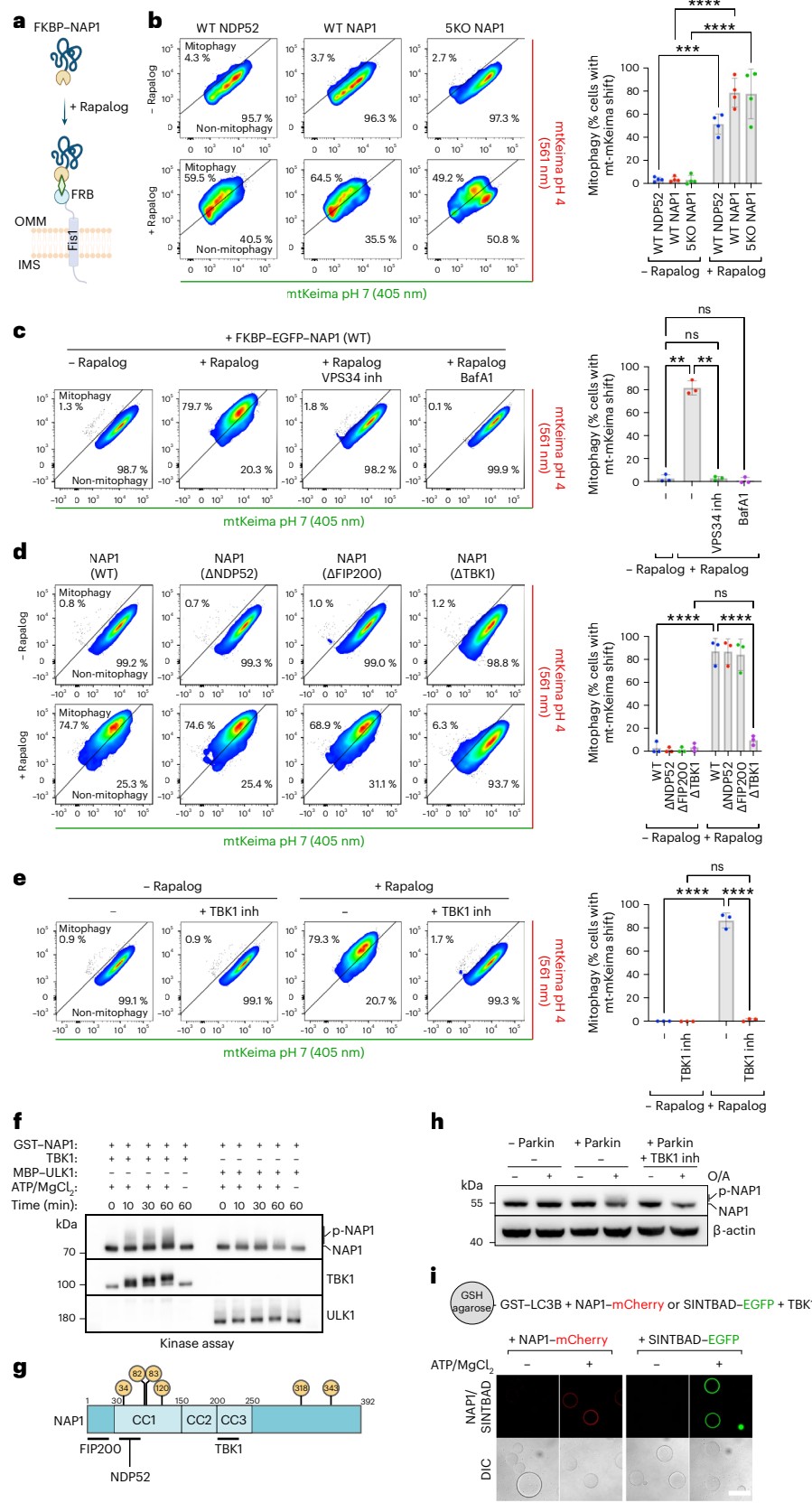

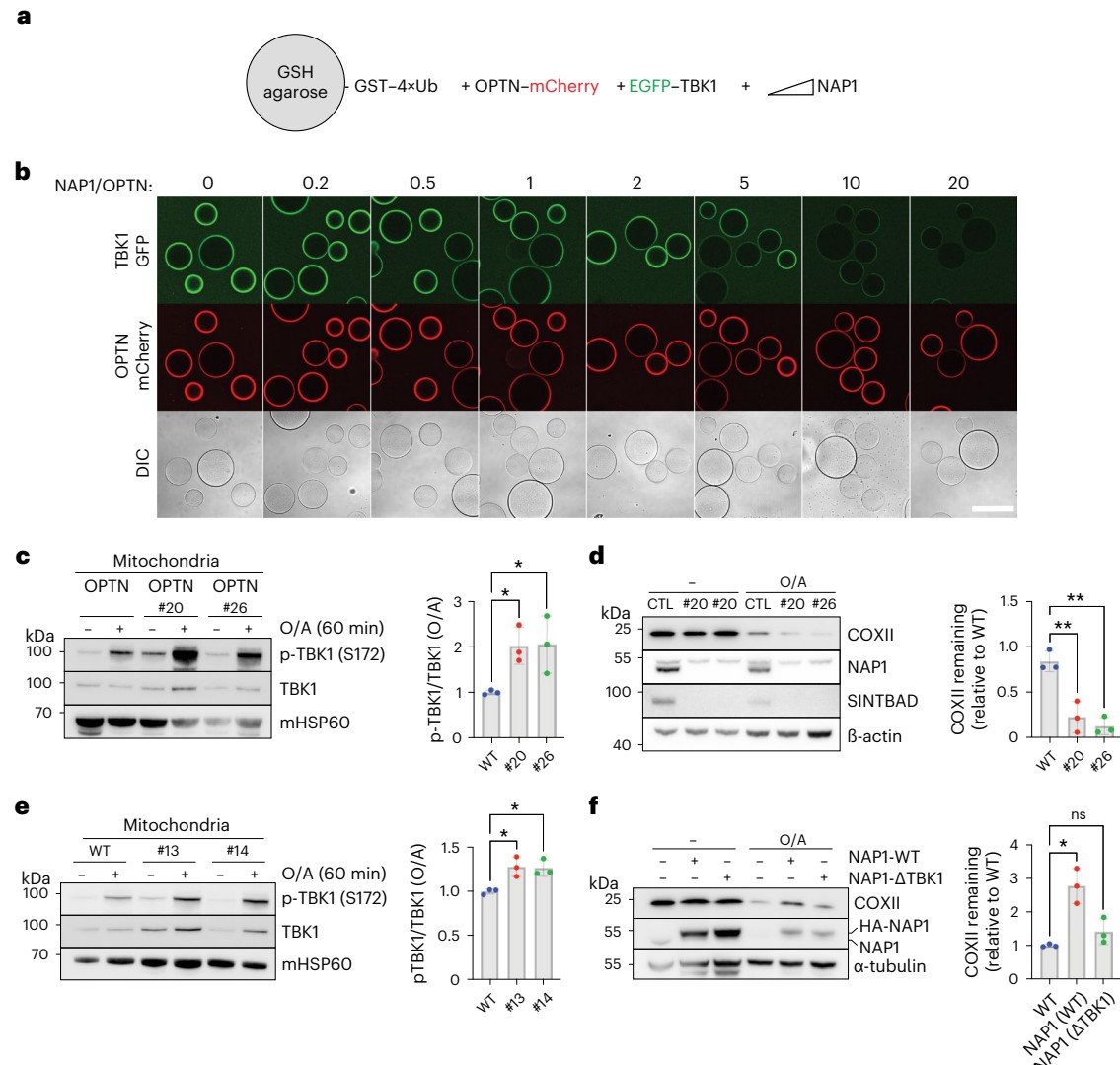

**Fig. 5 | NAP1 and SINTBAD compete with OPTN for TBK1 binding. a**, Diagram of the experimental setup to assess competition between OPTN and NAP1 for TBK1 binding. **b**, Biochemical reconstitution of the recruitment of GFP–TBK1 by mCherry–OPTN to GST–4×Ub-coated beads in the absence or presence of increasing amounts of NAP1. The experiment was performed without ATP, and samples were analyzed by confocal imaging. A representative experiment is shown ($n = 3$ biologically independent experiments). Scale bar, 100 μm. **c**, Crude mitochondria were isolated from pentaKO and NAP1/SINTBAD DKO/pentaKO (clones #20 and #26) expressing BFP–Parkin and GFP–OPTN untreated or treated with O/A for 1 h and analyzed by immunoblotting with indicated antibodies. Densitometric analysis was performed for the fraction of p-TBK1 over total TBK1 in O/A treated samples (mean ± s.d.) ($n = 3$ biologically independent experiments). **d**, Immunoblotting of COXII levels in pentaKO (parental control, CTL) and NAP1/SINTBAD DKO/pentaKO (clones #20 and #26) expressing BFP–Parkin and GFP–OPTN and treated with O/A for 16 h, analyzed by immunoblotting. The upper band for NAP1 is nonspecific. Densitometric

analysis of COXII relative to CTL in O/A samples (mean ± s.d.) ($n = 3$ biologically independent experiments). **e**, Crude mitochondria were isolated from WT HeLa cells and NAP1/SINTBAD DKO cells in WT background (clones #13 and #14) expressing BFP–Parkin untreated or treated with O/A for 60 min were analyzed by immunoblotting with indicated antibodies. Densitometric analysis was performed for the fraction of p-TBK1 over total TBK1 in O/A treated samples (mean ± s.d.) ($n = 3$ biologically independent experiments). **f**, WT HeLa cells expressing BFP–Parkin were stably transduced with HA-NAP1 WT or TBK1-binding deficient mutant (ΔTBK1). Cells were treated with O/A for 16 h and analyzed by immunoblotting. The percentage of COXII remaining after O/A treatment was quantified. Densitometric analysis of COXII relative to untransfected (WT) in O/A samples (mean ± s.d.) ($n = 3$ biologically independent experiments). One-way ANOVA with Dunnett's multiple comparison test was performed in **c**–**f**. *$P < 0.05$; **$P < 0.005$. Source numerical data, including exact $P$ values, and unprocessed blots are available in source data.

mitophagy. Following mitophagy induction in these cells, we observed increased TBK1 activation as indicated by higher levels of p-S172 TBK1 in the absence of NAP1/SINTBAD (Fig. 5c). This suggests that NAP1/SINTBAD suppress TBK1 activation in OPTN-mediated mitophagy, consistent with their competition with OPTN for TBK1 binding (Fig. 5b). We then examined whether increased TBK1 activation resulted in accelerated mitophagy. Indeed, measurements of mitophagy levels, indicated by COXII degradation in pentaKO cells rescued with OPTN, confirmed the acceleration of OPTN-driven

mitophagy in the absence of NAP1/SINTBAD (Fig. 5d). Consistently, we also detected accelerated mitophagy using the mt-mKeima assay (Extended Data Fig. 6).

Next, we quantified the amount of activated TBK1 relative to total TBK1 on the surface of purified mitochondria in wild-type cells versus NAP1/SINTBAD DKO cells expressing all five cargo receptors. This revealed increased TBK1 activation upon NAP1/SINTBAD deletion (Fig. 5e), suggesting that NAP1/SINTBAD are indeed competing with OPTN for TBK1 binding in the cell.

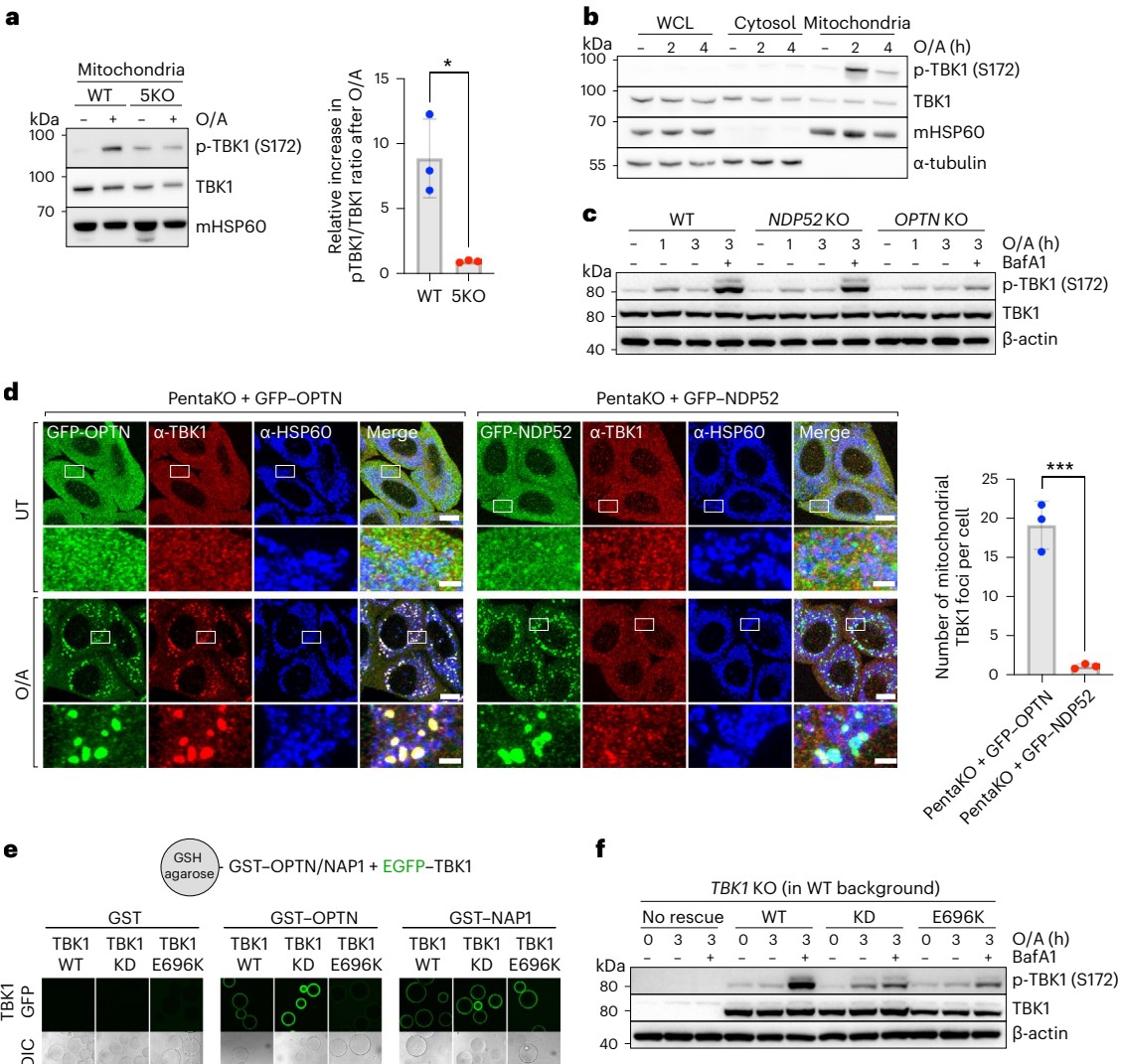

**Fig. 6 | OPTN is the primary recruiter and activator of TBK1 during mitophagy initiation. a**, Immunoblotting of crude mitochondrial fraction isolated from WT versus 5KO cells, expressing BFP–Parkin and treated with O/A for 1 h. Densitometric analysis was performed for the p-TBK1/TBK1 ratio in O/A treated samples relative to untreated samples (mean ± s.d.) (*n* = 3 biologically independent experiments). Two-tailed unpaired Student's *t*-test was performed. **P* < 0.05. **b**, Crude mitochondria were isolated from wild-type HeLa cells expressing BFP–Parkin, treated with O/A for the indicated times, and compared to the cytosolic and whole cell lysate (WCL) fractions by immunoblotting with indicated antibodies. **c**, Immunoblotting of phosphorylated TBK1 (S172) in wild-type versus OPTN or NDP52 knockout cells expressing BFP–Parkin treated with O/A in the absence or presence of BafA1 for the indicated times and analyzed by immunoblotting. **d**, PentaKO HeLa cells stably expressing BFP–Parkin and

rescued with GFP–OPTN or GFP–NDP52 were either left untreated or treated with O/A for 1 h and immunostained with indicated antibodies. Scale bars: overviews, 10 μm; insets: 2 μm. Number of mitochondrial TBK1 foci per cell was quantified (mean ± s.d.) (*n* = 3 biologically independent experiments). Two-tailed unpaired Student's *t*-test was performed. ****P* < 0.001. **e**, In vitro binding assay using glutathione-coupled agarose beads coated with GST, GST–OPTN or GST–NAP1 and incubated with EGFP–TBK1 WT, EGFP–TBK1 kinase-dead (KD) or EGFP–TBK1 E696K mutant (E696K). Samples were analyzed by confocal microscopy. Scale bar, 100 μm. **f**, Immunoblotting of TBK1 knockout HeLa cells that were either not rescued, rescued with TBK1 WT, TBK1 KD or TBK1 E696K mutant (E696K) and treated with O/A in the absence or presence of BafA1 for the indicated times. Source numerical data, including exact *P* values, and unprocessed blots are available in source data.

To further validate that NAP1/SINTBAD inhibit mitophagy, at least in part, through competition for TBK1 binding, we engineered a NAP1 mutant (L226Q/L233Q) deficient in TBK1 binding (Extended Data Fig. 3b) and assessed its inhibitory potential. Upon overexpression of wild-type NAP1 or the TBK1-binding deficient mutant in wild-type HeLa cells, we observed that wild-type NAP1 reduced the overall COXII degradation, as observed earlier (Fig. 2e). However, this effect was nearly completely abolished for the TBK1-binding deficient mutant (Fig. 5f and Extended Data Fig. 7), further supporting the notion that NAP1/SINTBAD restrict mitophagy initiation through competition for TBK1 binding.

Taken together, these findings reveal that NAP1/SINTBAD, through competition for TBK1 binding, can restrict the initiation of mitophagy.

## OPTN is the primary recruiter and activator of TBK1

The insights gathered above not only unveil a regulatory step at the onset of mitophagy but also shed light on a critical role for TBK1 in ensuring the efficient progression of mitophagy. Our results show that NAP1/SINTBAD restrict mitophagy initiation by limiting TBK1 recruitment by OPTN, hinting at a dominant role for OPTN in recruiting and activating TBK1. We therefore set out to dissect the underlying mechanisms of TBK1 recruitment and activation during mitophagy.

Consistent with prior research, we first confirmed that the activation of TBK1 strictly relies on the presence of cargo receptors, as their absence resulted in the absence of TBK1 activation (Fig. 6a)[30]. Furthermore, in line with the mechanism by which TBK1 is activated

through local clustering on the endoplasmic reticulum surface by the cGAS–STING complex[54–58], we find that TBK1 is also activated locally on the mitochondrial surface during mitophagy (Fig. 6b). This aligns with the requirement of TBK1 dimers to be brought into close proximity, enabling trans autophosphorylation, as the kinase domain cannot access the activation loop in *cis* and the two kinase domains in the dimer face away from one another[59–61]. Our data thus propose an essential role for cargo receptors in locally clustering TBK1 dimers on the mitochondrial surface. This is consistent with a recently proposed model, positing that TBK1 is activated from a local platform of OPTN molecules[52].

To test whether TBK1 activation predominantly relies on OPTN, as implied by our NAP1/SINTBAD results, we compared TBK1 activation in wild-type HeLa cells to cells lacking either OPTN or NDP52. This comparison revealed a severe reduction in TBK1 activation upon OPTN deletion as evident from decreased TBK1 phosphorylation (Fig. 6c). By contrast, NDP52 deletion had a relatively minor impact on TBK1 activation (Fig. 6c). We then compared the amount of TBK1 recruitment during OPTN-driven versus NDP52-driven mitophagy by rescuing the pentaKO cells with either OPTN or NDP52. This revealed the pronounced ability of OPTN to recruit TBK1 to the mitochondrial surface upon mitophagy induction, while NDP52 recruited TBK1 to a lesser extent (Fig. 6d).

To further corroborate this result, we used the ALS-causing TBK1-E696K mutation. This mutant failed to bind OPTN in vitro (Fig. 6e), in line with prior research[39,44,62,63]. However, this mutation retained its binding capacity to NAP1 (Fig. 6e). In wild-type HeLa cells, expressing both OPTN and NDP52, the TBK1-E696K mutant was previously shown to be no longer recruited to damaged mitochondria[39,62]. Consistently, we show that this is accompanied by a drastic reduction of TBK1 activation (Fig. 6f), reinforcing the importance of clustering for TBK1 activation. Moreover, these findings are also consistent with OPTN having a primary role in recruiting and clustering TBK1 on the mitochondrial surface, which cannot be sufficiently compensated for by the NDP52–NAP1/SINTBAD axis in HeLa cells. This underscores the importance of OPTN-mediated TBK1 recruitment.

Together, our results provide evidence for the crucial role of OPTN in recruiting and activating TBK1 during mitophagy, explaining how interference with this interaction by NAP1/SINTBAD can effectively restrict mitophagy initiation.

## Crosstalk between OPTN-axis and NDP52-axis stimulates mitophagy

Next, we asked whether the crucial role of OPTN in TBK1 activation might also influence the NDP52 axis. Previous research revealed that either cargo receptor alone is sufficient to initiate mitophagy[30]. However, several tissues express both cargo receptors. In tissues such as the brain, where NDP52 expression is low[30], the NDP52-related protein TAX1BP1 is expressed. Therefore, we reasoned that crosstalk might exist between OPTN and NDP52, allowing each receptor to leverage its

strengths so that their combined presence results in robust mitophagy control and progression.

To test this premise, we designed a system that enabled us to exploit the capacity of OPTN to recruit TBK1 during mitophagy, but omitting its ability to interact with other components of the autophagy machinery[41,64,65]. To this end, we created a rapalog-induced dimerization assay, linking only the minimal sequence of OPTN (residues 2–119) essential for TBK1 binding to FKBP (Fig. 7a). Using this system in the pentaKO background, we tested whether this truncated OPTN fragment could effectively recruit and activate TBK1 at the mitochondrial surface. Indeed, purification of mitochondria from rapalog-treated HeLa cells revealed that rapalog induced the translocation of FKBP–OPTN(2–119) and TBK1 (Fig. 7b). Crucially, the recruitment of TBK1 to the mitochondrial surface was sufficient to induce TBK1 activation, as demonstrated by the increase in phosphorylated TBK1 in the mitochondrial fraction upon rapalog treatment.

We then assessed the extent to which this minimal OPTN peptide could initiate mitophagy, as most of its essential autophagy-driving protein domains had been removed. Yet treating cells with rapalog for 24 h led to a notable fraction of cells undergoing mitophagy, as demonstrated by the mt-mKeima conversion (Extended Data Fig. 8), albeit to a lesser extent than with full-length OPTN. This observation shows that recruitment of TBK1 is sufficient for mitophagy initiation, consistent with our earlier finding that TBK1 can recruit the PI3KC3C1 complex[40].

With this minimal OPTN peptide at hand, we sought to elucidate whether TBK1 recruited through this truncated OPTN axis could enhance NDP52-driven mitophagy. To test this hypothesis, we rescued pentaKO cells with NDP52 and further transduced them with FKBP–OPTN(2–119) and Fis1–FRB. This experimental setup enabled us to measure mitophagy rates by NDP52 upon mitochondrial depolarization by O/A, both in the presence and absence of additional TBK1 recruited through rapalog treatment. Rapalog alone resulted in relatively slow mitophagy activation, displaying only minimal activation from 4 h onwards, whereas the combined treatment of O/A and rapalog substantially accelerated mitophagy flux (Fig. 7c and Extended Data Fig. 9a). Importantly, this increase in mitophagy flux was not a merely additive effect, based on the kinetics of rapalog treatment alone, especially during the first 3 h of treatment during which we observed minimal mitophagy induction by rapalog alone, suggesting that the recruitment of TBK1 by OPTN synergistically enhances NDP52-driven mitophagy in cells. This underscores the pivotal role of TBK1 recruitment by OPTN, not only for OPTN's own function but also for NDP52-mediated mitophagy, as the proximity of TBK1 recruitment by OPTN probably also augments NDP52-mediated mitophagy.

To further dissect the interplay between OPTN and NDP52, we conducted biochemical reconstitution experiments using agarose beads coated with GST–4×Ub to mimic damaged mitochondrial surfaces. We incubated these beads with OPTN, TBK1 and NAP1 in the presence or absence of NDP52. This confirmed that NAP1 negatively

**Fig. 7 | Crosstalk between the OPTN-axis and NDP52-axis stimulates mitophagy. a**, Diagram of the experimental setup and the effect of rapalog treatment, resulting in the tethering of FKBP–OPTN(2–119) to the outer mitochondrial membrane. **b**, WCL, cytosol and mitochondrial (Mito) fractions from untreated versus 3 h rapalog-treated cells were analyzed by immunoblotting for phosphorylated TBK1 (S172) and other indicated antibodies. An anti-GFP antibody was used to detect FKBP–GFP–OPTN(2–119) (*n* = 3 biologically independent experiments). **c**, PentaKO cells expressing BFP–Parkin and GFP–NDP52 were further transduced with Fis1–FRB and FKBP–EGFP–OPTN(2–119) and treated with rapalog alone, O/A alone or rapalog plus O/A for the indicated times. Mitophagy flux was measured by flow cytometry (mean ± s.d.) (*n* = 5 biologically independent experiments). Two-way ANOVA with Tukey's multiple comparisons test was performed. \*\**P* < 0.005; \*\*\*\**P* < 0.0001; ns, not significant. **d**, Biochemical reconstitution of the recruitment of GFP–TBK1 by mCherry–OPTN to GST–4×Ub-coated beads in

the presence or absence of increasing amounts of MBP–NAP1. In the indicated wells, unlabeled NDP52 was also added. Top, diagram of the experimental setup; bottom, experimental results obtained by confocal imaging (*n* = 3 biologically independent experiments). Scale bar, 100 µm. **e**, Pull-down assay of mCherry–OPTN, mCherry–NDP52 and NAP1 by GFP–TBK1. GFP–TBK1 was pre-loaded onto GFP-Trap beads and then incubated with the protein mixtures as indicated. The relative amounts of mCherry–OPTN, NAP1 and mCherry–NDP52 bound to TBK1 were quantified for the indicated lanes and one representative plot is shown (right) (*n* = 3 biologically independent experiments). **f**, Biochemical reconstitution of the recruitment of GFP–TBK1 by mCherry–OPTN to GST–4×Ub-coated beads in the presence or absence of unlabeled NDP52, MBP–NAP1 and/ or ATP/MgCl₂ (indicated as 'ATP'). Top, diagram of the experimental setup; bottom, experimental results obtained by confocal imaging (*n* = 3 biologically independent experiments). Scale bar, 100 µm. Source numerical data, including exact *P* values, and unprocessed blots are available in source data.

regulates the recruitment of TBK1 towards ubiquitin-bound OPTN in the absence of NDP52, as we showed above (Fig. 5a). However, when we added NDP52 to concentrations of NAP1 that would prevent any detectable TBK1 recruitment to the beads, we observed restoration

and even a trend towards a slight enhancement of the TBK1 signal on the beads (Fig. 7d). To confirm the specificity of the NAP1 sequestration by NDP52, we replaced wild-type NAP1 with an NDP52-binding mutant and observed complete disappearance of TBK1 signal on the

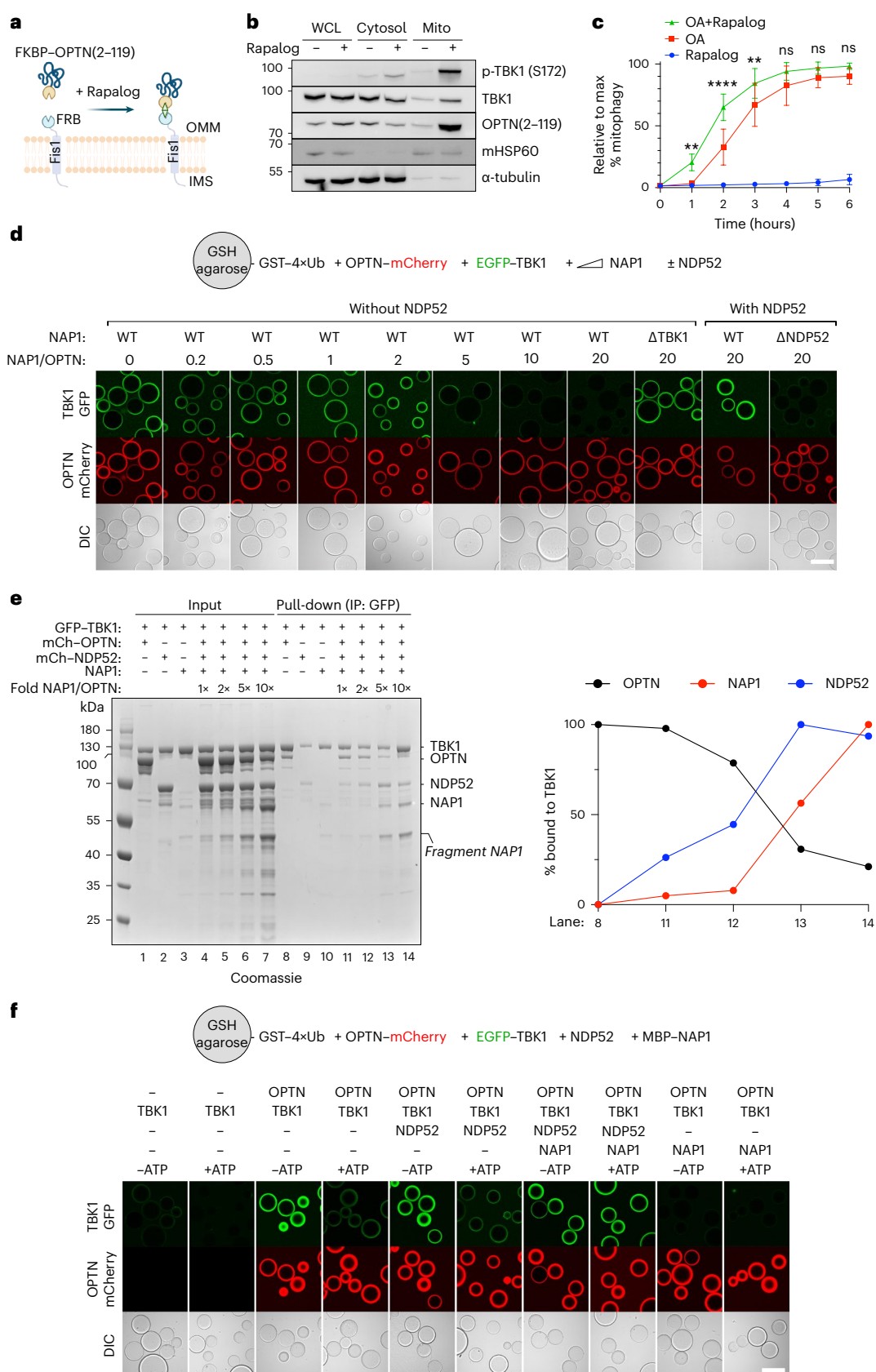

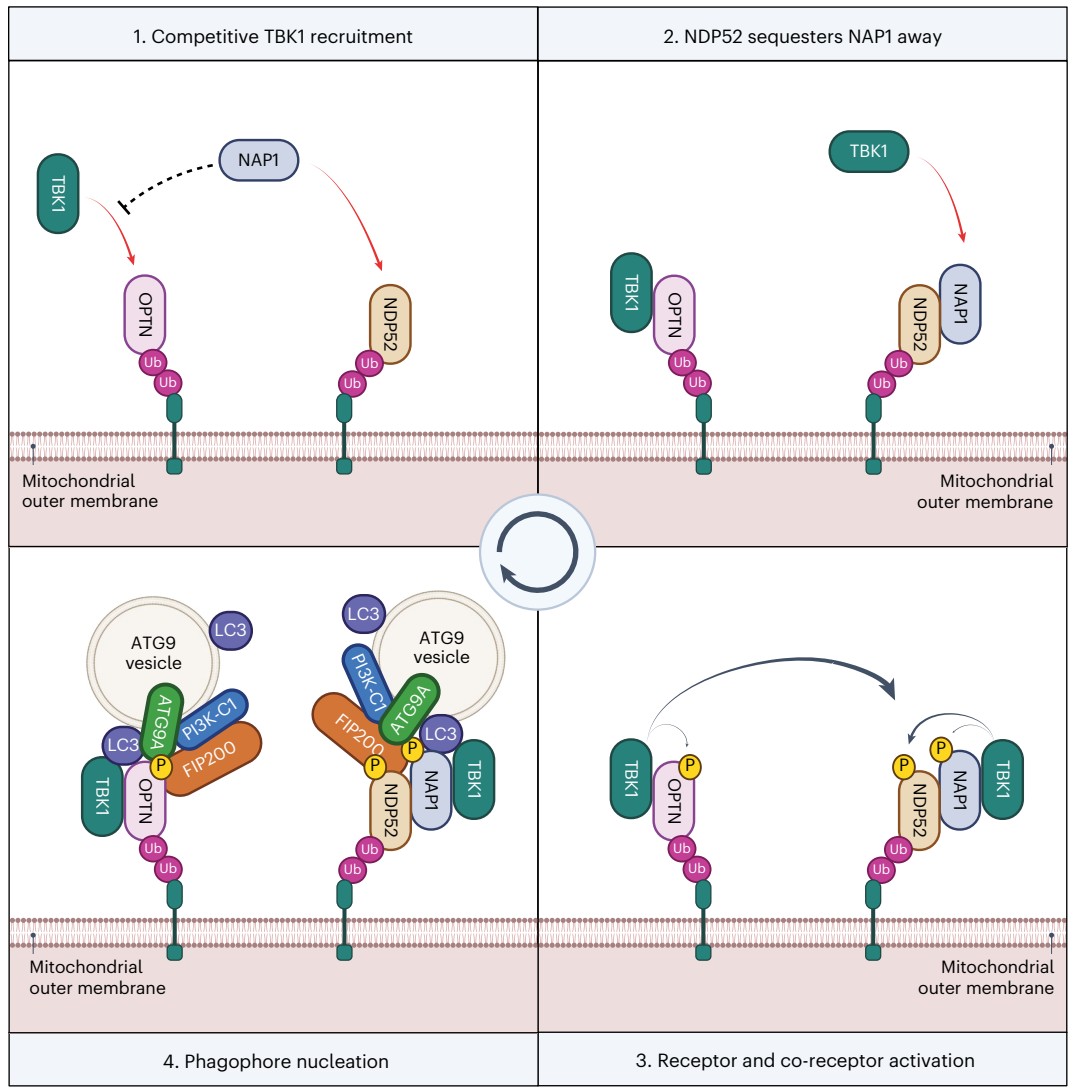

**Fig. 8 | Working model for mitophagy initiation in cells expressing both mitophagy receptors OPTN and NDP52. 1**, Cargo receptors OPTN and NDP52 are recruited to damaged mitochondria upon accumulation of ubiquitin and phospho-ubiquitin on their surface. OPTN recruits TBK1 but is restricted by cytosolic NAP1, which competes with OPTN for TBK1-binding. **2**, However, NDP52 recruits NAP1 to the mitochondrial surface and sequesters NAP1 away, allowing OPTN to recruit and activate more TBK1. **3**, Clustered TBK1 phosphorylates and activates the cargo receptors and cargo co-receptors, including crosstalk from the OPTN to the NDP52 axis. **4**, This, in return, facilitates the recruitment of downstream autophagy complexes and ATG9 vesicles, which will form the seed for the autophagosome and from where, upon lipid influx by ATG2, an expanding phagophore will arise.

ubiquitin-coated beads (Fig. 7d). Similarly, NDP52 was able to sequester SINTBAD to the ubiquitin-coated surface under conditions that would otherwise prevent TBK1 recruitment to the beads (Extended Data Fig. 9b). This suggests that the interaction of NAP1/SINTBAD with NDP52 on the cargo allows the mitophagy machinery to overcome the inhibitory effect of NAP1/SINTBAD in terms of TBK1 recruitment during mitophagy initiation.

To assess whether TBK1 indeed converted from binding the OPTN-axis to the NDP52–NAP1/SINTBAD axis, we performed a reverse pull-down experiment using GFP-trap beads coated with EGFP–TBK1. This confirmed that the relative amounts of NAP1 and OPTN determined which of the two cargo receptors, OPTN versus NDP52, was predominately recruited to TBK1 (Fig. 7e). These results suggest that NDP52 cooperates with OPTN-driven mitophagy by harnessing NAP1/SINTBAD to recruit further TBK1.

Integrating these insights from above, we reconstituted mitophagy initiation in the presence and absence of ATP/MgCl$_2$, to allow TBK1 to phosphorylate the different cargo receptors and cargo co-receptors. This revealed that upon addition of ATP/MgCl$_2$, the interaction between

OPTN and TBK1 weakened, suggesting that TBK1 is partially released after its activation through local clustering by OPTN (Extended Data Fig. 10). Adding this component to the reconstitution of mitophagy initiation, we observed that TBK1 is also released from OPTN bound to GST–4×Ub-coated beads in the presence of ATP/MgCl$_2$ (Fig. 7f). However, addition of NDP52 and NAP1, but not NDP52 alone, resulted in TBK1 being retained on the surface of the beads even in the presence of ATP/MgCl$_2$. Our data therefore support a model in which OPTN boosts NDP52 by recruiting and activating TBK1, whereas NDP52 cooperates with OPTN by using NAP1/SINTBAD to retain the locally activated and released TBK1 on the mitochondrial surface.

In summary, our findings propose a model in which NAP1/SINTBAD initially set a threshold for mitophagy activation by constraining TBK1 activation via the mitophagy receptor OPTN (Fig. 8). This is because OPTN fulfills a primary role in recruiting TBK1 during mitophagy. However, when mitochondrial damage is severe enough, NAP1/SINTBAD transition into a supportive role, acting as cargo co-receptors that bolster NDP52-driven mitophagy. Their sequestration by NDP52 increases TBK1 activation through increased recruitment by OPTN, and

this, in return, then boosts NDP52-driven mitophagy again owing to the crosstalk from OPTN–TBK1 towards the NDP52 axis, providing an effective feedforward loop once the mitophagy pathway is set in motion.

## Discussion

The regulatory mechanisms that prevent PINK1/Parkin-dependent mitophagy and other selective autophagy pathways from overreacting while ensuring swift progression once initiated are largely elusive. By focusing on the roles of the TBK1 adaptors NAP1/SINTBAD, we uncovered how tightly they are interwoven into this pathway by regulating key activities of the OPTN and NDP52 cargo receptors in completely different ways. In particular, we find that NAP1/SINTBAD act as rheostats that inhibit mitophagy initiation by restricting recruitment and activation of TBK1 by OPTN while enhancing NDP52-mediated engulfment of damaged mitochondria.

NAP1/SINTBAD drew our attention owing to the central role of TBK1 as a key regulator of selective autophagy pathways and their involvement in supporting NDP52-dependent xenophagy[7,8,10]. In addition, we found that NAP1/SINTBAD are recruited and co-degraded with damaged mitochondria (Fig. 1). The observation that deletion of NAP1/SINTBAD in wild-type HeLa cells results in acceleration rather than a deceleration of mitophagy (Fig. 2) was, therefore, unexpected. Using cellular and in vitro reconstitutions, we dissected how NAP1/SINTBAD interact with NDP52 and OPTN, the key cargo receptors in PINK1/Parkin-dependent mitophagy[30]. In NDP52-driven mitophagy, they exert a stimulatory role (Fig. 3a,b), similar to their function in xenophagy, by bridging NDP52 with TBK1 to activate this kinase on the mitochondrial surface. Furthermore, they stabilize the interaction with the core autophagy factor FIP200 (Fig. 3d). By contrast, in OPTN-driven mitophagy, NAP1/SINTBAD counteract TBK1 recruitment and activation by directly competing for the same TBK1 binding site (Fig. 5). Overall, the inhibitory role of NAP1/SINTBAD seems to prevail, as evidenced by the increased presence of activated TBK1 on damaged mitochondria in the absence of NAP1/SINTBAD in cells expressing both OPTN and NDP52 (Fig. 5e).

Our study highlights the central role of TBK1 in coordinating the different mitophagy mechanisms and uncovers an interplay between OPTN-mediated mitophagy and NDP52-mediated mitophagy. This interplay suggests a finely tuned regulation of mitophagy, which may be particularly important for specific cell types. For example, in the brain, where NDP52 expression is low and OPTN is the primary mitophagy receptor, competition for TBK1 activation may prevent excessive initiation of mitochondrial degradation. Given the post-mitotic nature of neurons, an excess in mitochondrial degradation could be as detrimental as insufficient activation. This is exemplified by disease-causing mutations in FBXL4, which results in excessive mitochondrial degradation through the NIX/BNIP3 pathway[66–69] and leads to severe mitochondrial encephalopathy[70,71]. Conversely, in cells expressing both mitophagy receptors OPTN and NDP52, NAP1/SINTBAD initially compete with OPTN for TBK1 binding until mitophagy is adequately activated. Subsequently, NAP1/SINTBAD convert into mitophagy-promoting factors by supporting NDP52. The conversion from inhibitory to stimulatory might be mediated by TBK1 phosphorylation, which warrants further investigation.

Our results also highlight the dominant role of OPTN in recruiting and activating TBK1 during mitophagy. This is in line with the recent finding that OPTN forms a platform for TBK1 activation in which it engages with TBK1 in a positive feedback loop, and observations made with the ALS-causing TBK1-E696K mutant, which has lost its OPTN-binding capacity[39,52,62] but not its binding to NAP1, as we show here. These results additionally provide a potential explanation why the phenotype of the TBK1-E696K mouse model only partially phenocopies the TBK1 knockout mouse[72]. Although NDP52 can recruit and activate TBK1 in the absence of other cargo receptors, our findings indicate that when OPTN and NDP52 are co-expressed, mitophagy is accelerated when OPTN can more easily recruit TBK1. This hints at a two-tiered mechanism, with OPTN being the first cargo receptor to drive mitophagy at very early stages, followed by NDP52 in a second phase. Although the existence of such a mechanism in mitophagy remains in part speculative at this point, previous work has indicated that OPTN and NDP52 are not kinetically interchangeable, with OPTN being more dominant for mitophagy at early time points[39]. Two-step cargo receptor recruitment has also been observed in xenophagy in which NDP52 is recruited initially to invading pathogens by recognition of exposed Galectin 8 molecules[42,73], subsequently leading to the recruitment of the E3 ubiquitin ligases, such as LUBAC and LRSAM1, which coat the bacterial surface with poly-ubiquitin chains[74–76]. This, in turn, triggers the recruitment of other cargo receptors like OPTN and SQSTM1/p62 (refs. 41,77). Future research should address whether a similar two-step recruitment mechanism or other diversification mechanisms between cargo receptors underlie our findings.

Additionally, the recent identification of TNIP1 as another mitophagy inhibitor[78] suggests that the inhibitory effect of NAP1/SINTBAD may constitute a more widespread mechanism. TNIP1 was proposed to compete with autophagy receptors for FIP200 binding, which is distinct from how NAP1/SINTBAD inhibit mitophagy, as our results demonstrate that they instead strengthen the NDP52–FIP200 interaction. Nevertheless, identifying these regulatory steps during the early steps of autophagosome biogenesis could offer new therapeutic opportunities, especially in conditions in which damaged mitochondria are insufficiently cleared.

In summary, our study uncovers an unexpected additional layer of regulation governing mitophagy initiation and expands our understanding of the complex interplay among various components involved in maintaining mitochondrial quality control. This additional layer may enable cells to respond better to cellular demands and may offer new opportunities for developing new therapeutic strategies aimed at modulating mitophagy in various pathological conditions associated with mitochondrial dysfunction.

## Online content

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

¹Max Perutz Labs, Vienna Biocenter Campus (VBC), Vienna, Austria. ²University of Vienna, Max Perutz Labs, Department of Biochemistry and Cell Biology, Vienna, Austria. ³Aligning Science Across Parkinson's (ASAP) Collaborative Research Network, Chevy Chase, MD, USA. ⁴Walter and Eliza Hall Institute of Medical Research, Parkville, Victoria, Australia. ⁵Department of Biochemistry and Molecular Biology, Biomedicine Discovery Institute, Monash University, Melbourne, Victoria, Australia. ⁶Department of Medical Biology, University of Melbourne, Melbourne, Victoria, Australia. ⁷Max Perutz Labs BioOptics FACS Facility, Max Perutz Labs, University of Vienna, Vienna BioCenter Campus (VBC), Vienna, Austria. ⁸Present address: Harry Perkins Institute of Medical Research, QEII Medical Centre, The University of Western Australia, Nedlands, Western Australia, Australia. ⁹Present address: Telethon Kids Institute, Northern Entrance, Perth Children's Hospital, Nedlands, Western Australia, Australia. ¹⁰These authors contributed equally: Elias Adriaenssens, Thanh Ngoc Nguyen. ✉e-mail: elias.adriaenssens@univie.ac.a; Lazarou.m@wehi.edu.au; sascha.martens@univie.ac.at

## Methods

### Reagents

The following chemicals were used in this study: oligomycin (A5588, ApexBio), antimycin A (A8674, Sigma-Aldrich), Q-VD-OPh (A1901, ApexBio), rapalog A/C hetero-dimerizer (635057, Takara), bafilomycin A1 (sc-201550, Santa Cruz Biotech), TBK1 inhibitor GSK8612 (S8872, Selleck Chemicals), TBK1 inhibitor BX795 (ENZ-CHM189-0005, Enzo Life Sciences), ULK1/2 inhibitor (MRT68921, BLDpharm), Vps34-IN1 inhibitor (APE-B6179, ApexBio) and DMSO (D2438, Sigma-Aldrich).

### Plasmid construction

The sequences of all cDNAs were obtained by amplifying existing plasmids, HAP1 cDNA, or through gene synthesis (Genscript). For insect cell expressions, the sequences were codon-optimized and gene-synthesized (Genscript). With the exception of the NAP1-6×Ala mutant, which was obtained through gene synthesis (Genscript), all other plasmids were generated by Gibson cloning. Gibson reactions were transformed into DH5-alpha competent *E. coli* cells (18265017, ThermoFisher). Single colonies were picked, grown overnight in liquid cultures and pelleted for DNA plasmid extraction using the GeneJet Plasmid Miniprep kit (Thermo Fisher). The purified plasmid DNA was submitted for DNA Sanger sequencing (MicroSynth). All insert sequences were verified by Sanger sequencing. Positive clones were further analyzed by whole plasmid sequencing (Plasmidsaurus). A detailed protocol is available (https://doi.org/10.17504/protocols.io.8epv5x11ng1b/v1).

### Cell lines

All cell lines were cultured at 37 °C in a humidified 5% $CO_2$ atmosphere. HeLa (RRID: CVCL_0058) and HEK293T (RRID: CVCL_0063) cells were acquired from the American Type Culture Collection (ATCC). HAP1 parental (RRID: CVCL_Y019) cells, HAP1 FIP200 knockout (RRID: CVCL_TI59) and HAP1 ATG5 knockout (RRID: CVCL_SE00) were purchased from Horizon Discovery. HeLa and HEK293T cells were grown in DMEM (Thermo Fisher) supplemented with 10% (v/v) FBS (Thermo Fisher), 25 mM HEPES (15630080, Thermo Fisher), 1% (v/v) non-essential amino acids (11140050, Thermo Fisher) and 1% (v/v) penicillin–streptomycin (15140122, Thermo Fisher). HAP1 cells were cultured in Iscove's modified Dulbecco's medium (Thermo Fisher) supplemented with 10% (v/v) FBS (Thermo Fisher) and 1% (v/v) penicillin–streptomycin (15140122, Thermo Fisher). All cell lines were tested regularly for mycoplasma contamination. A detailed protocol is available (https://doi.org/10.17504/protocols.io.n2bvj3y5blk5/v1).

### Generation of CRISPR–Cas9 knockout cells

All knockout cell lines were generated using CRISPR–Cas9. Candidate single-guide RNAs (sgRNAs) were identified using CHOPCHOP (RRID: SCR_015723; https://chopchop.cbu.uib.no). The sgRNAs were selected to target all common splicing variants. Using Gibson Cloning, the sgRNAs were ordered as short oligonucleotides (Sigma-Aldrich) and cloned into pSpCas9(BB)-2A-GFP vector (RRID: Addgene_48138). The successful insertion of the sgRNAs was verified by Sanger sequencing. A detailed description of this cloning is available (https://doi.org/10.17504/protocols.io.j8nlkkzo6l5r/v1).

Plasmids containing a sgRNA were transfected into HeLa cells with X-tremeGENE8 (Roche). Single GFP-positive cells were sorted by fluorescence-activated cell sorting (FACS) into 96-well plates. Single-cell colonies were expanded and collected for screening to identify positive clones by immunoblotting. Clones that showed a loss of protein expression for the target of interest were further analyzed by Sanger sequencing of the respective genomic regions. After DNA extraction, the regions of interest surrounding the sgRNA target sequence were amplified by PCR and analyzed by Sanger sequencing. The DNA sequences were compared to sequences from the parental line, and the edits were identified using the Synthego ICE v2 CRISPR Analysis Tool (https://www.synthego.com/products/bioinformatics/crispr-analysis).

A detailed protocol is available (https://doi.org/10.17504/protocols.io.8epv59yx5g1b/v1).

For NAP1 and SINTBAD single knockout lines, we transfected sgRNAs for the respective target genes into naive HeLa cells (RRID: CVCL_0058) to obtain NAP1 knockout clones #32 (RRID: CVCL_D2YA) and #62 (RRID: CVCL_D2YB) or SINTBAD knockout clone #7 (RRID: CVCL_D2Y9). In cases in which we generated multiple gene knockouts in the same cell line, we sequentially transfected sgRNAs for the respective target genes. For NAP1/SINTBAD DKO clones #13 (RRID: CVCL_C9DV) and #14 (RRID: CVCL_D2Q0), the cells were first transfected with NAP1 sgRNA-targeting plasmids, and positive clones were then transfected with SINTBAD sgRNA-targeting plasmids. For NAP1/SINTBAD DKO clones #20 (RRID: CVCL_C8QB) and #26 (RRID: CVCL_D2Q1) in the pentaKO background, the pentaKO line (RRID: CVCL_C2VN), first described in a previous publication[30], was transfected with NAP1 and SINTBAD sgRNA-targeting plasmids. For NAP1/SINTBAD/ULK1/2 4KO in the pentaKO background, ULK1/2 were first knocked out in the pentaKO line (RRID: CVCL_C2VS), and this cell line was then used further to delete NAP1/SINTBAD (RRID: CVCL_C9DW). An overview of the CRISPR–Cas9 knockout clones generated in this study can be found in Supplementary Table 1.

### Generation of stable cell lines

Stable cell lines were generated using lentiviral or retroviral expression systems. For retroviral transductions, HEK293T cells (RRID: CVCL_0063) were transfected with VSV-G (a kind gift from Richard Youle), Gag-Pol (a kind gift from Richard Youle) and pBMN constructs containing our gene of interest using Lipofectamine 3000 or Lipofectamine LTX (L3000008 or A12621, Thermo Fisher). The next day, the medium was exchanged with fresh media. Viruses were collected 48 h and 72 h after transfection. The retrovirus-containing supernatant was collected and filtered to avoid crossover of HEK293T cells into our HeLa cultures. HeLa cells, seeded at a density of 800,000 per well, were infected by the retrovirus-containing supernatant in the presence of 8 mg ml$^{-1}$ polybrene (Sigma-Aldrich) for 24 h. The infected HeLa cells were expanded, and 10 days after infection, they were sorted by FACS to match equal expression levels where possible. A detailed protocol is available (https://doi.org/10.17504/protocols.io.81wgbyez1vpk/v1).

The following retroviral vectors were used in this study: pBMN-HA-NAP1 (RRID: Addgene_208868), pBMN-HA-NAP1delta-TBK1 (L226Q/L233Q) (RRID: Addgene_208869), pBMN-HA-SINTBAD (RRID: Addgene_210209), pBMN-mEGFP-OPTN (RRID: Addgene_188784), pBMN-mEGFP-NDP52 (RRID: Addgene_188785), pBMN-BFP-Parkin (RRID: Addgene_186221) and pCHAC-mito-mKeima (RRID: Addgene_72342). Empty backbones used to generate these retroviral vectors were pBMN-HA-C1 (RRID: Addgene_188645), pBMN-mEGFP (RRID: Addgene_188643) and pBMN-BFP-C1 (RRID: Addgene_188644).

For lentiviral transductions, HEK293T cells (RRID: CVCL_0063) were transfected with VSV-G (a kind gift from Gijs Versteeg), Gag-Pol (a kind gift from Gijs Versteeg) and pHAGE constructs containing our gene-of-interest using Lipofectamine 3000 (L3000008, Thermo Fisher). The next day, the medium was exchanged with fresh media. Viruses were collected 48 h and 72 h after transfection. The lentivirus-containing supernatant was collected and filtered to avoid crossover of HEK293T cells into our HeLa cultures. HeLa cells, seeded at a density of 800,000 per well, were infected by the lentivirus-containing supernatant in the presence of 8 mg ml$^{-1}$ polybrene (Sigma-Aldrich) for 24 h. The infected HeLa cells were expanded, and 10 days after infection, they were used for experiments. A detailed protocol is available (https://doi.org/10.17504/protocols.io.6qpvr3e5pvmk/v1).

The following lentiviral vectors were used in this study: pHAGE–FKBP–GFP–NDP52 (RRID: Addgene_135296), pHAGE–FKBP–GFP–NAP1 (RRID: Addgene_208862), pHAGE–FKBP–GFP–NAP1delta-NDP52(S37K/A44E) (RRID: Addgene_208863), pHAGE–FKBP–GFP–NAP1delta-FIP200 (I11S/L12S) (RRID: Addgene_208864), pHAGE–FKBP–GFP–NAP1

delta-TBK1 (L226Q/L233Q) (RRID: Addgene_208865), pHAGE–FKBP–GFP–SINTBAD (RRID: Addgene_216840), pHAGE–FKBP–GFP–OPTN (RRID: Addgene_208866), pHAGE–FKBP–GFP–OPTN(2–119) (RRID: Addgene_208867) and pHAGE–mt-mKeima–P2A–FRB–Fis1 (RRID: Addgene_135295).

### Mitophagy experiments
To induce mitophagy, cells were treated with 10 µM oligomycin (A5588, ApexBio) and 4 µM antimycin A (A8674, Sigma-Aldrich). In case cells were treated for more than 8 h, we also added 10 µM Q-VD-OPh (A1901, ApexBio) to suppress apoptosis. Samples were then analyzed by SDS–PAGE and western blot or flow cytometry. A detailed protocol is available (https://doi.org/10.17504/protocols.io.n2bvj3yjnlk5/v1).

### Nutrient starvation experiments
To induce bulk autophagy, cells were starved by culturing them in Hank's balanced salt medium (Thermo Fisher). Cells were collected and analyzed by SDS–PAGE and western blot analysis. A detailed protocol is available (https://doi.org/10.17504/protocols.io.4r3l228b3l1y/v1).

### Rapalog-induced chemical dimerization experiments
The chemically induced dimerization experiments were performed using the FRB–Fis1 and FKBP fused to our gene-of-interest system. After consecutive lentiviral transduction of HeLa cells with both constructs, in which the FRB–Fis1 also expresses mt-mKeima, cells were treated with the Rapalog A/C hetero-dimerizer rapalog (635057, Takara) for 24 h. Cells were then analyzed by flow cytometry. A detailed protocol is available (https://doi.org/10.17504/protocols.io.n92ldmyynl5b/v1).

### Flow cytometry
For mitochondrial flux experiments, 700,000 cells were seeded in six-well plates 1 day before the experiment. Mitophagy was induced by treating the cells for the indicated times with a cocktail of O/A, as described above. Cells were collected by removing the medium, washing with PBS (14190169, Thermo Fisher), trypsinization (T3924, Sigma-Aldrich) and then resuspending in complete DMEM medium (41966052, Thermo Fisher). The cells were then filtered through 35 µm cell-strainer caps (352235, Falcon) and analyzed by an LSR Fortessa Cell Analyzer (BD Biosciences). Lysosomal mt-mKeima was measured using dual excitation ratiometric pH measurements at 405 nm (pH 7) and 561 nm (pH 4) lasers with 710/50 nm and 610/20 nm detection filters, respectively. Additional channels used for fluorescence compensation were BFP and GFP. Single fluorescence vector-expressing cells were prepared to adjust photomultiplier tube voltages to ensure that the signal was within detection limits and to calculate the compensation matrix in BD FACSDiva Software (RRID: SCR_001456; http://www.bdbiosciences.com/instruments/software/facsdiva/index.jsp). Depending on the experiment, we gated for BFP-positive, GFP-positive and mKeima-positive cells with the appropriate compensation. For each sample, 10,000 mKeima-positive events were collected, and data were analyzed in FlowJo (v.10.9.0) (RRID: SCR_008520; https://www.flowjo.com/solutions/flowjo). Our protocol was based on a previously described protocol (https://doi.org/10.17504/protocols.io.q26g74e1qgwz/v1).

For rapalog-induced mitophagy experiments, cells were seeded as described above and treated for 24 h with 500 nM rapalog A/C hetero-dimerizer (Takara). Cells were collected as described above, and the mt-mKeima ratio (561/405) was quantified by an LSR Fortessa Cell Analyzer (BD Biosciences). The cells were gated for GFP/mt-mKeima double-positive cells. Data were analyzed using FlowJo (v.10.9.0). A detailed protocol is available (https://doi.org/10.17504/protocols.io.n92ldmyynl5b/v1).

### Cellular fractionation and mitochondrial isolation
Mitochondria were isolated as described previously[80]. In brief, cells were collected from 15 cm dishes by trypsinization, after treatment with DMSO or O/A where indicated, and centrifuged at 300×$g$ for 5 min at 4 °C. The cell pellet was washed with PBS, and a fraction of the PBS-washed cell pellet was transferred to a new tube and lysed in RIPA buffer (50 mM Tris-HCl pH 8.0, 150 mM NaCl, 0.5% sodium deoxycholate, 0.1% SDS, 1% NP-40) supplemented by cOmplete EDTA-free protease inhibitors (11836170001, Roche) and phosphatase inhibitors (Phospho-STOP; 4906837001, Roche). This sample served as a whole cell lysate reference. The remaining PBS-washed cells were processed further for mitochondrial isolation. The PBS-washed cell pellet was resuspended in 1 ml mitochondrial isolation buffer (250 mM mannitol, 0.5 mM EGTA, and 5 mM HEPES-KOH pH 7.4) and lysed by 15 strokes with a 26.5 gauge needle (303800, Becton Dickinson). The homogenate was centrifuged twice at 600×$g$ for 10 min at 4 °C to pellet cell debris, nuclei and intact cells. The supernatant was collected and centrifuged twice at 7,000×$g$ for 10 min at 4 °C to pellet mitochondria. The supernatant was removed, and the mitochondrial pellet was resuspended in 1 ml of mitochondrial isolation buffer. The resuspended mitochondrial pellets were centrifuged two more times at 10,000×$g$ for 10 min at 4 °C. After removal of the supernatant, the pellets were resuspended in the mitochondrial isolation buffer. The final mitochondrial pellet was lysed in RIPA buffer and processed further for western blot analysis. A detailed protocol is available (https://doi.org/10.17504/protocols.io.kqdg3x4zzg25/v1).

### SDS–PAGE and western blot analysis
For SDS–PAGE and western blot analysis, we collected cells by trypsinization and centrifugation. Cell pellets were washed with PBS and lysed in RIPA buffer (50 mM Tris-HCl pH 8.0, 150 mM NaCl, 0.5% sodium deoxycholate, 0.1% SDS, 1% NP-40) supplemented by cOmplete EDTA-free protease inhibitors (11836170001, Roche) and phosphatase inhibitors (Phospho-STOP; 4906837001, Roche). After incubating for 20 min on ice, samples were cleared by centrifugation and soluble fractions were collected. Protein concentrations were adjusted for equal loading and loaded on 4–12% SDS–PAGE gels (Thermo Fisher) with PageRuler Prestained protein marker (Thermo Fisher). Proteins were transferred onto nitrocellulose membranes (RPN132D, GE Healthcare) for 1 h at 4 °C using the Mini Trans-Blot Cell (Bio-Rad). After the transfer, membranes were blocked with 5% milk powder dissolved in PBS-Tween (0.1% Tween 20) for 1 h at 20–25 °C. The membranes were incubated overnight at 4 °C with primary antibodies dissolved in the blocking buffer, washed and incubated with species-matched secondary horseradish-peroxidase-coupled antibodies diluted 1:10,000 for 1 h at room temperature. Membranes were processed for western blot detection with SuperSignal West Femto Maximum Sensitivity Substrate (34096, Thermo Fisher) and imaged with a ChemiDoc MP Imaging system (Bio-Rad). Images were analyzed with ImageJ[81] (RRID: SCR_003070; https://imagej.net/). A detailed protocol is available (https://doi.org/10.17504/protocols.io.eq2lyj33plx9/v1). Details on primary and secondary antibodies can be found in the Reporting Summary.

### Immunofluorescence and confocal microscopy
Cells were seeded on HistoGrip (Thermo Fisher) coated glass coverslips in 24-well plates. Cells were allowed to adhere overnight and treated as indicated before fixation. Cells were fixed in 4% (w/v) paraformaldehyde and diluted in 100 mM phosphate buffer for 10 min at room temperature. The paraformaldehyde was removed and samples were washed three times with 1× PBS. Cells were permeabilized with 0.1% (v/v) Triton X-100 and diluted in 1× PBS for 10 min. After permeabilization, samples were blocked for 15 min with 3% (v/v) goat serum diluted in 1× PBS with 0.1% Triton X-100. Primary antibodies, diluted in blocking buffer, were incubated with the samples for 90 min. Unbound antibodies were removed in three washing steps with 1× PBS. Secondary antibodies, diluted in blocking buffer, were incubated with the samples for 60 min. Unbound secondary antibodies were removed by three washes with 1× PBS before coverslips were mounted onto glass slides

with DABCO-glycerol mounting medium. Coverslips were imaged with an inverted Leica SP8 confocal laser scanning microscope equipped with an HC Plan Apochromat CS2 ×63/1.40 oil immersion objective (Leica Microsystems). Images were acquired in three dimensions using z-stacks, with a minimum range of 1.8 μM and a maximum voxel size of 90 nm laterally (x, y) and 300 nm axially (z), using a Leica HyD Hybrid detector (Leica Microsystems) and the Leica Application Suite X (LASX v.2.0.1) (RRID: SCR_013673; https://www.leica-microsystems.com/products/microscope-software/details/product/leica-las-x-ls). The z-stack images are displayed as maximum-intensity projections. Three images were taken for each sample. A detailed protocol is available (https://doi.org/10.17504/protocols.io.5qpvobz99l4o/v1). Details on primary and secondary antibodies can be found in the Reporting Summary.

### Protein expression in bacteria
After the transformation of the pGEX-4T1 or pETDuet vectors encoding our proteins of interest in E. coli Rosetta pLysS cells (70956-4, Novagen), cells were grown in LB medium at 37 °C until an optical density at 600 nm of 0.4 and then continued at 18 °C; once the cells reached an optical density of 0.8, protein expression was induced with 100 μM isopropyl β-D-1-thiogalactopyranoside for 16 h at 18 °C. Cells were collected by centrifugation, resuspended in lysis buffer and stored at −80 °C until purification.

### Baculovirus generation and protein expression in insect cells
To purify recombinant proteins from insect cells, we purchased gene-synthesized codon-optimized sequences from Genscript in pFastBac-Dual vectors. Constructs were used to generate bacmid DNA, using the Bac-to-Bac system, by amplification in DH10BacY cells[82]. After the bacmid DNA was verified by PCR for insertion of the transgene, we purified bacmid DNA for transfection into Sf9 insect cells (12659017, Thermo Fisher; RRID: CVCL_0549). To this end, we mixed 2,500 ng of plasmid DNA with FuGene transfection reagent (Promega) and transfected 1 million Sf9 cells seeded in a six-well plate. About 7 days after transfection, the V0 virus was collected and used to infect 40 ml of 1 million cells per ml of Sf9 cells. The viability of the cultures was closely monitored, and upon the decrease in viability and confirmation of yellow fluorescence, we collected the supernatant after centrifugation and stored this as V1 virus. For expressions, we infected 1 l of Sf9 cells, at 1 million cells per ml, with 1 ml of V1 virus. When the viability of the cells decreased to 90–95%, cells were collected by centrifugation, washed with 1× PBS and flash-frozen in liquid nitrogen. Pellets were stored at −80 °C until purification.

### Protein expression in HEK293 cells
Proteins were expressed in FreeStyle HEK293-F cells (RRID: CVCL_6642), grown at 37 °C in FreeStyle 293 Expression Medium (12338-026, Thermo). The day before transfection, cells were seeded at a density of $0.7 × 10^6$ cells per ml. On the day of transfection, a 400 ml culture was transfected with 400 μg of the MAXI-prep DNA, diluted in 13 ml of Opti-MEMR I Reduced Serum Medium (31985-062, Thermo) and 800 μg Polyethylenimine (PEI 25K; 23966-1, Polysciences), also diluted in 13 ml of Opti-MEM media. At 1 day post transfection, the culture was supplemented with 100 ml EXCELL R 293 Serum-Free Medium (14571C-1000 ml, Sigma-Aldrich). Another 24 h later, cells were collected by centrifugation at 270×g for 20 min. The pellet was washed with PBS to remove medium and then flash-frozen in liquid nitrogen. Pellets were stored at −80 °C until purification.

### Protein purification
Linear tetra-ubiquitin fused to GST (GST–4×Ub) was cloned into a pGEX-4T1 vector (RRID: Addgene_199779), expressed in E. coli as described above and resuspended in lysis buffer (50 mM Tris-HCl pH 7.4, 300 mM NaCl, 2 mM $MgCl_2$, 1 mM dithiothreitol (DTT), cOmplete

EDTA-free protease inhibitors (Roche) and DNase (Sigma-Aldrich)). Cell lysates were sonicated twice for 30 s. Lysates were cleared by centrifugation at 45,000×g for 45 min at 4 °C in a SORVAL RC6+ centrifuge with an F21S-8x50Y rotor (Thermo Scientific). The supernatant was collected and incubated with pre-equilibrated Glutathione Sepharose 4B beads (GE Healthcare) for 2 h at 4 °C with gentle shaking to bind GST–4×Ub. Samples were centrifuged to pellet the beads and remove the unbound lysate. Beads were then washed twice with wash buffer (50 mM Tris-HCl pH 7.4, 300 mM NaCl, 1 mM DTT), once with high-salt wash buffer (50 mM Tris-HCl pH 7.4, 700 mM NaCl, 1 mM DTT) and two more times with wash buffer (50 mM Tris-HCl pH 7.4, 300 mM NaCl, 1 mM DTT). Beads were incubated overnight with 4 ml of 50 mM reduced glutathione dissolved in wash buffer (50 mM Tris-HCl pH 7.4, 300 mM NaCl, 1 mM DTT) at 4 °C to elute GST–4×Ub from the beads. To collect the supernatant, the beads were collected by centrifugation. The beads were washed twice with 4 ml of wash buffer and the supernatant was collected. The supernatant fractions were pooled, filtered through a 0.45 μm syringe filter, concentrated with a 10 kDa cut-off Amicon filter (Merck Millipore) and loaded onto a pre-equilibrated Superdex 200 Increase 10/300 GL column (Cytiva). Proteins were eluted with SEC buffer (25 mM Tris-HCl pH 7.4, 150 mM NaCl, 1 mM DTT). Fractions were analyzed by SDS–PAGE and Coomassie staining. Fractions containing purified GST–4×Ub were pooled. After concentrating the purified protein, the protein was aliquoted and snap-frozen in liquid nitrogen. Proteins were stored at −80 °C. A detailed protocol is available (https://doi.org/10.17504/protocols.io.q26g7pbo1gwz/v1).

For GST–LC3B, as previously described[83], we inserted human LC3B cDNA in a pGEX-4T1 vector (RRID: Addgene_216836). The last five amino acids of LC3B were deleted to mimic the cleavage by ATG4. Proteins were expressed in E. coli as described above and purified using a similar GST-batch protocol as described for linear GST–4×Ub. A detailed protocol is available (https://doi.org/10.17504/protocols.io.3byl4qnbjvo5/v1).

For OPTN-GST, we cloned human OPTN cDNA in a pETDuet-1 vector with a carboxy-terminal GST-tag (RRID: Addgene_216843). Proteins were expressed in E. coli as described above and purified using a similar GST-batch protocol as described for linear GST–4×Ub. A detailed protocol is available (https://doi.org/10.17504/protocols.io.dm6gp3nb8vzp/v1).

For mCherry–OPTN, we cloned human OPTN cDNA in a pETDuet-1 vector with an amino-terminal 6×His tag followed by a tobacco etch virus (TEV) cleavage site (RRID: Addgene_190191). Proteins were expressed in E. coli as described above and cells were resuspended in lysis buffer (50 mM Tris-HCl pH 7.4, 300 mM NaCl, 2 mM $MgCl_2$, 5% glycerol, 10 mM Imidazole, 2 mM β-mercaptoethanol, cOmplete EDTA-free protease inhibitors (Roche), CIP protease inhibitor (Sigma-Aldrich) and DNase (Sigma-Aldrich)). Cell lysates were sonicated twice for 30 s. Lysates were cleared by centrifugation at 45,000×g for 45 min at 4 °C in a SORVAL RC6+ centrifuge with an F21S-8x50Y rotor (Thermo Scientific). The supernatant was filtered through a 0.45 μm filter and loaded onto a pre-equilibrated 5 ml His-Trap HP column (Cytiva). After His-tagged proteins were bound to the column, the column was washed with three column volumes of wash buffer (50 mM Tris-HCl pH 7.4, 300 mM NaCl, 10 mM imidazole, 2 mM β-mercaptoethanol). Proteins were then eluted with a stepwise imidazole gradient (30, 75, 100, 150, 225 and 300 mM). Fractions at 75–100 mM imidazole contained the 6×His-TEV–mCherry–OPTN and were pooled. The pooled samples were incubated overnight with TEV protease at 4 °C. After the 6×His tag was cleaved off, the protein was concentrated using a 50 kDa cut-off Amicon filter (Merck Millipore) and loaded onto a pre-equilibrated Superdex 200 Increase 10/300 GL column (Cytiva). Proteins were eluted with SEC buffer (25 mM Tris-HCl pH 7.4, 150 mM NaCl, 1 mM DTT). Fractions were analyzed by SDS–PAGE and Coomassie staining. Fractions containing purified mCherry–OPTN were pooled. After concentrating the purified protein, the protein was aliquoted and snap-frozen in liquid nitrogen.

Proteins were stored at −80 °C. A detailed protocol is available (https://doi.org/10.17504/protocols.io.4r3l225djl1y/v1).

For mCherry–NDP52, we cloned human NDP52 cDNA in a pETDuet-1 vector with an N-terminal 6×His tag followed by a TEV cleavage site (RRID: Addgene_187829). Proteins were expressed in *E. coli* as described above and purified using a similar His-Trap protocol as described for mCherry–OPTN. A detailed protocol is available (https://doi.org/10.17504/protocols.io.5qpvobdr9l4o/v1).

Human NDP52 cDNA was cloned into a pGST2 vector with an N-terminal GST tag followed by a TEV cleavage site (RRID: Addgene_187828). Proteins were expressed in *E. coli* as described above and purified using a similar GST-batch protocol as for linear GST–4×Ub. A detailed protocol is available (https://doi.org/10.17504/protocols.io.36wgq35xklk5/v1).

To purify NAP1 or GST–NAP1, human NAP1 cDNA was synthesized and cloned in a pcDNA3.1 vector (RRID: Addgene_216837), after which it was subcloned into a pGEX-4T1 vector with an N-terminal GST tag followed by a TEV cleavage site (RRID: Addgene_217610). After expression of unlabeled NAP1 in *E. coli* (which we used in Figs. 5b and 7e and Extended Data Fig. 5) or GST–NAP1 (which we used in Fig. 6e and Extended Data Fig. 4) as described above, proteins were purified using a similar GST-batch protocol as described for linear GST–4×Ub. A detailed protocol is available (https://doi.org/10.17504/protocols.io.kqdg3xk41g25/v1).

To purify NAP1–mCherry, as described previously[40], human NAP1 cDNA was subcloned together with N-terminal GST–TEV and mCherry tags into a pCAGG by the Vienna BioCenter Core Facilities Protech Facility (RRID: Addgene_198036). Proteins were expressed in HEK293 cells as described above and purified using a similar GST-batch protocol as described for linear GST–4×Ub. A detailed protocol is available (https://doi.org/10.17504/protocols.io.5jyl8jw6dg2w/v1).

To purify MBP–NAP1, human NAP1 cDNA was gene-synthesized (by Genscript) and subcloned into a pGEX-4T1 vector with an N-terminal MBP-tag followed by a TEV cleavage site before wild-type NAP1 (RRID: Addgene_208871), NAP1 delta-NDP52 (S37K/A44E) (RRID: Addgene_208872) or NAP1 delta-TBK1 (L226Q/L233Q) (RRID: Addgene_208873). After expression of MBP–TEV–NAP1 in *E. coli* (which we used for Fig. 7d) as described above, proteins were purified using a similar protocol as described for linear GST–4×Ub, except using Amylose Beads (Biolabs) instead of Glutathione Sepharose 4B Beads (GE Healthcare). A detailed protocol is available (https://doi.org/10.17504/protocols.io.ewov1q2ykgr2/v1).

To purify SINTBAD–GFP and SINTBAD–mCherry from insect cells, we purchased gene-synthesized codon-optimized GST–TEV–SINTBAD–EGFP and GST–TEV–SINTBAD–mCherry in a pFastBac-Dual vector from Genscript (RRID: Addgene_198035 and RRID: Addgene_208874). Proteins were expressed in insect cells as described above and purified using a similar GST-batch protocol as for linear GST–4×Ub. A detailed protocol is available (https://doi.org/10.17504/protocols.io.rm7vzb1o8vx1/v1).

To purify TBK1, GFP–TBK1 or mCherry–TBK1, we purchased gene-synthesized codon-optimized GST–TEV–TBK1, GST–TEV–EGFP–TBK1 and GST–TEV–mCherry–TBK1 in a pFastBac-Dual vector from Genscript (RRID: Addgene_208875, RRID: Addgene_187830, RRID: Addgene_198033) for expression in insect cells. Proteins were expressed in insect cells as described above and purified using a similar GST-batch protocol as for linear GST–4×Ub. A detailed protocol is available (https://doi.org/10.17504/protocols.io.81wgb6wy1lpk/v1).

To purify FIP200–GFP from insect cells, we purchased gene-synthesized codon-optimized GST–3C-FIP200–EGFP in a pGB-02-03 vector from Genscript (RRID: Addgene_187832). Proteins were expressed in insect cells as described above and purified using a similar GST-batch protocol as for linear GST–4×Ub. A detailed protocol is available (https://doi.org/10.17504/protocols.io.dm6gpbkq5lzp/v1).

To purify the MBP-ULK1 from HEK293 GnTI cells (RRID: CVCL_A785), we expressed the ULK1 kinase from a pCAG backbone encoding MBP–TSF–TEV–ULK1 (RRID: Addgene_171416). The protein was expressed in HEK293 cells as described above, collected 48 h post transfection and the cell pellet was resuspended in 25 ml lysis buffer (50 mM Tris-HCl pH 7.4, 200 mM NaCl, 1 mM MgCl$_2$, 10% glycerol, 0.5% CHAPS, 1 mM TCEP, 1 µl benzonase (Sigma-Aldrich), cOmplete EDTA-free protease inhibitors (Roche), CIP protease inhibitor (Sigma-Aldrich)). Cells were homogenized with a douncer. Cell lysates were cleared by centrifugation at 10,000×*g* for 45 min at 4 °C with a SORVAL RC6+ centrifuge with an F21S-8x50Y rotor (Thermo Scientific). The soluble supernatant was collected and loaded on a StrepTrap 5 ml HP column for binding of the Twin-Strep-tagged ULK1 protein, washed with six column volumes of wash buffer (50 mM Tris-HCl pH 7.4, 200 mM NaCl, 1 mM DTT) and eluted with elution buffer (50 mM Tris-HCl pH 7.4, 200 mM NaCl, 1 mM DTT and 2.5 mM desthiobiotin). Fractions were analyzed by SDS–PAGE and Coomassie staining. Fractions containing MBP–ULK1 were pooled and concentrated with a 50 kDa cut-off Amicon filter (Merck Millipore). The proteins were loaded onto a pre-equilibrated Superose 6 Increase 10/300 GL column (Cytiva). Proteins were eluted with SEC buffer (25 mM HEPES pH 7.5, 150 mM NaCl, 1 mM DTT). Fractions were analyzed by SDS–PAGE and Coomassie staining. Fractions containing purified MBP–ULK1 were pooled. After concentrating the purified protein, the protein was aliquoted and snap-frozen in liquid nitrogen. Proteins were stored at −80 °C. A detailed protocol is available (https://doi.org/10.17504/protocols.io.bvn2n5ge).

## Microscopy-based bead assay

Glutathione Sepharose 4B beads (GE Healthcare) were used to bind GST-tagged bait proteins. To this end, 20 µl of beads were washed twice with distilled water (dH$_2$O) and equilibrated with bead assay buffer (25 mM Tris-HCl pH 7.4, 150 mM NaCl, 1 mM DTT). Beads were then resuspended in 40 µl bead assay buffer, to which bait proteins were added at a final concentration of 5 µM. Beads were incubated with the bait proteins for 1 h at 4 °C at a horizontal tube roller. Beads were then washed three times to remove unbound GST-tagged bait proteins and resuspended in 30 µl bead assay buffer. Where indicated, we also added MgCl$_2$ and ATP to the buffer to allow the phosphorylation of targets by TBK1. Glass-bottom 384-well microplates (Greiner Bio-One) were prepared with 20 µl samples containing prey proteins at the concentrations described below and diluted in bead assay buffer, and 3 µl of beads were added per well. For the experiments in Fig. 3d, NDP52 was used at a final concentration of 50 nM, FIP200–GFP, SINTBAD–mCherry and TBK1 were used at a final concentration of 100 nM. For Fig. 5b, mCherry–OPTN and GFP–TBK1 were used at a final concentration of 250 nM, and NAP1 was used from 100 nM to 10 µM. For Fig. 6e, GFP–TBK1 was used at a final concentration of 250 nM. For Fig. 7d, mCherry–OPTN, GFP–TBK1 and unlabeled NDP52 were used at a final concentration of 250 nM, while wild-type and mutant forms of NAP1 were used from 100 nM to 10 µM. For Fig. 7f, mCherry–OPTN, GFP–TBK1 and unlabeled NDP52 were used at a final concentration of 250 nM, while wild-type MBP–NAP1 was used at 10 µM. For Extended Data Fig. 3b, NDP52–mCherry, FIP200–GFP and mCherry–TBK1 were used at a final concentration of 250 nM. For Extended Data Fig. 9b, unlabeled OPTN, GFP–TBK1 and unlabeled NDP52 were used at a final concentration of 250 nM, while wild-type mCherry–SINTBAD was used from 100 nM to 10 µM. For Extended Data Fig. 10, GFP–TBK1 was used at a final concentration of 250 nM. The beads were incubated with the prey proteins for 30 min before imaging, with the exception of Fig. 3d and Extended Data Fig. 3, in which proteins were co-incubated for 4 h and 1 h, respectively before imaging. Samples were imaged with a Zeiss LSM 700 confocal microscope equipped with Plan Apochromat ×20/0.8 WD 0.55 mm objective and using the Zeiss ZEN Microscopy software (v.2022; RRID: SCR_013672). Three biological replicates were performed for each experimental condition.

For quantification, we used an artificial intelligence script that automatically quantifies signal intensities from microscopy images by drawing line profiles across beads and recording the difference between the minimum and maximum gray values along the lines. The artificial intelligence was trained to recognize beads using cellpose[84]. Processing is composed of two parts, with the first operating in batch mode. Multichannel input images are split into individual TIFF images and passed to cellpose (running in a Python environment) (RRID: SCR_008394; http://www.python.org). The labeled images produced by cellpose are re-assembled into multichannel images. Circular regions of interest (ROIs) are fitted to the segmented particles, and a pre-defined number of line profiles (here set to 20) are drawn automatically, starting at the center of the ROI and extending beyond the border of the circular ROI. This results in line profiles from the center of the bead into the inter-bead space of the well, allowing us to quantify the signal intensities at the rim of the beads. To prevent line profiles from protruding into adjacent beads, a combined ROI containing all beads was used. The results generated by artificial intelligence were inspected manually for undetected beads, incorrect line profiles or false-assigned bead structures. For each bead, a mean fluorescence and standard deviation are obtained based on the 20 line profiles per bead. Beads with standard deviations equal to or greater than half the mean value were either excluded or subjected to manual inspection for correction. To correct for inter-experiment variability in absolute values, the mean values for each bead were divided by the average bead intensity of the control condition. These values are then plotted and subjected to statistical significance calculations. A detailed protocol is available (https://doi.org/10.17504/protocols.io.14egn38pzl5d/v1).

## GFP pull-down assay

GFP-tagged TBK1 was mixed with 20 µl of GFP-Trap agarose beads (Chromotek) at a final concentration of 1 µM. To this end, 20 µl of beads were washed twice with dH$_2$O and equilibrated with bead assay buffer (25 mM Tris-HCl pH 7.4, 150 mM NaCl, 1 mM DTT). Beads were then resuspended in 40 µl bead assay buffer, to which GFP–TBK1 was added at a final concentration of 5 µM. Beads were incubated with GFP–TBK1 for 1 h at 4 °C at a horizontal tube roller. Beads were washed three times to remove unbound GFP-tagged bait protein. Protein master mixes with prey protein were prepared in bead assay buffer at the following concentrations: mCherry–OPTN (1 µM); mCherry–NDP52 (1 µM); GST–NAP1 (1–10 µM). The protein master mixes were added to the beads and incubated for 1 h at 4 °C at a horizontal tube roller. Beads were washed three times to remove unbound proteins, diluted in 60 µl of 1× protein loading dye and heat-inactivated at 95 °C for 5 min. Samples were analyzed by SDS–PAGE and Coomassie staining as described above. A detailed protocol is available (https://doi.org/10.17504/protocols.io.e6nvwd6x2lmk/v1).

## In vitro kinase assay

Recombinant proteins TBK1 and NAP1 or MBP–ULK1 and NAP1 were mixed in kinase buffer (20 mM Tris-HCl pH 7.4, 150 mM NaCl, 1 mM DTT). The kinases were used at 50 nM and mixed with 250 nM NAP1. The kinase reactions were started by the addition of 2× ATP/MgCl$_2$ kinase buffer to a final concentration of 10 mM MgCl$_2$ and 100 mM ATP. Protein mixtures were prepared as master mixes and divided over the number of time points. To control for potential protein instability, we induced the latest time point first and then went gradually to the shortest time point. In this way, all protein mixtures were kept at room temperature for the same time, and reactions could be terminated together. Termination of reactions was achieved by the addition of 6× protein loading dye and heat inactivation at 95 °C for 5 min. Samples were separated on 4–12% SDS–PAGE gels (NP0321BOX, NP0322BOX or NP0323BOX, Thermo Fisher) with PageRuler Prestained protein marker (Thermo Fisher). After the run, the SDS–PAGE gel was either stained with Coomassie or transferred to nitrocellulose membranes

for western blot analysis. In the case of Coomassie staining, the gel was incubated for 10 min in Coomassie solution, fixed for 10 min with fixation solution (40% ethanol, 10% acetic acid, 50% dH$_2$O) and then destained overnight in dH$_2$O. The band corresponding to NAP1 was cut from the gel with a fresh scalpel and submitted for mass spectrometry analysis. In the case of western blotting, the proteins were transferred onto nitrocellulose membranes (RPN132D, GE Healthcare) for 1 h at 4 °C using the Mini Trans-Blot Cell (Bio-Rad). The membranes were then processed further for western blot analysis, as described above. A detailed protocol for in vitro kinase assays is available (https://doi.org/10.17504/protocols.io.4r3l225xjl1y/v1).

## Immunoprecipitation

FIP200 knockout HAP1 cells were transiently transfected with pcDNA3.1 NAP1–EGFP (RRID: Addgene_216837) or empty pcDNA3.1 vector as a negative control, using Lipofectamine 2000 (Thermo Fisher). This cell line was selected as FIP200 deletion results in TBK1 hyperactivation and thus increased NAP1 phosphorylation. After 48 h, cells were collected by trypsinization and the cell pellet was washed with PBS once before cells were lysed in lysis buffer (100 mM KCl, 2.5 mM MgCl$_2$, 20 mM Tris-HCl pH 7.4, 0.5% NP-40). Samples were lysed for 20 min on ice before cell lysates were cleared by centrifugation at 20,000×$g$ for 10 min at 4 °C. Protein concentrations of the cleared protein lysates were then determined with the Pierce Detergent Compatible Bradford Assay Kit (23246, Thermo Fisher). For both samples, negative control and NAP1–EGFP lysates, 12 mg of cell lysate was incubated overnight with 20 µl or GFP-Trap agarose beads (GTA-20, Chromotek). In the morning, samples were washed three times in lysis buffer before the beads were resuspended in protein loading dye, supplemented with 100 mM DTT and boiled for 5 min at 95 °C. Samples were loaded on 4–12% SDS–PAGE gels (NP0322BOX, Thermo Fisher) with PageRuler Prestained protein marker (Thermo Fisher). After the run, the SDS–PAGE gel was stained with Coomassie and destained overnight. The band corresponding to NAP1–EGFP was cut from the gel with a fresh scalpel and submitted for mass spectrometry analysis. A detailed protocol is available (https://doi.org/10.17504/protocols.io.5jyl8pmndg2w/v1).

## Sample preparation for mass spectrometry analysis

Coomassie-stained gel bands were prepared and analyzed by mass spectrometry as previously described[40]. In brief, bands were cut out and destained with a mixture of acetonitrile and 50 mM ammonium bicarbonate. Disulfide bridges were reduced using DTT, and free SH-groups were subsequently alkylated by iodoacetamide. The digestion with trypsin (Trypsin Gold, Mass Spec Grade; Promega, V5280) was carried out overnight at 37 °C, while the digestion with chymotrypsin was carried out at 25 °C for 5 h. Then the digestion was stopped by adding 10% formic acid to an end concentration of approximately 5%. Peptides were extracted from the gel with 5% formic acid by repeated sonication.

## Liquid chromatography–mass spectrometry analysis

Peptides were separated on an Ultimate 3000 RSLC nano-HPLC system using a pre-column for sample loading (Acclaim PepMap C18, 2 cm × 0.1 mm, 5 µm), and a C18 analytical column (Acclaim PepMap C18, 50 cm × 0.75 mm, 2 µm; all HPLC parts Thermo Fisher Scientific), applying a linear gradient from 2% to 35% solvent B (80% acetonitrile, 0.08 % formic acid; solvent A, 0.1 % formic acid) at a flow rate of 230 nl min$^{-1}$ over 60 min. A Q Exactive HF-X Orbitrap mass spectrometer (Thermo Fisher) coupled to the HPLC via Proxeon nano-spray-source (all Thermo Fisher Scientific) equipped with coated emitter tips (New Objective) was used with the following settings. The mass spectrometer was operated in data-dependent acquisition mode. Survey scans were obtained in a mass range of 375–1500 $m/z$ with lock mass on, at a resolution of 120,000 at 200 $m/z$ and a normalized AGC target of 3E6. The 15 most intense ions were selected with an isolation width of 1.2 $m/z$,

for a maximum of 100 ms at a normalized AGC target of 1E5 and then fragmented in the HCD cell at 28% normalized collision energy. Spectra were recorded at a resolution of 30,000. Peptides with a charge of +1 or >+7 were excluded from fragmentation, the peptide match feature was set to 'preferred' and the exclude isotope feature was enabled. Selected precursors were dynamically excluded from repeated sampling for 20 s. An Exploris 480 Orbitrap mass spectrometer (Thermo Fisher) coupled to the column with a FAIMS Pro ion source (Thermo-Fisher) using coated emitter tips (PepSep, MSWil) was used with the following settings. The mass spectrometer was operated in data-dependent acquisition mode with four FAIMS compensation voltages set to −35, −45, −60 or −70 and 0.8 s cycle time per compensation voltage. The survey scans were obtained in a mass range of 350–1500 $m/z$, at a resolution of 60,000 at 200 $m/z$ and a normalized AGC target at 100%. The most intense ions were selected with an isolation width of 1.2 $m/z$, fragmented in the HCD cell at 28% collision energy and the spectra recorded for a maximum of 100 ms at a normalized AGC target of 100% and a resolution of 30,000. Peptides with a charge of +2 to +6 were included for fragmentation, the peptide match feature was set to preferred, the exclude isotope feature was enabled and selected precursors were dynamically excluded from repeated sampling for 20 s. In addition, four doubly charged phosphorylated peptides with $m/z$ values of 909.8989, 956.4566, 797.3509 and 898.8982 were added to an inclusion list.

## Mass spectrometry data analysis

HFx raw files were directly used, while Exploris raw files were first split according to compensation voltages (−35, −45, −60, −70) using FreeStyle v.1.7 software (Thermo Scientific; https://www.thermofisher.com/us/en/home/technical-resources/technical-reference-library/mass-spectrometry-support-center/liquid-chromatography-mass-spectrometry-software-support/freestyle-software-support/freestyle-software-support-troubleshooting.html#:~:text=FreeStyle%20is%20free%20of%20charge). They were searched using MaxQuant[85] software v.1.6.17.0 (RRID: SCR_014485; http://www.biochem.mpg.de/5111795/maxquant) against the Uniprot human proteome database (release 2021_03) and a database of common laboratory contaminants. Enzyme specificity was set to 'Trypsin/P' with two miss cleavages or 'chymotrypsin+' with four miss cleavages and the minimal peptide length was set to seven. Carbamidomethylation of cysteine was searched as a fixed modification. 'Acetyl (Protein N-term)', 'Oxidation (M)', 'Phospho (STY)' were set as variable modifications. All data were filtered at 1% PSM+protein+site FDR; reverse hits were removed. The identification and quantification information of sites and proteins was obtained from the MaxQuant 'Phospho (STY) Sites' and 'ProteinGroups' tables.

## Quantification and statistical analysis

For the quantification of immunoblots, we performed a densitometric analysis using Fiji software (RRID: SCR_002285; http://fiji.sc). Graphs were plotted using GraphPad Prism v.9.5.1 (RRID: SCR_002798). For the quantification of microscopy-based bead assays, we used an in-house developed artificial intelligence tool to automate the recognition and quantification of the signal intensity for each bead, which resulted in a mean bead intensity value. These values were plotted and subjected to statistical testing. Depending on the number of samples, and as specified in the figure legends, we used either a Student's $t$-test or a one-way or two-way ANOVA test with appropriate multiple comparison tests. Statistical significance is indicated as *$P < 0.05$; **$P < 0.005$; ***$P < 0.001$; ****$P < 0.0001$; ns, not significant. Error bars are reported as means ± s.d. To ensure the reproducibility of experiments not quantified or subjected to statistical analysis, we showed one representative replicate in the paper of at least three replicates with similar outcomes for the main figures or at least two replicates for supplementary figures, as indicated in figure legends.

## Reporting summary

Further information on research design is available in the Nature Portfolio Reporting Summary linked to this article.

## Data availability

Raw files associated with this work are available on Zenodo (https://doi.org/10.5281/zenodo.10637353)[86]. The mass spectrometry proteomics data have been deposited to the ProteomeXchange Consortium via the PRIDE partner repository[87] with the dataset identifier PXD049184. For screening CRISPR knockout clones, we used the Human Genome Variation Society (HGVS; http://varnomen.hgvs.org). The mass spectrometry dataset was analyzed using the Uniprot human proteome database (release 2021_03) and a database of common laboratory contaminants. Further information on the research design is available in the Nature Research Reporting Summary linked to this article. Source data are provided with this paper.

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

## Acknowledgements

We thank members of the Martens lab, Lazarou lab, J. H. Hurley, E. L. F. Holzbaur, E. Park, G. Hummer, D. Fracchiolla and other members of the Aligning Science Across Parkinson's (ASAP) Team for their advice and helpful discussions. We thank the Max Perutz Labs BioOptics, Flow Cytometry and Mass Spectrometry facilities for their technical support. Proteomics analyses were performed by the Mass Spectrometry Facility at Max Perutz Labs using the Vienna BioCenter Core Facilities (VBCF) instrument pool. We thank J. Neuhold and the rest of the VBCF Protech Facility for help with cloning and HEK cell expressions. We thank members from the Versteeg lab for training and assistance with lentiviral work. The summarizing schematic was generated with BioRender. This work was supported by a Marie Skłodowska–Curie MSCA Postdoctoral fellowship (101062916 to E.A.), a travel grant from the Flanders Fund for Scientific Research (FWO-Flanders to E.A.), the National Health and Medical Research Council (GNT1106471 to M.L.), the Australian Research Council Discovery Project (DP200100347 to M.L.) and the Rebecca Cooper Foundation Fellowship (RC20241396 to M.L.). This research was funded in whole or in part by ASAP (ASAP-000350 to S.M. and M.L.) through the Michael J. Fox Foundation for Parkinson's Research. For the purpose of open access, the authors have applied a CC-BY 4.0 public copyright license to all Author Accepted Manuscripts arising from this submission.

## Author contributions

E.A., M.L. and S.M. conceived the project. E.A., T.N.N., B.S.P., M.L. and S.M. designed the experiments. E.A., T.N.N., J.S.M., G.K., M.S., S.S., E.M.W., K.D.C. and B.S.P. performed the experiments. E.A. and S.M. wrote the original draft to which all authors contributed by editing and reviewing.

## Competing interests

S.M. is a member of the scientific advisory board of Casma Therapeutics. M.L. is a member of the scientific advisory board and co-founder of Automera. All other authors have no competing interests to declare.

## Additional information

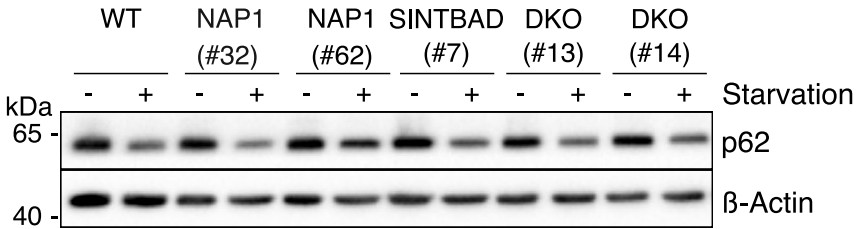

**Extended Data Fig. 1 | NAP1 and SINTBAD are not essential for non-selective starvation autophagy.** Immunoblotting of p62 levels in wild-type (WT), NAP1 knockout, SINTBAD knockout, and NAP1/SINTBAD double knockout (DKO) HeLa cells expressing YFP-Parkin, untreated or treated with EBSS starvation medium for 8 h ($n$ = 3 biologically independent experiments). Unprocessed blots are available in source data.

**A**

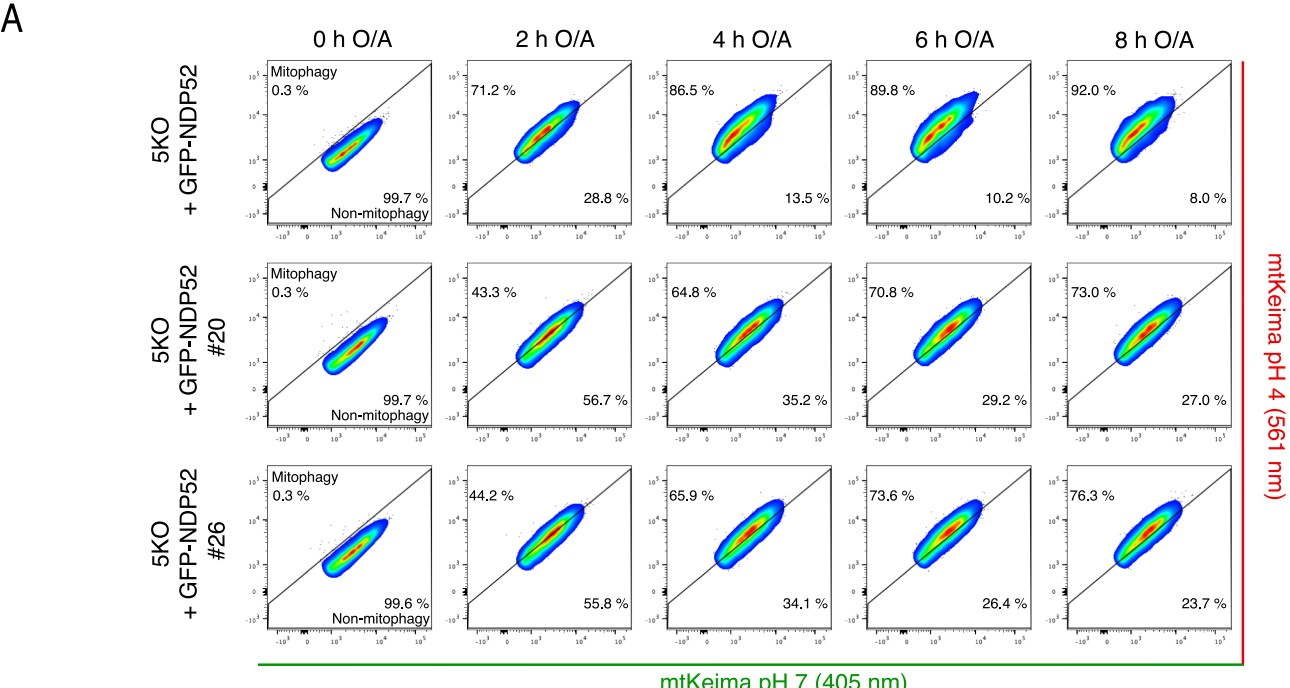

**B**

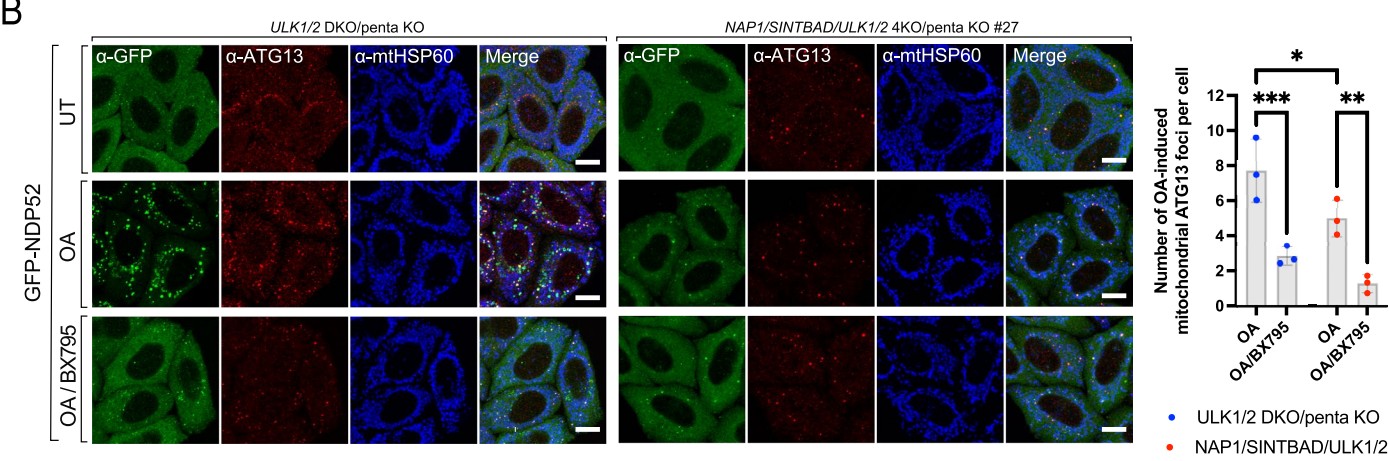

**Extended Data Fig. 2 | Mt-mKeima mitophagy flux assay and immunofluorescence assay in pentaKO cells rescued with NDP52 and in the presence or absence of NAP1/SINTBAD.** (a) Representative replicate showing reduced mitophagy initiation in both NAP1/SINTBAD DKO clones (#20 and #26) in a pentaKO (5KO) background, expressing BFP-Parkin, and rescued with GFP-NDP52. Cells were either untreated (time point 0 h) or treated with O/A for the indicated times. The mt-mKeima signal was analyzed by flow cytometry and quantified. The percentage of mitophagy-induced cells (upper left) is quantified ($n$ = 3 biologically independent experiments). (**b**) PentaKO with ULK1/2 DKO and pentaKO with ULK1/2/NAP1/SINTBAD 4KO HeLa cells stably expressing BFP-Parkin were left untreated or treated with O/A or O/A plus TBK1 inhibitor (BX795) for 1 h, and immunostained with indicated antibodies. Note the defect in GFP-NDP52 recruitment in #27 due to failure of recruiting downstream ATG8-molecules, which feedback and stabilize NDP52[86]. Number of mitochondrial ATG13 foci per cell was quantified (mean ± s.d.) ($n$ = 3 biologically independent experiments). A two-way ANOVA with Fisher's test was performed. *$P$ < 0.05, **$P$ < 0.005, ****$P$ < 0.0001. Scale bars are 10 μm. Source numerical data, including exact $P$ values, are available in source data.

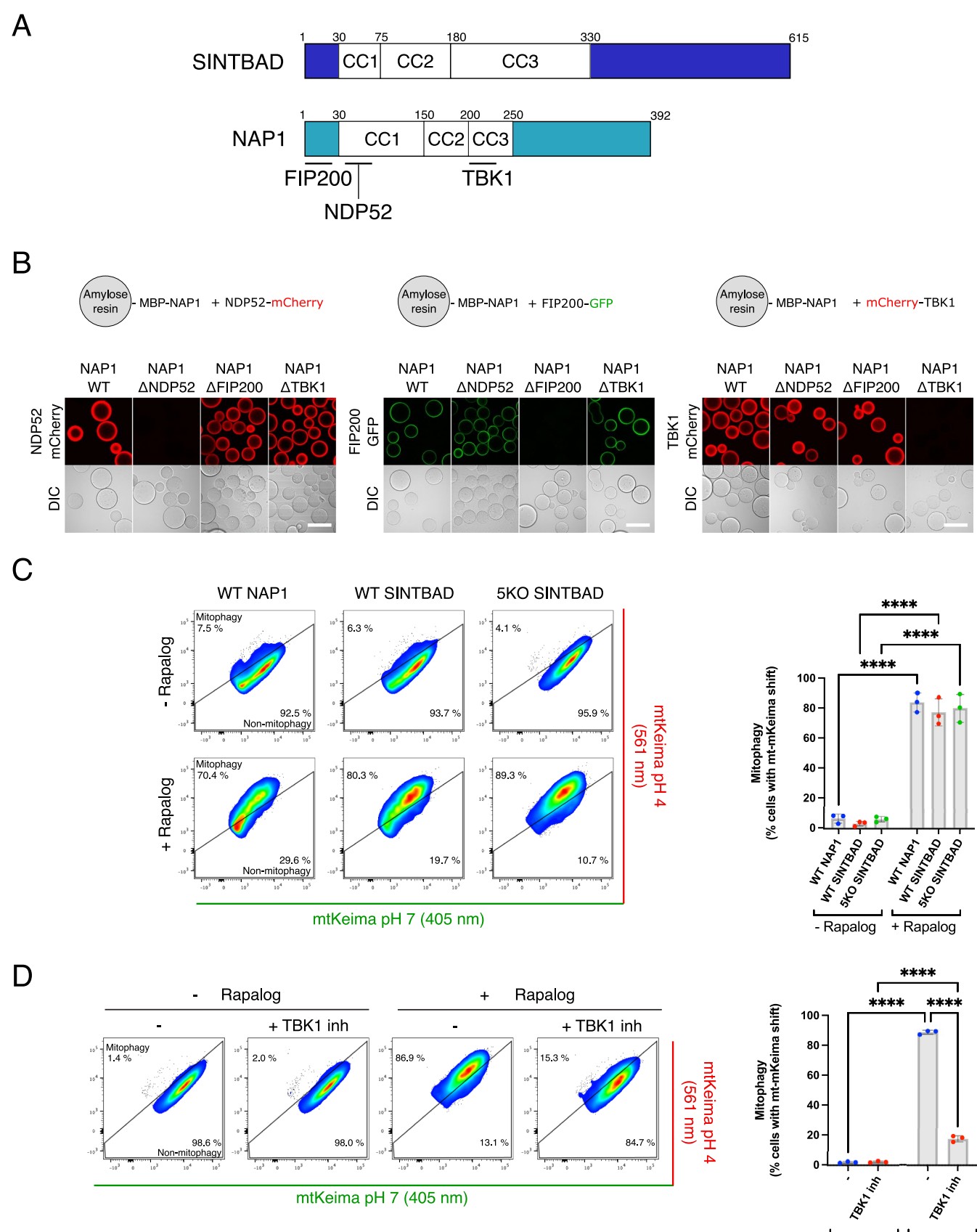

**Extended Data Fig. 3 | See next page for caption.**

**Extended Data Fig. 3 | Validation of NAP1 mutants and side-by-side comparison of SINTBAD.** (**a**) Diagram of the domain structure of NAP1 and SINTBAD, with binding sites for NDP52, FIP200, and TBK1 indicated. Coiled coil (CC) domains are also indicated based on the predicted structure by AF2. (**b**) Microscopy-based bead assay to validate point mutants in NAP1 in the binding regions for NDP52 (S37K/A44E), FIP200 (I11L/L12S), and TBK1 (L226Q/L233Q). These loss-of-binding mutants were employed in Fig. 4. (**c**) Mitophagy flux was measured by flow cytometry in wild-type (WT) or pentaKO (5KO) HeLa cells expressing BFP-PARKIN, FRB-Fis1, FKBP-GFP-NAP1 or FKBP-GFP-SINTBAD, and mt-mKeima, not induced or induced for 24 h by rapalog treatment. The percentage of mitophagy-induced cells (upper left) is quantified (mean ± s.d.) (*n* = 3 biologically independent experiments). Scale bars are 100 μm. Two-way ANOVA with Tukey's multiple comparisons test was performed. ****$P < 0.0001$. (**d**) As in (C) for FKBP-GFP-SINTBAD, with the pentaKO background, but with and without the addition of the TBK1 inhibitor (GSK8612) (mean ± s.d.) (*n* = 3 biologically independent experiments). Two-way ANOVA with Tukey's multiple comparisons test was performed. ****$P < 0.0001$. Source numerical data, including exact *P* values, are available in source data.

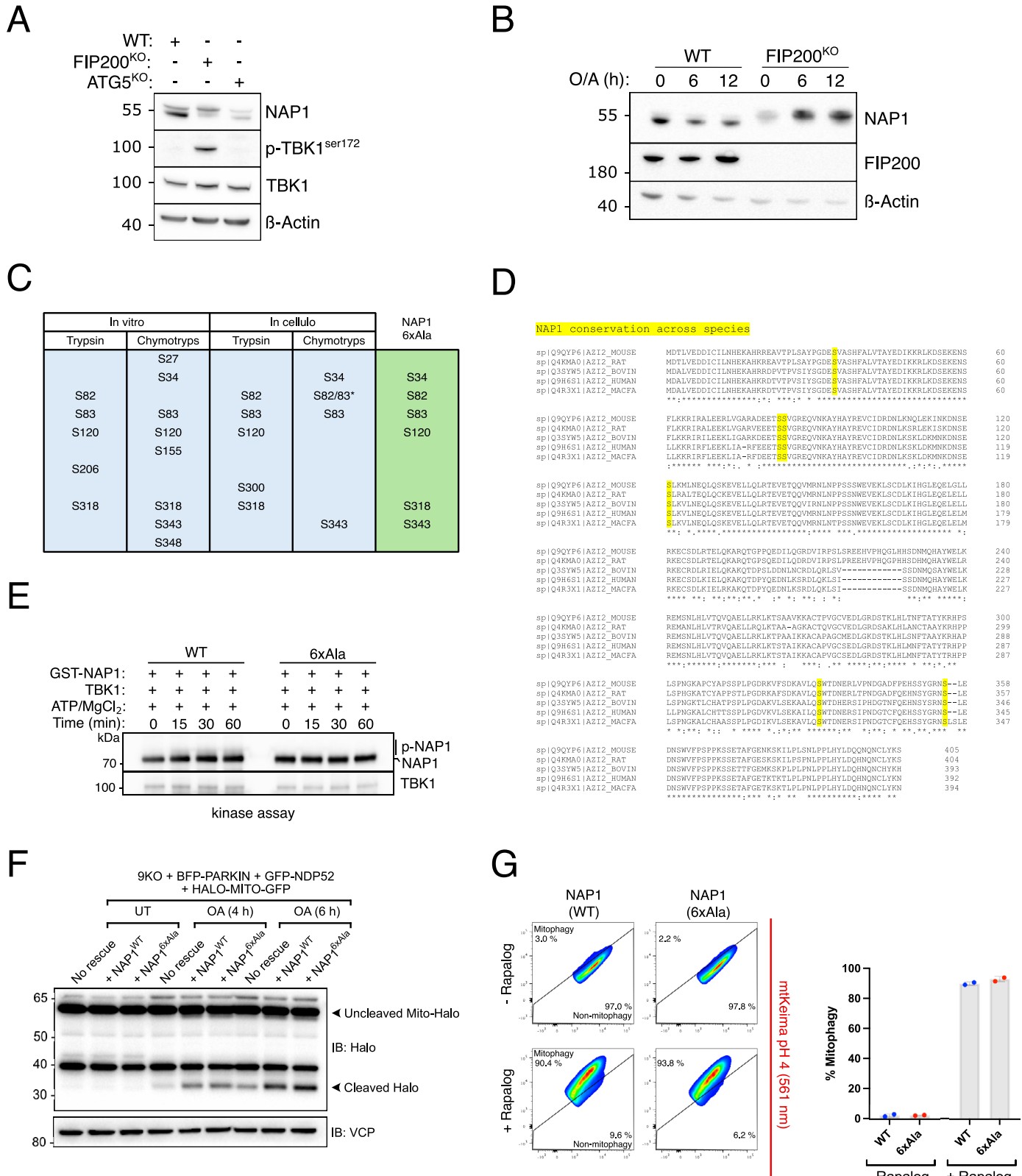

**Extended Data Fig. 4 | See next page for caption.**

**Extended Data Fig. 4 | Identification and validation of NAP1 phosphorylation sites.** (**a**) Western blot analysis of cell lysates from wild-type, FIP200 knockout (KO), and ATG5 knockout (KO) HAP1 cells. The shift in the NAP1 band in the FIP200 knockout line suggests phosphorylation ($n = 2$ biologically independent experiments). (**b**) Wild-type and FIP200 knockout HAP1 cells were treated with O/A for the indicated time and analyzed via immunoblotting with indicated antibodies ($n = 3$ biologically independent experiments). (**c**) Summary of the mass spectrometry results obtained by analyzing in vitro phosphorylation of NAP1 by TBK1 or immunoprecipitated as GFP-NAP1 from FIP200 knockout HAP1 cells. Samples were treated with trypsin or chymotrypsin (chymotryp) to maximize the sequence coverage, as detected by the mass spectrometer. Only phosphorylation sites that were detected in both the in vitro sample and in cells were retained (see green column) and mutated from serine to alanine. * Indicates a peptide on which the phosphorylation residue could not be determined with certainty between Ser82 and Ser83. ($n = 1$ biologically independent samples per condition analyzed by mass spectrometry). (**d**) Conservation analysis of the six NAP1 residues detected both in vitro and in cells. The residues are marked in yellow. (**e**) In vitro phosphorylation assay of wild-type NAP1 and 6xAla NAP1 in which the six residues identified by mass spectrometry are mutated from serine to alanine. Samples were incubated with TBK1 in presence or absence of ATP/MgCl$_2$ for the indicated time and analyzed by SDS-PAGE and immunoblotting for the indicated antibodies ($n = 3$ biologically independent experiments). (**f**) PentaKO with ULK1/2/NAP1/SINTBAD 4KO (9KO) expressing BFP-Parkin, GFP-NDP52, and mito-Halo were treated with O/A for the indicated time and analyzed by immunoblotting for the amount of lysosomal-cleaved Halo ($n = 3$ biologically independent experiments). (**g**) Mitophagy flux was measured by flow cytometry in pentaKO (5KO) HeLa cells expressing BFP-Parkin and mt-mKeima, not induced or induced for 24 h by rapalog treatment. The percentage of mitophagy-induced cells (upper left) is quantified (mean ± s.d.) ($n = 2$ biologically independent experiments). Unprocessed blots are available in source data.

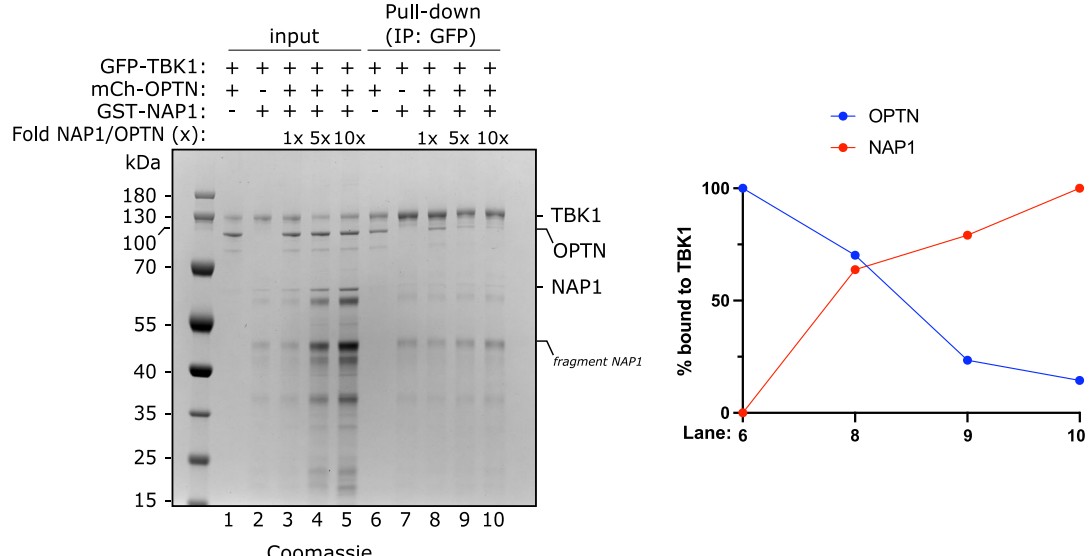

**Extended Data Fig. 5 | Pull-down of GFP-TBK1 with mCherry-OPTN and GST-NAP1.** Pull-down assay of mCherry-OPTN versus GST-NAP1 by GFP-TBK1. GFP-TBK1 was pre-loaded onto GFP-Trap beads and then incubated with the protein mixtures as indicated. The relative amounts of mCherry-OPTN and GST-NAP1 bound to TBK1 were quantified for the indicated lanes and one representative plot is shown (right) ($n = 3$ biologically independent experiments). Source numerical data and unprocessed gels are available in source data.

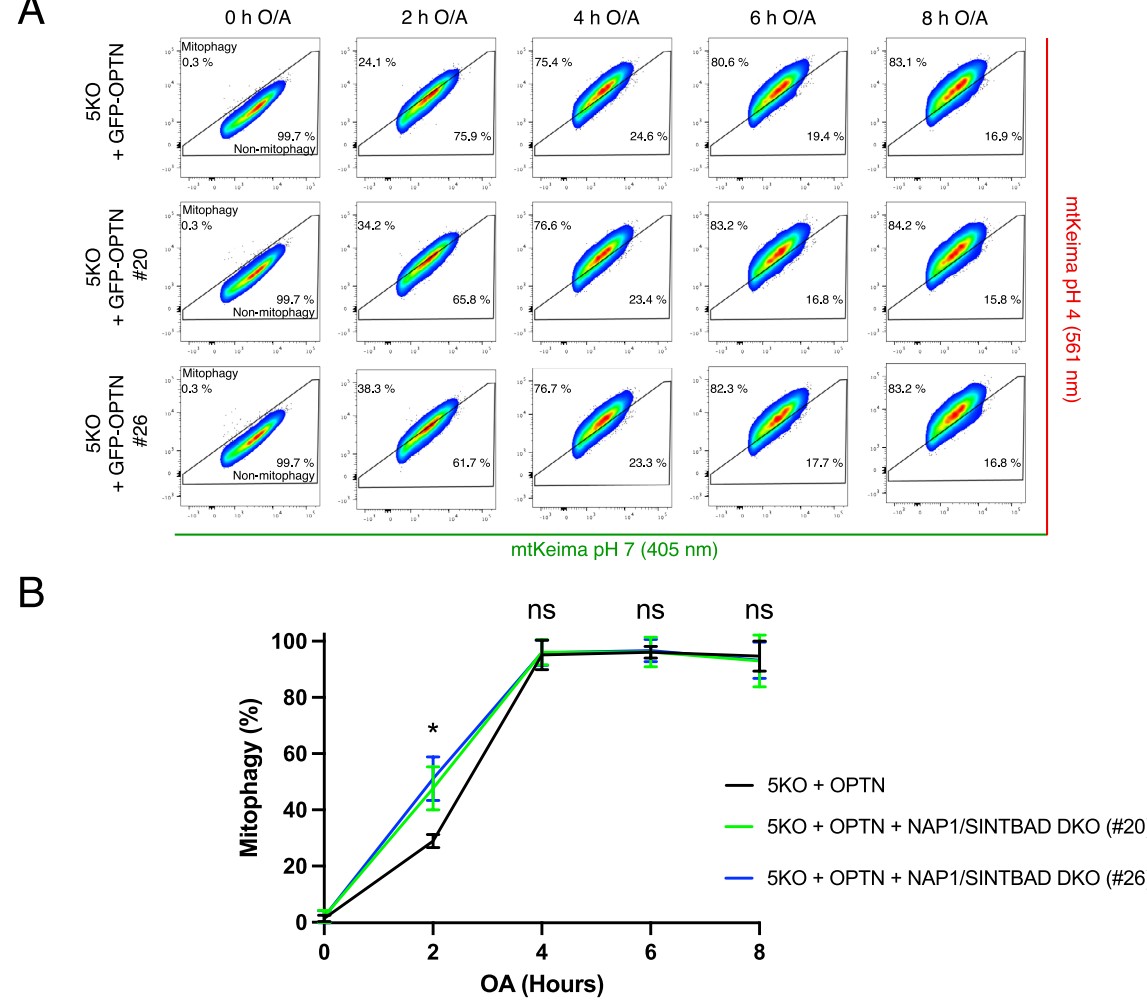

**Extended Data Fig. 6 | Mt-mKeima mitophagy flux assay in pentaKO cells rescued with OPTN and in presence or absence of NAP1/SINTBAD. (a-b)** pentaKO (5KO) and NAP1/SINTBAD DKO/5KO (clones #20 and #26)) expressing BFP-Parkin, GFP-OPTN and mt-mKeima were left untreated and treated with O/A for indicated time points and mitophagy flux was measured by flow cytometry (A) and quantified (B) (mean ± s.d) (*n* = 3 biologically independent experiments). Two-way ANOVA with Tukey's multiple comparisons test was performed. *$P$ < 0.05. ns, not significant. Source numerical data, including exact $P$ values, are available in source data.

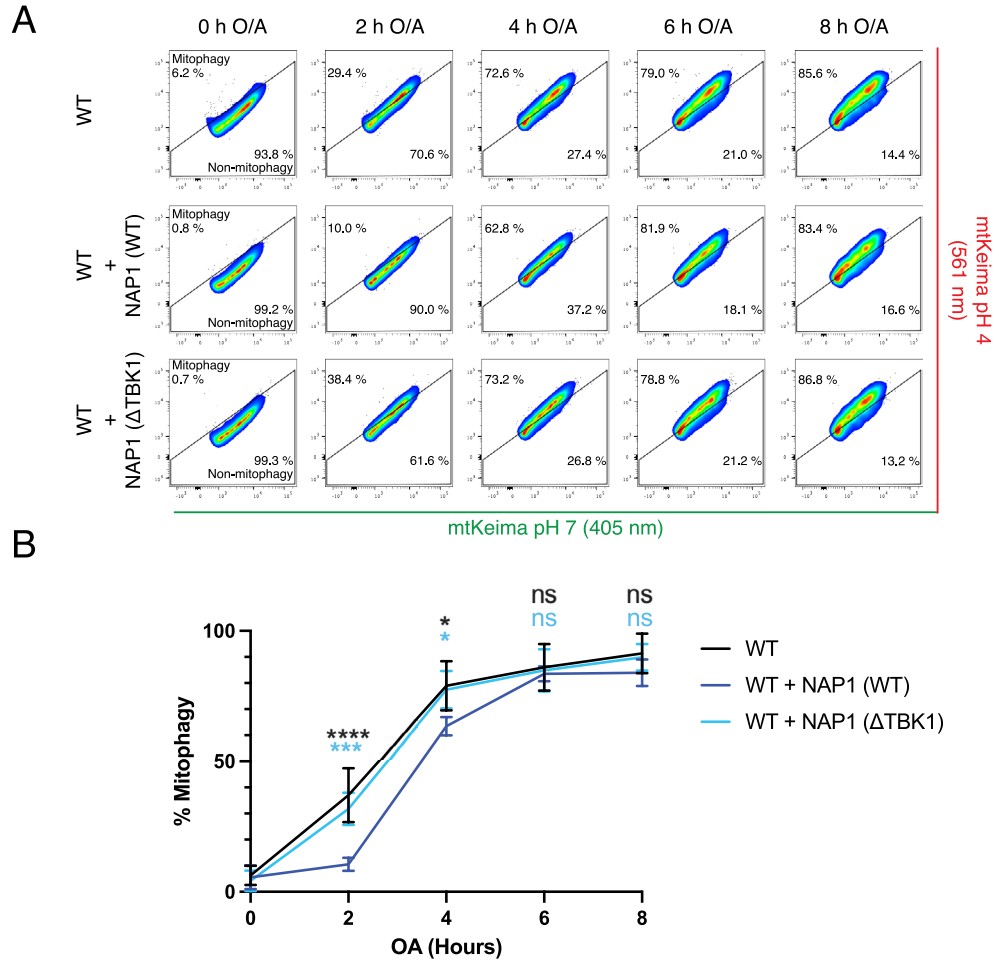

**Extended Data Fig. 7 | Mitophagy flux assay for wild-type and TBK1-binding deficient NAP1. (a-b)** Wild-type HeLa cells expressing BFP-Parkin and mt-mKeima with and without overexpression of NAP1 wild-type (WT) or TBK1-binding deficient mutant (L226Q/L233Q) (ΔTBK1), were left untreated or treated with O/A for indicated time points and mitophagy flux was measured by flow cytometry (A) and quantified (B) (mean ± s.d) (n = 3 biologically independent experiments). Two-way ANOVA with Tukey's multiple comparisons test was performed. *P < 0.05, **P < 0.005, ***P < 0.001, ****P < 0.0001. ns, not significant. Source numerical data, including exact P values, are available in source data.

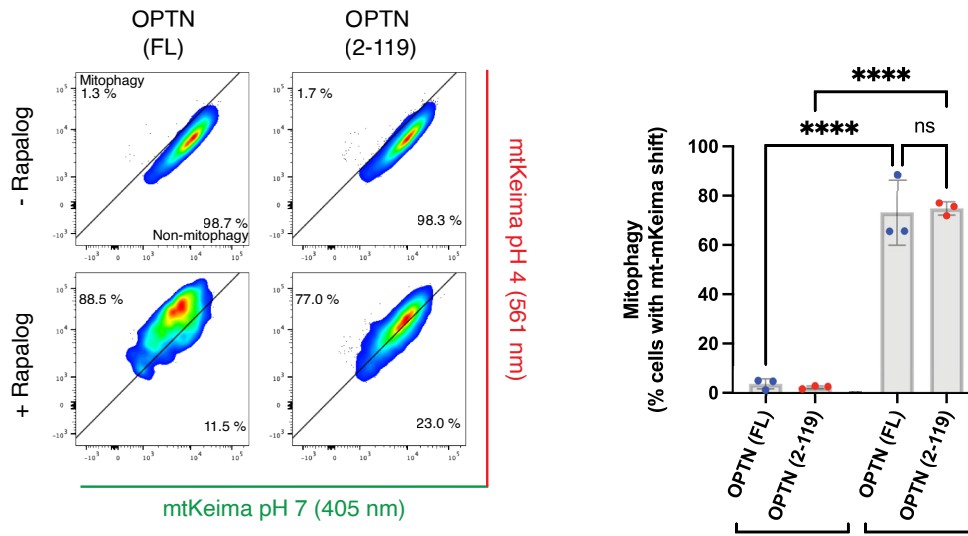

**Extended Data Fig. 8 | Chemical-dimerization assay with full-length OPTN and OPTN (2-119).** PentaKO HeLa cells expressing mt-mKeima, Fis1-FRB and FKBP-OPTN wild-type (WT) or amino acids 2-119 (2-119) were left untreated or treated with rapalog for 24 hours as indicated. The mitophagy flux was analyzed by flow cytometry. Representative FACS plots are shown. The percentage of mitophagy-induced cells (upper left) is quantified (mean ± s.d) (*n* = 3 biologically independent experiments). Two-way ANOVA with Tukey's multiple comparisons test. ****P < 0.0001. Source numerical data, including exact *P* values, are available in source data.

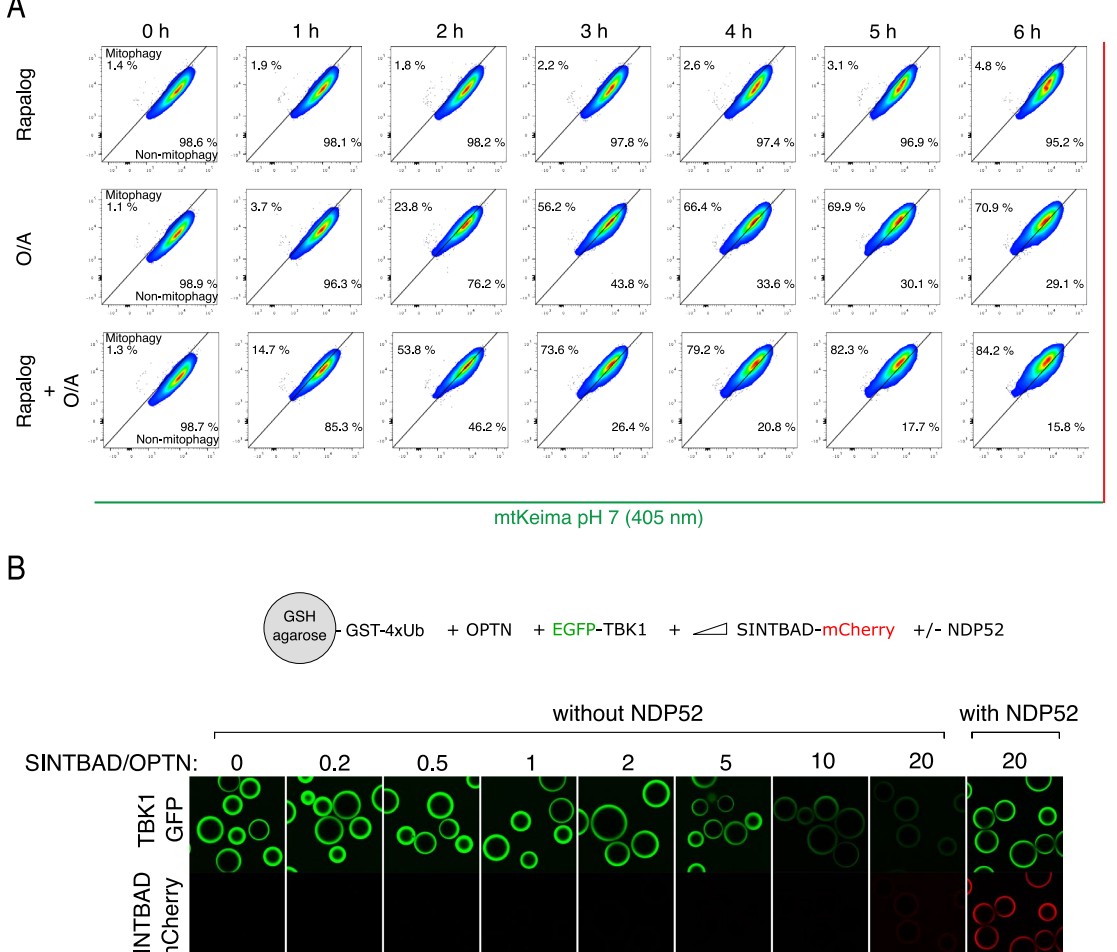

**Extended Data Fig. 9 | Chemical-dimerization assay with OPTN (2-119) in pentaKO HeLa cells rescued with GFP-NDP52 and biochemical reconstitution of the TBK1 recruitment by OPTN in presence of SINTBAD and/or NDP52.**
(**a**) PentaKO HeLa cells expressing BFP-Parkin, GFP-NDP52, and mt-mKeima were further transduced with Fis1-FRB and FKBP-OPTN amino acids 2-119 (2-119). Cells were either left untreated (time point 0 h) or treated with rapalog alone, O/A alone, or rapalog plus O/A for the indicated times. The mt-mKeima signal was measured by flow cytometry ($n$ = 3 biologically independent experiments). (**b**) Biochemical reconstitution of the recruitment of GFP-TBK1 by OPTN to GST-4xUb coated beads in the presence or absence of increasing amounts of SINTBAD-mCherry. In the indicated wells, unlabeled NDP52 was also added. Diagram of the experimental set-up (top) and experimental results obtained by confocal imaging (bottom). One representative experiment is shown ($n$ = 3 biologically independent experiments). Scale bar is 100 μm.

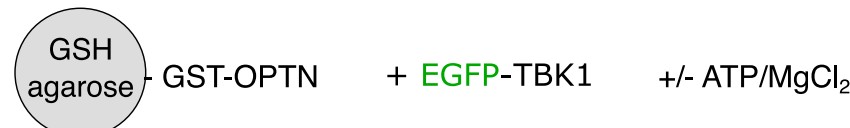

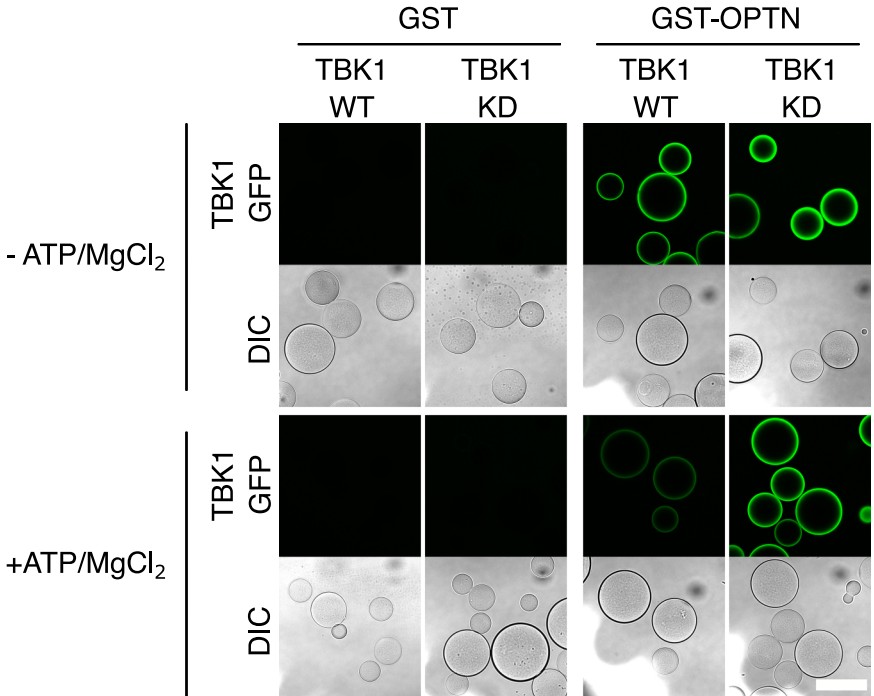

**Extended Data Fig. 10 | Microscopy-based bead assay of the OPTN-TBK1 interaction in presence and absence of ATP.** Reconstitution of the OPTN-TBK1 interaction in presence and absence of ATP/MgCl2 by tethering OPTN-GST to GSH-agarose beads and incubating it with GFP-TBK1. Confocal images are shown representing the binding of wild-type (WT) and kinase dead (KD) GFP-TBK1 to OPTN. One representative experiment is shown ($n$ = 3 biologically independent experiments). Scale bar is 100 μm.

Michael Lazarou
Sascha Martens

# Reporting Summary

## Statistics

For all statistical analyses, confirm that the following items are present in the figure legend, table legend, main text, or Methods section.

| n/a | Confirmed | |
|---|---|---|
| ☐ | ☒ | The exact sample size (*n*) for each experimental group/condition, given as a discrete number and unit of measurement |
| ☐ | ☒ | A statement on whether measurements were taken from distinct samples or whether the same sample was measured repeatedly |
| ☐ | ☒ | The statistical test(s) used AND whether they are one- or two-sided *Only common tests should be described solely by name; describe more complex techniques in the Methods section.* |
| ☒ | ☐ | A description of all covariates tested |
| ☐ | ☒ | A description of any assumptions or corrections, such as tests of normality and adjustment for multiple comparisons |
| ☐ | ☒ | A full description of the statistical parameters including central tendency (e.g. means) or other basic estimates (e.g. regression coefficient) AND variation (e.g. standard deviation) or associated estimates of uncertainty (e.g. confidence intervals) |
| ☐ | ☒ | For null hypothesis testing, the test statistic (e.g. *F*, *t*, *r*) with confidence intervals, effect sizes, degrees of freedom and *P* value noted *Give P values as exact values whenever suitable.* |
| ☒ | ☐ | For Bayesian analysis, information on the choice of priors and Markov chain Monte Carlo settings |
| ☒ | ☐ | For hierarchical and complex designs, identification of the appropriate level for tests and full reporting of outcomes |
| ☒ | ☐ | Estimates of effect sizes (e.g. Cohen's *d*, Pearson's *r*), indicating how they were calculated |

*Our web collection on statistics for biologists contains articles on many of the points above.*

## Software and code

Policy information about availability of computer code

| Data collection | Confocal microscopy images were collected using ZEN software version 2022 (Carl Zeiss Microscopy, GmbH, Germany) connected to a LSM700 or an inverted Leica SP8 confocal laser scanning microscope and the Leica Application Suite X (LASX v2.0.1). |
|---|---|
| Data analysis | 1. FlowJo software (Version 10.9.0; Tree Star Inc., Ashland, OR, USA) for FACS-data analysis (RRID:SCR_008520).<br>2. PRISM software (Version 9.5.1; Graphpad Software, La Jolla, CA, USA) for statistical analysis and generating graphs.<br>3. ImageJ software (Schindelin et al. 2015) (version 2.14.0/1.54f) for immunofluorescence microscopy image analysis (RRID:SCR_003070).<br>4. FACSDiva software (BD FACSDiva software) for flow cytometry experiments (RRID:SCR_001456).<br>5. Synthego ICE v2 CRISPR Analysis tool (Synthego) for CRISPR knockout experiments.<br>6. FreeStyle 1.7 software (Thermo Scientific) for mass spectrometry data.<br>7. MaxQuant software version 1.6.17.0 for mass spectrometry data.<br>8.  Leica Application Suite X (LASX v2.0.1)  (RRID:SCR_013673) |

For manuscripts utilizing custom algorithms or software that are central to the research but not yet described in published literature, software must be made available to editors and reviewers. We strongly encourage code deposition in a community repository (e.g. GitHub). See the Nature Portfolio guidelines for submitting code & software for further information.

## Data

Policy information about availability of data

All manuscripts must include a data availability statement. This statement should provide the following information, where applicable:

- Accession codes, unique identifiers, or web links for publicly available datasets
- A description of any restrictions on data availability
- For clinical datasets or third party data, please ensure that the statement adheres to our policy

> Raw files associated with this work are available on Zenodo (https://doi.org/10.5281/zenodo.10637353). The mass spectrometry proteomics data have been deposited to the ProteomeXchange Consortium via the PRIDE partner repository [88] with the dataset identifier PXD049184. For screening CRISPR knockout clones we used Human Genome Variation Society (HGVS; http://varnomen.hgvs.org/). Mass spectrometry dataset was analyzed using Uniprot human proteome database (release 2021_03) and a database of common laboratory contaminants.

## Research involving human participants, their data, or biological material

Policy information about studies with human participants or human data. See also policy information about sex, gender (identity/presentation), and sexual orientation and race, ethnicity and racism.

| | |
|---|---|
| Reporting on sex and gender | N/A |
| Reporting on race, ethnicity, or other socially relevant groupings | N/A |
| Population characteristics | N/A |
| Recruitment | N/A |
| Ethics oversight | N/A |

Note that full information on the approval of the study protocol must also be provided in the manuscript.

# Field-specific reporting

Please select the one below that is the best fit for your research. If you are not sure, read the appropriate sections before making your selection.

☒ Life sciences ☐ Behavioural & social sciences ☐ Ecological, evolutionary & environmental sciences

For a reference copy of the document with all sections, see nature.com/documents/nr-reporting-summary-flat.pdf

# Life sciences study design

All studies must disclose on these points even when the disclosure is negative.

| | |
|---|---|
| Sample size | No statistical methods were applied to pre-evaluate sample size because of the exploratory nature of our study which entails that effect sizes were not known prior to the study. Experiments were performed at least as three replicates, according to current practices in the field. Statistical analysis was performed on experiments for which the sample size included at least 3 biological replicates. Sample sizes were based on previous experience and current standards in the field. |
| Data exclusions | No data were excluded from the analyses. |
| Replication | All experiments were replicated at least three times with similar findings. Samples sizes are provided in the figure legends. |
| Randomization | Samples were allocated into experimental groups by genotype of knockout condition. Covariates were controlled for by maintaining all samples in the same growth and media conditions. |
| Blinding | The investigators were not blinded to treatment or genotype allocations during this study. For cell based experiments, it was not possible to blind the experimenter as researchers needed to know experimental conditions for performing experiments. |

# Reporting for specific materials, systems and methods

We require information from authors about some types of materials, experimental systems and methods used in many studies. Here, indicate whether each material, system or method listed is relevant to your study. If you are not sure if a list item applies to your research, read the appropriate section before selecting a response.

## Materials & experimental systems

| n/a | Involved in the study |
|-----|----------------------|
| ☐ | ☒ Antibodies |
| ☐ | ☒ Eukaryotic cell lines |
| ☒ | ☐ Palaeontology and archaeology |
| ☒ | ☐ Animals and other organisms |
| ☒ | ☐ Clinical data |
| ☒ | ☐ Dual use research of concern |
| ☒ | ☐ Plants |

## Methods

| n/a | Involved in the study |
|-----|----------------------|
| ☒ | ☐ ChIP-seq |
| ☐ | ☒ Flow cytometry |
| ☒ | ☐ MRI-based neuroimaging |

## Antibodies

**Antibodies used**

The primary antibodies used in this study for western blotting are:
anti-β-Actin (1:5000, Abcam Cat# ab20272, RRID:AB_445482)
anti- COXII (1:1000, Abcam Cat# ab110258, RRID:AB_10887758)
anti-COXII (1:1000, Cell Signaling Technology Cat# 31219, RRID:AB_2936222)
anti-FIP200 (1:1000, Cell Signaling Technology Cat# 12436, RRID:AB_2797913)
anti-HA (1:500, Cell Signaling Technology Cat# 2367, RRID:AB_10691311)
anti-HA (1:1000, Cell Signaling Technology Cat# 3724, RRID:AB_1549585)
anti-mHSP60 (1:1000, Abcam Cat# ab128567, RRID:AB_11145464)
anti-NAP1 (1:1000, Abcam Cat# ab192253, RRID:AB_2941051)
anti-OPTN (1:500, Sigma Aldrich Cat# HPA003279, RRID:AB_1079527)
anti-p62/SQSTM1 (1:1000, Abnova Cat# H00008878-M01, RRID:AB_437085)
anti-SINTBAD (1:1000, Cell Signaling Technology Cat# 8605, RRID:AB_10839270)
anti-TBK1 (1:1000, Cell Signaling Technology Cat# 38066, RRID:AB_2827657)
anti-TBK1 (1:1000, Cell Signaling Technology Cat# 3013, RRID:AB_2199749)
anti-phospho-TBK1 S172 (1:1000, Cell Signaling Technology Cat# 5483, RRID:AB_10693472)
anti-α-Tubulin (1:5000, Abcam Cat# ab7291, RRID:AB_2241126)
anti-PARKIN (1:200 Santa Cruz Biotechnology Cat# sc-32282, RRID:AB_628104)
anti-phospho-Ubiquitin S65 (1:2000, Millipore Cat# ABS1513-I, RRID:AB_2858191)
anti-ULK1 (1:1000, Cell Signaling Technology Cat# 8054, RRID:AB_11178668)

The secondary antibodies for western blotting used in this study are:
HRP conjugated polyclonal goat anti-mouse (Jackson ImmunoResearch Labs Cat# 115-035-003, RRID:AB_10015289)
HRP conjugated polyclonal goat anti-rabbit (Jackson ImmunoResearch Labs Cat# 111-035-003, RRID:AB_2313567)

The primary antibodies used in this study for immunofluorescence are:
anti-HA (1:1000, Cell Signaling Technology Cat# 3724, RRID:AB_1549585)
anti-mHSP60 (1:1000, Abcam Cat# ab128567, RRID:AB_11145464)
anti-mHSP60 (1:500, Novus Biologicals Cat#NBP3-05536, RRID:AB_3086708)
anti-GFP (1:250-1:500, Thermo Fisher, Cat# A10262; RRID:AB_2534023)
anti-WIPI2 (1:500, Abcam, Cat# ab105459; RRID: AB_10860881)
anti-B17.2L (a kind gift from Prof. Mike Ryan (Monash University) and previously generated in rabbits using recombinant full length B17.2L antigen)
anti-TBK1 (1:200, Cell Signaling Technology Cat# 38066, RRID:AB_2827657)
anti-ATG13 (1:200, Cell Signaling Technology, Cat# 13468; RRID: AB_2797419)

The secondary antibodies for immunofluorescence used in this study are:
AlexaFluor-488 goat anti-rabbit IgG (H+L) (1:250-500, Thermo Fisher, Cat# A11008; RRID: AB_143165)
AlexaFluor-555 goat anti-rabbit IgG (H+L) (1:250-500, Thermo Fisher, Cat# A21428; RRID: AB_2535849)
AlexaFluor-647 goat anti-rabbit IgG (H+L) (1:250-500, Thermo Fisher, Cat# A21244; RRID: AB_2535812)
AlexaFluor-488 goat anti-mouse IgG (H+L) (1:250-500, Thermo Fisher, Cat# A21202; RRID: AB_141607)
AlexaFluor-555 goat anti-mouse IgG (H+L) (1:250-500, Thermo Fisher, Cat# A21422; RRID: AB_2535844)
AlexaFluor-647 goat anti-mouse IgG (H+L) (1:250-500, Thermo Fisher, Cat# A21235; RRID: AB_2535804)
AlexaFluor-488 goat anti-chicken IgY (H+L) (1:250-1:500, Thermo Fisher, Cat# A32931, RRID: AB_2762843)
AlexaFluor-647 goat anti-chicken IgY (H+L) (1:250-1:500, Thermo Fisher, Cat# A21449; RRID: AB_2535866)

**Validation**

Antibodies were selected based on their use in other publications and/or validation by the manufacturers for their respective application. Where possible, knockout cell lines were used to validate the specificity of the antibodies further. These were FIP200, NAP1, OPTN, p62,/SQSTM1, SINTBAD, TBK1, p-TBK1, Parkin, WIPI2, ATG13, and ULK1. Alternatively, some antibodies were validated by verifying their known localization when we separated cytosol from mitochondria in cell fractionation experiments, allowing us to verify the specificity of antibodies based on the cellular localization of the target protein, such as our loading controls mHSP60, B-Actin, a-Tubulin, COXII. Antibodies against affinity tags were validated by comparing overexpression versus non-transfected cell lines (anti-HA). The phospho-Ub antibody was validated by its known upregulation upon O/A treatment of cells and this was confirmed to lead to an increased signal, in line with what is found in literature (PMID: 32142685). The anti-B17.2L was previously published and validated (PMID: 30679426). Specificity of secondary antibodies was validated by omitting primary antibodies.

# Eukaryotic cell lines

Policy information about <u>cell lines and Sex and Gender in Research</u>

| | |
|---|---|
| Cell line source(s) | All parental cell lines (HeLa, HEK293T, HEK293F) were acquired from the American Type Culture Collection (ATCC). HAP1 cells (RRID:CVCL_Y019) were acquired from Horizon Discovery. HAP1 knockout lines ATG5 (RRID:CVCL_SE00) and FIP200/RB1CC1 (RRID:CVCL_TI59) were acquired from Horizon Discovery. HeLa knockout cell lines were generated during this study and submitted to Cellosaurus. Sf9 insect cells were acquired from Thermo Fisher (12659017, RRID:CVCL_0549). |
| Authentication | Authentication was performed upon first arrival in the lab based on morphology and karyotyping. |
| Mycoplasma contamination | All cell lines were routinely tested for mycoplasma contamination. All cell lines were negative throughout the study. |
| Commonly misidentified lines (See <u>ICLAC</u> register) | The cell lines used in this study are not listed as commonly misidentified cell lines. This was verified in the ICLAC table of commonly misidentified cell lines. |

# Flow Cytometry

## Plots

Confirm that:

☒ The axis labels state the marker and fluorochrome used (e.g. CD4-FITC).

☒ The axis scales are clearly visible. Include numbers along axes only for bottom left plot of group (a 'group' is an analysis of identical markers).

☒ All plots are contour plots with outliers or pseudocolor plots.

☒ A numerical value for number of cells or percentage (with statistics) is provided.

## Methodology

| | |
|---|---|
| Sample preparation | HeLa cells were transduced with lentiviral or retroviral vectors that would express the fluorophore. Cells were treated according the experimental protocol and then collected by removing the medium, washing the cells with 1x PBS (14190169, Thermo Fisher), trypsinization (T3924, Sigma), and resuspending in complete DMEM medium (41966052, Thermo Fisher). Filtered through 35 µm cell-strainer caps (352235, Falcon) and analyzed by an LSR Fortessa Cell Analyzer (BD Biosciences). Lysosomal mt-mKeima was measured using dual excitation ratiometric pH measurements at 405 (pH 7) and 561 (pH 4) nm lasers with 710/50-nm and 610/20-nm detection filters, respectively. Additional channels used for fluorescence compensation were BFP and GFP. Single fluorescence vector expressing cells were prepared to adjust photomultiplier tube voltages to make sure the signal was within detection limits, and to calculate the compensation matrix in BD FACSDiva Software. Depending on the experiment, we gated for BFP-positive, GFP-positive, and mKeima-positive cells with the appropriate compensation. For each sample, 10,000 mKeima-positive events were collected, and data were analyzed in FlowJo (version 10.9.0). |
| Instrument | LSR Fortessa Cell Analyzer (BD Biosciences) |
| Software | BD FACSDiva software during data collection and FlowJo10 software (Tree Star Inc., Ashland, OR, USA) for data analysis. |
| Cell population abundance | Cells were only included when they were viable, single cells (exclusion doublets), and depending on the experiment whether they were GFP-, BFP-, and mt-mKeima positive. |
| Gating strategy | Gating was optimized, depending on the experiment, for GFP- and/or BFP- and mt-mKeima positive cells after viable singlets were separated from potentially dead cells or doublets based on scatter. |

☒ Tick this box to confirm that a figure exemplifying the gating strategy is provided in the Supplementary Information.

