## [Peer Review File · Nature Structural & Molecular Biology]

Peer Review Information

Manuscript Title: Control of mitophagy initiation and progression by the TBK1 adaptors NAP1 and SINTBAD

Corresponding author name(s): Sascha Martens, Michael Lazarou, Elias Adriaenssens

Reviewer Comments & Decisions:

Decision Letter, initial version:

Message: 26th Oct 2023

Dear Professor Martens,

Thank you again for submitting your manuscript "Control of mitophagy initiation and progression by the TBK1 adaptors NAP1 and SINTBAD". We now have comments (below) from the 2 reviewers who evaluated your paper. In light of these reports, we remain interested in your study and would like to see your response to the comments of the referees, in the form of a revised manuscript.

You will see that though the experts appreciate the potential importance and novelty of the findings, they raise important concerns both at a technical and conceptual level. More specifically, the experts note several technical issues (missing controls, lack of relevant quantifications, statistical analyses, and experimental clarity in certain areas) that must be addressed in their entirety. As importantly, both reviewers pose functional and mechanistic questions and request relevant experiments, as well as secondary supporting experiments to exclude alternative mechanistic explanations of the current conclusions, to boost the conceptual advance imparted by this study. We note that we would be reluctant to consult the referees again in the absence of major revisions that address several of these questions, as we think that addressing them would significantly elevate the value of the manuscript. Finally, as always, we ask you to please be sure to address/respond to all concerns of the referees in full in a point-by-point response and highlight all changes in the revised manuscript text file.

We appreciate the requested revisions are extensive. We thus expect to see your revised manuscript within 6 months. If you cannot send it within this time, please let us know. We will be happy to consider your revision as long as nothing similar has been accepted for publication at NSMB or published elsewhere. Should your manuscript be substantially delayed without notifying us in advance and your article is eventually published, the received date would be that of the revised, not the original, version.

Reporting Summary:

When submitting the revised version of your manuscript, please pay close attention to our [href="https://www.nature.com/nature-portfolio/editorial-policies/image-integrity">Digital Image Integrity Guidelines](https://www.nature.com/nature-portfolio/editorial-policies/image-integrity). and to the following points below:

We require deposition of coordinates (and, in the case of crystal structures, structure factors) into the Protein Data Bank with the designation of immediate release upon publication (HPUB). Electron microscopy-derived density maps and coordinate data must

be deposited in EMDB and released upon publication. Deposition and immediate release of NMR chemical shift assignments are highly encouraged. Deposition of deep sequencing and microarray data is mandatory, and the datasets must be released prior to or upon publication. To avoid delays in publication, dataset accession numbers must be supplied with the final accepted manuscript and appropriate release dates must be indicated at the galley proof stage. Please find the complete NRG policies on data availability at <http://www.nature.com/authors/policies/availability.html>.

Nature Structural & Molecular Biology is committed to improving transparency in authorship. As part of our efforts in this direction, we are now requesting that all authors identified as 'corresponding author' on published papers create and link their Open Researcher and Contributor Identifier (ORCID) with their account on the Manuscript Tracking System (MTS), prior to acceptance. This applies to primary research papers only. ORCID helps the scientific community achieve unambiguous attribution of all scholarly contributions. You can create and link your ORCID from the home page of the MTS by clicking on 'Modify my Springer Nature account'. For more information please visit please visit www.springernature.com/orcid.

[Redacted]

Sincerely,

Dimitris Typas
Associate Editor
Nature Structural & Molecular Biology
ORCID: 0000-0002-8737-1319

Reviewers' Comments:

Reviewer #1:

Remarks to the Author:

Felix Randow's group showed that NDP52 recognizes ubiquitin-coated Salmonella enterica in human cells and, by binding the adaptor proteins NAP1 and SINTBAD, recruits TBK1 (Nat Immunol 2009. PMID: 19820708) and further that NDP52 forms a trimeric complex with FIP200 and SINTBAD/NAP1, which are subunits of the autophagy-initiating ULK and the TBK1 kinase complex, respectively (Mol Cell 2019. PMID: 30853402). A previous study by the authors also showed by recombinant proteins combined with a fluorescence microscopy-based bead assay, NDP52 interacts with TBK1 via binding partners NAP1 or SINTBAD (Mol Cell 2023. PMID: 37207627). The crystal structures of optineurin/TBK1 complex and the related NAP1/TBK1 complex revealed that OPTN and NAP1 are mutually

exclusive in binding to TBK1, and likely to form distinct TBK1/adaptor complexes for different intracellular signalling pathways (Nat Commun 2016 PMID: 27620379). Koji Yamano et al., found that the accumulation of OPTN at a contact site between the autophagy machinery and ubiquitin-coated mitochondria provides a platform for TBK1 hetero-autophosphorylation by adjacent TBK1 at the contact site (bioRxiv 2023 doi: <https://doi.org/10.1101/2023.02.24.529790>). Considering those previous reports, this study lacks conceptual advance. The issues that should be clarified in this study would be: 1. how is OPTN-mediated mitophagy initiated by O/A treatment (i.e., how are NAP and SINDBAD removed from OPTN by O/A treatment)?, 2. How is NDP52-mediated mitophagy induced following the OPTN-mediated mitophagy? (i.e., how do NAP and SINDBAD become high affinity to NDP52?). In addition, structural based data might become explainable for the author's model (transition of NAP1/SINDBAD from OPTN to NDP52.). This study does not reach a criteria to publish in Nature Structure & Molecular Biology, at least current version.

Other comments

1. In Figure 1C, the exposure of HeLa cells with O/A decreased the levels of NAP and SINDBAD, which were partial recovered by the treatment of BafA1. There is a possibility that the exposure of HeLa cells with O/A decreased gene expression of NAP and SINDBAD. Alternatively, some population of NAP and SINDBAD may be released from damaged mitochondria. The authors should investigate such possibilities.
2. In Figure 3B, difference between Penta KO (parental control, CTL) and NAP1/SINDBAD DKO/penta KO (clones 20 and 26) expressing BFP-Parkin and GFP-NDP52 on mitophagy activity was subtle, and this reviewer is not sure whether NAP1 and SINDBAD are involved in NDP52-mediated mitophagy.
3. In Figure 3D, the meaning of the difference in salt concentration is not explained.
4. In Figure 3F-G, the number of mitochondrial WIPI2 and ATG13 puncta per cell should be quantified.
5. Figure 4C, it is unclear which cells (wild-type (WT) or penta KO (5KO) HeLa cells expressing BFP-Parkin and mt-mKeima) are used.
6. Figure 4D, the domain structures of NAP1 and which region is the NDP52, FIP200, TBK1 binding region should be indicated. Also, binding defect of each deletion mutant should be shown in experiments.
7. In Figure 5, ITC analysis is available for determination of each Kd value.
8. Statistic analyses are required for Figure S3, S4 and S6.
9. In addition to immunoblot analysis shown in Figure 5F, the authors should perform mtKeima assay.
10. Figure 6D, quantification is required.

Reviewer #2:

Remarks to the Author:

In this manuscript, the authors studied the role of the TBK1 adaptors, NAP1 and SINDBAD, in mitophagy. They first found that NAP1/SINDBAD has an inhibitory role in PINK1/Parkin-dependent mitophagy. Interestingly, in NDP52-mediated mitophagy, NAP1/SINDBAD acts as a cargo co-receptor by recruiting TBK1 to NDP52, stabilizing its interaction with FIP200. Conversely, NAP1/SINDBAD inhibits OPTN-mediated mitophagy by competing with OPTN for TBK1 binding. Finally, the authors analyzed the crosstalk of OPTN-mediated and NDP52-mediated mitophagy regulation by NAP1/SINDBAD and found that NAP1/SINDBAD acts as cargo receptor rheostats and elevates the threshold for mitophagy initiation by

OPTN while promoting the progression of the pathway once set in motion by supporting NDP52.

Overall, the experiments are well designed and the data support most of the conclusions and are potentially important. However, there are still several major concerns that need to be addressed by the authors.

1. In many experiments other than in vitro assays, the expression levels of NDP52 and OPTN have not been examined. Their expression levels may affect TBK1 activity and mitophagy. It is particularly important to show NDP52 and OPTN expressed against PentaKO cells in comparison to endogenous expression levels in control HeLa cells.

2. In this manuscript, NAP1 and SINTBAD are analyzed as having the same function because of their structural similarity, and most experiments use only NAP1. However, it is possible that NAP1 and SINTBAD have different functions, but this is rarely mentioned; SINTBAD should be used in at least some key experiments to see if SINTBAD yields the same results as NAP1.

3. In Figure 6D, the authors conclude that OPTN is able to accumulate more TBK1 on the mitochondrial surface than NDP52. However, it seems possible that NDP52 is less accumulated on mitochondria or less expressed than OPTN and therefore can bind less TBK1. A more quantitative interpretation of Figure 6D is needed to show whether OPTN or NDP52 accumulates more TBK1.

4. The crosstalk between OPTN-axis and NDP52-axis is not fully elucidated.

4.1. In Figure 7C, OA+Rapalog induces mitophagy slightly more strongly than OA alone at 2 hours mitophagy induction, but no difference is observed at other induction times. This result alone is not sufficient to conclude that the recruitment of TBK1 by OPTN enhances mitophagy via NDP52.

4.2. The recovery of TBK1 on the beads by adding DNP52 in Figure 7D may simply be a result of NDP52 on the beads accumulating TBK1 via NAP1. TBK1 accumulation on beads with and without OPTN is needed to be compared to confirm that NDP52 supports OPTN-driven mitophagy.

4.3. If the recruitment of TBK1 by NDP52 supports OPTN-driven mitophagy, the binding between TBK1 and OPTN should be stronger in the experiment in lane 10 of Figure 7E with NDP52 than without NDP52, so this need to be confirmed.

5. Figure 5A, 6E, and 7D show the diagram of the experiment, but Figure 3D does not, making it difficult to understand.

6. Figure 8, panel 4: Is the pale orange circle with LC3 bound a phagophore or an autophagosome? If it is a phagophore, it should be flat or cup-shaped, not circular, and should be a single membrane; if it is an autophagosome, it should be described as enclosing the mitochondria.

Author Rebuttal to Initial comments

Reviewers' Comments:

Reviewer #1:

Remarks to the Author:

Felix Randow's group showed that NDP52 recognizes ubiquitin-coated *Salmonella enterica* in human cells and, by binding the adaptor proteins NAP1 and SINTBAD, recruits TBK1 (Nat Immunol 2009. PMID: 19820708) and further that NDP52 forms a trimeric complex with FIP200 and SINTBAD/NAP1, which are subunits of the autophagy-initiating ULK and the TBK1 kinase complex, respectively (Mol Cell 2019. PMID: 30853402). A previous study by the authors also showed by recombinant proteins combined with a fluorescence microscopy-based bead assay, NDP52 interacts with TBK1 via binding partners NAP1 or SINTBAD (Mol Cell 2023. PMID: 37207627). The crystal structures of optineurin/TBK1 complex and the related NAP1/TBK1 complex revealed that OPTN and NAP1 are mutually exclusive in binding to TBK1, and likely to form distinct TBK1/adaptor complexes for different intracellular signalling pathways (Nat Commun 2016 PMID: 27620379). Koji Yamano et al., found that the accumulation of OPTN at a contact site between the autophagy machinery and ubiquitin-coated mitochondria provides a platform for TBK1 hetero-autophosphorylation by adjacent TBK1 at the contact site (bioRxiv 2023 doi: <https://doi.org/10.1101/2023.02.24.529790>). Considering those previous reports, this study lacks conceptual advance.

We would like to thank the reviewer for taking the time to carefully read our manuscript and are grateful for the comments described below. We have addressed the comments with additional experiments and feel that this has strengthened the manuscript considerably.

The issues that should be clarified in this study would be:

1. how is OPTN-mediated mitophagy initiated by O/A treatment (i.e., how are NAP and SINTBAD removed from OPTN by O/A treatment)?

We thank the reviewer for this question and apologize in case this was not clear from our manuscript. However, NAP1/SINTBAD do not directly bind to OPTN (see data below). We observed an interaction of NAP1 with NDP52, and also with TAX1BP1, but none of the other soluble cargo receptors. This is consistent with the fact that NAP1 was shown to interact with the SKICH domain of NDP52 and TAX1BP1 (Ravenhill et al. 2019 Mol Cell; Fu et al. 2021, Sci Adv), which is missing in other cargo receptors. This may help to resolve this question from the reviewer, as NAP1/SINTBAD do not have to be removed from OPTN. Instead, they perform their inhibitory role from the cytoplasm by limiting the amount of TBK1 that can be recruited from the cytosol to the mitochondrial surface by OPTN.

In addition, it is also important to think about this indirect inhibitory mechanism by means of local concentration. This implies that while NAP1 and SINTBAD may have a higher binding affinity for TBK1 compared to OPTN (as was elegantly shown by the Lifeng Pan lab in Li et al. 2016 Nat Comms), the recruitment of OPTN to the mitochondrial surface results in a locally increased OPTN concentration (Yamano et al. 2024, EMBO J) to outcompete NAP1/SINTBAD from TBK1. This mechanism also helps to explain how OPTN is prevented from interacting with TBK1 in the absence of mitochondrial damage, as OPTN is then dispersed in the cytosol and unable to cluster TBK1, which is required for its activation by trans-autophosphorylation.

We hope this addresses the reviewer's question regarding how OPTN-mediated mitophagy is initiated and how (in the absence of NDP52) OPTN can outcompete NAP1 and SINTBAD from binding TBK1. We have also scanned our manuscript carefully to ascertain that no statements suggestive of a direct interaction between OPTN and NAP1/SINTBAD are included. In addition, we have adjusted the text underneath the schematic in Figure 8 to make this point clearer and avoid any possible confusion. Furthermore, we have added a sentence to the introduction in which we clearly state that NAP1/SINTBAD bind to SKICH-domain containing cargo receptors.

2. How is NDP52-mediated mitophagy induced following the OPTN-mediated mitophagy? (i.e., how do NAP and SINDBAD become high affinity to NDP52?).

We thank the reviewer for this comment and agree this transition from inhibitory to stimulatory may be subject to another layer of regulation. To investigate the possible existence of a conversion mechanism, we hypothesized that NAP1 itself could be subject to TBK1-mediated phosphorylation. Since cargo receptors are phosphorylated by TBK1 to enhance their ubiquitin- and LC3-binding capabilities, we wondered whether the cargo co-receptors NAP1 and SINTBAD would be subjected to a similar activation mechanism.

We therefore started by testing whether TBK1 can phosphorylate recombinant NAP1 *in vitro*. To this end, we performed an *in vitro* kinase assay and submitted the samples for mass spectrometry analysis. This revealed several NAP1 residues that were phosphorylated by TBK1 (Revision Figure 2.1).

Revision Figure 2.1

We then mutated all six residues to alanine and used an *in vitro* kinase assay to determine whether TBK1 could still phosphorylate the mutant NAP1, by assessing NAP1's smearing pattern on SDS-PAGE. This revealed that while wild-type NAP1 would rapidly smear upon incubation with TBK1, this smearing was nearly completely lost in the 6xAla mutant, indicating these six residues form the primary TBK1 targets (Revision Figure 2.2).

Revision Figure 2.2

We then tested whether NAP1 is also phosphorylated by TBK1 in cells. To this end, we first used a FIP200 knockout HAP1 cell line as a tool. FIP200 knockout cells were previously shown to have hyperactivated TBK1, presumably due to accumulation of autophagic cargo material that cannot be degraded in absence of the key autophagy factor FIP200 (Schlütermann et al. 2012 Sci Rep; Yamano et al. 2024, EMBO J). Indeed, also in our hands TBK1 was hyperactive in the FIP200 knockout line, as confirmed by the strong increase in p-TBK1 S172 (Revision Figure 2.3A). We then blotted for NAP1 and observed a shift for NAP1, which would be consistent with phosphorylation of NAP1 in the FIP200 knockout line. Moreover, upon induction of mitophagy by O/A treatment, this band accumulated strongly in the FIP200 knockout line (Revision Figure 2.3B), consistent with increased phosphorylation upon mitophagy stimulation.

Revision Figure 2.3

We then continued to verify whether this higher NAP1 band corresponds to the phosphorylated form of NAP1. To this end, we incubated the HAP1 protein lysates with recombinant lambda phosphatase. This showed that lambda phosphatase was able to dephosphorylate NAP1 (Revision Figure 2.4). The protein 4E-BP1 served as a positive control to confirm the activity of the lambda phosphatase.

Revision Figure 2.4

Next, we set out to identify the kinase responsible for this NAP1 phosphorylation. We speculated that TBK1 would be the responsible kinase but hypothesized that the autophagy-associated kinase ULK1 represents another plausible candidate. To this end, we incubated the FIP200 knockout cells with TBK1 or ULK1 inhibitors and tested whether we could see a shift on the gel from phosphorylated to non-phosphorylated NAP1. Note that the TBK1 inhibitor (GSK8612) is considered very specific but that the ULK1 inhibitor shows some weak cross reactivity towards TBK1 due to the structural similarities between both kinases. This revealed that with increasing concentrations of TBK1 inhibitor, we observed a complete disappearance of NAP1 phosphorylation (Revision Figure 2.5). In contrast, the ULK1/2 inhibitor did also reduce the smearing but only at very high concentrations, at which we suspect it would also start to cross-react and inhibit TBK1. However, it was not able to entirely erase the NAP1 phosphorylation. This led us conclude that TBK1 is the primary kinase responsible for phosphorylating NAP1 in the FIP200 knockout cells.

Revision Figure 2.5

As our data above could not rule out a contribution from ULK1 towards NAP1 phosphorylation, we tested this more formally in an *in vitro* kinase assay and incubated TBK1 and ULK1 side-by-side with NAP1. This revealed that TBK1, but not the ULK1 kinase complex, could potentially phosphorylate NAP1 as indicated by the smearing pattern on the SDS-PAGE gel (Revision Figure 2.6). Together, these *in vitro* and cellular data reveal that NAP1 can

be phosphorylated both *in vitro* and *in cellulo* and that TBK1 is the primary kinase responsible for this NAP1 phosphorylation.

Revision Figure 2.6

We then went on to investigate if the phosphorylated NAP1 residues in the FIP200 knockout cells were the same as the six residues we identified in the TBK1 *in vitro* kinase assay. To this end, we transfected the FIP200 knockout cells with pcDNA3.1-GFP-NAP1 and immunocaptured NAP1 using GFP-trap beads 48h after transfection. These beads were then submitted for mass spectrometry analysis to identify the phosphorylated NAP1 residues. Satisfyingly, this revealed that many of the sites we had identified in our *in vitro* kinase assay were also phosphorylated *in vivo* (Revision Figure 2.7). Moreover, the six residues which were shared between our *in vitro* and *in cellulo* samples were also highly conserved across species (Revision Figure 2.8), further indicating that these residues may fulfil an important role for NAP1's function.

In vitro		In cellulo		NAP1 6xAla
Trypsin	Chymotryp	Trypsin	Chymotryp	
S82	S27 S34	S82	S34	S34
S83	S83	S83	S82/83*	S82
S120	S120	S120	S83	S83
S206	S155			S120
S318	S318	S300 S318	S343	S318
	S343 S348			S343

Revision Figure 2.7

To determine the function of NAP1 phosphorylation by TBK1, we hypothesized that, as for other cargo receptors like OPTN and p62, this phosphorylation could boost the interaction of NAP1 with LC3/GABARAP proteins. To this end, we designed microscopy bead-assays in which we assessed the interaction between NAP1 or SINTBAD and LC3B, mixed with recombinant TBK1, and in presence or absence of ATP/MgCl₂. This revealed that upon phosphorylation by TBK1, both NAP1 and SINTBAD bound stronger to LC3B (Revision Figure 2.10) – consistent with how TBK1 activates cargo receptors.

Revision Figure 2.10

Taken together, we identify NAP1 and SINTBAD phosphorylation by TBK1 and propose that this could be the mechanism by which NAP1 and SINTBAD are converted from their inhibitory role into their stimulatory role upon recruitment to the mitochondrial surface by NDP52, which would bring NAP1 and SINTBAD in the vicinity of clustered and activated TBK1. We have added these data to Fig. 4 and Fig. S4.

3. In addition, structural based data might become explainable for the author's model (transition of NAP1/SINDBAD from OPTN to NDP52.).

We thank the reviewer for this comment. As mentioned above, we apologize in case this was insufficiently clear from our manuscript. NAP1 and SINTBAD do not bind OPTN directly. Instead, their negative regulation of mitophagy occurs indirectly by limiting the amount of TBK1 that OPTN can recruit and cluster on the mitochondrial surface, which is necessary for TBK1 to become activated and initiate a cascade of mitophagy-promoting events.

Other comments

1. In Figure 1C, the exposure of HeLa cells with O/A decreased the levels of NAP and SINDBAD, which were partially recovered by the treatment of BafA1. There is a possibility that the exposure of HeLa cells with O/A decreased gene expression of NAP and SINDBAD. Alternatively, some population of NAP and SINDBAD may be released from damaged mitochondria. The authors should investigate such possibilities.

We thank the reviewer for this comment and agree that the partial rescue upon BafA1 treatment is likely complemented by alternative pathways. To this end, we first asked if the proteasome would be one such complementing system. This revealed that in absence of mitochondrial damage (elicited by O/A treatment), the preferred route of degradation for both factors appears to be proteasomal. Upon mitochondrial damage, a fraction of NAP1/SINTBAD are degraded

by autophagy along with the damaged mitochondria, in addition to a significant fraction of the NAP1/SINTBAD pool that is degraded via proteasomes. This therefore indicates that the missing 50% (from Figure 1C) is not due to drastic reductions in gene expression but, instead, can be explained by proteasomal turnover. This is further supported by the accumulation of SINTBAD in the condition of OA/MG132 which exceeds basal expression levels, indicating the continued transcription and translation of NAP1 and SINTBAD during O/A treatment.

While the reviewer marks an important point about alternative degradation pathways and transcriptional responses of NAP1/SINTBAD upon mitophagy-induction with O/A treatment, it requires a long explanation for the observed results to be discussed in the manuscript. We felt this would deviate the reader's attention from the main message of the paper, which is the involvement of NAP1/SINTBAD in mitophagy. Hence, we have not added these revision experiments to the main manuscript.

2. In Figure 3B, difference between Penta KO (parental control, CTL) and NAP1/SINTBAD DKO/penta KO (clones 20 and 26) expressing BFP-Parkin and GFP-NDP52 on mitophagy activity was subtle, and this reviewer is not sure whether NAP1 and SINTBAD are involved in NDP52-mediated mitophagy.

We thank the reviewer for this comment and agree that the differences between the NAP1/SINTBAD double-knockout cells and the control cells are relatively modest. However, also mentioned in the manuscript (page 6 - lines 148-153), NDP52 can drive mitophagy through either ULK1/2 or TBK1. In other words, removal of NAP1/SINTBAD abrogates the recruitment and activation of TBK1 during NDP52-driven mitophagy but through the compensation by the ULK1/2 kinases, the overall reduction in mitophagy is relatively modest. To overcome this limitation, we assessed whether NAP1 and SINTBAD are truly involved and important for NDP52-mediated mitophagy by eliminating the (compensatory) contribution by ULK1/2. To this end, we generated ULK1/2 double knockouts in the pentaKO background, allowing us to assess NDP52-mediated mitophagy in cells that exclusively depend on TBK1. When we then deleted NAP1/SINTBAD on top, we observed a complete blockage of mitophagy (Figure 3E) and mitochondrial recruitment of WIPI2 (Figure 3F) because NAP1 and SINTBAD are essential for the recruitment of TBK1 during NDP52-mediated mitophagy.

Figure 3 of the manuscript.

3. In Figure 3D, the meaning of the difference in salt concentration is not explained.

We thank the reviewer for this question. To clarify, the salt concentration provides stringency to the biochemical protein-protein interactions, where increasing salt concentrations may break/reduce weaker protein-protein interactions while stronger protein-protein interactions are still retained. This therefore allowed us to determine relative binding strengths and visually confirm that the NDP52-FIP200 interaction was weaker in absence of SINTBAD. Addition of SINTBAD (and TBK1) substantially strengthened this interaction, as SINTBAD can bind to both NDP52 and FIP200, thereby stabilizing the tertiary complex. Following the reviewer's

suggestion to state this more clearly in the manuscript, we have now added following sentence to the manuscript:

Moreover, the deletion of NAP1/SINTBAD may weaken the NDP52-FIP200 interaction in cells as in vitro reconstitution revealed that FIP200 was more robustly retained on NDP52 coated beads in presence of SINTBAD. This was particularly evident when the protein-protein interactions were weakened with increased salt concentrations (Fig. 3D). In cells, the absence of NAP1/SINTBAD resulted in a reduction of ATG13 recruitment to mitochondria during NDP52-mediated mitophagy (Fig. S2B) underscoring the importance of NAP1/SINTBAD in this critical early step of mitophagy initiation.

4. In Figure 3F-G, the number of mitochondrial WIPI2 and ATG13 puncta per cell should be quantified.

We agree with the reviewer's helpful comment that the manuscript would benefit from a more quantitative assessment of these panels. We therefore quantified the number of ATG13 and WIPI2 foci colocalizing with the mitochondrial marker HSP60 and can now confirm that the recruitment of WIPI2 is drastically reduced in absence of NAP1/SINTBAD. For ATG13, we observed a weak reduction. We therefore amended the text in the following way and added the quantification to Figure 3F and supplementary Figure S2 (as shown below).

In the absence of ULK1/2, NAP1/SINTBAD emerged as crucial factors for NDP52-mediated mitophagy as evident from significantly reduced COXII turnover (Fig. 3E), and a reduction in WIPI2 recruitment to mitochondria upon O/A treatment (Fig. 3F). The impact on WIPI2 recruitment was comparable to the inhibition of TBK1 using the small molecule BX795.

Cropped from Figure 3 and S2 of the manuscript.

5. Figure 4C, it is unclear which cells (wild-type (WT) or penta KO (5KO) HeLa cells expressing BFP-Parkin and mt-mKeima) are used.

We thank the reviewer for pointing this out and agree that the figure legend was insufficiently clear. We have therefore added that all these experiments were performed in pentaKO (5KO) cells to rule out any contribution from indirect NDP52-recruitment. Moreover, the experiments with NAP1 mutants in Figure 4D were performed in 5KO + NAP1/SINTBAD DKO cells. The latter was important to rule out any contribution from endogenous wild-type NAP1. Since NAP1 can dimerize through its coiled coil domains, we wanted to make sure that our mutant NAP1 does not dimerize with endogenous wild type NAP1. We have now also specified this more carefully in the figure legend.

6. Figure 4D, the domain structures of NAP1 and which region is the NDP52, FIP200, TBK1 binding region should be indicated. Also, binding defect of each deletion mutant should be shown in experiments.

We appreciate the reviewer's valuable comment and have performed the requested control experiments to confirm the loss-of-binding for each of the respective mutants. This confirmed that each of the indicated mutants were deficient in binding NDP52, FIP200, or TBK1, respectively. Note that NDP52 and FIP200 mutants had been published previously by the Randow lab and our data confirmed their earlier findings. The TBK1 mutant was designed by us in this study and its validation was already incorporated in Figure S5. However, we agree

with the reviewer that our manuscript benefits from a comprehensive validation figure regarding the different mutants we have therefore incorporated these data as Figure S3. As suggested by the reviewer, we also added a schematic about the domain structure of NAP1 and the location of the respective binding sites for each partner.

7. In Figure 5, ITC analysis is available for determination of each Kd value.

We thank the reviewer for this comment and agree that information on the binding strength of the OPTN-TBK1 versus NAP1-TBK1 complexes is important to be taken in consideration. However, this information is already available as the Lifeng Pan lab previously showed that NAP1 binds about four times stronger to TBK1 than OPTN (Li et al. 2016 Nat Comms). Their binding strengths were obtained by ITC measurements of the minimal binding regions of each factor. As our study was conducted with full-length proteins (instead of the minimal binding fragments), we attempted to confirm that these previously published binding strengths would also hold true for the full-length proteins. However, our full-length NAP1 protein was not stable enough for ITC analysis and aggregated along the course of the assay. We were therefore not able to extend our analysis on binding strengths to the full-length proteins.

8. Statistic analyses are required for Figure S3, S4 and S6.

We have now quantified Figure S4 (now S6) and S6 (now S8) and added to the manuscript. Additionally, we have also quantified Figure 6A as we felt this would further strengthen the manuscript.

Regarding Figure S3 (now S5), we have quantified all three replicates (see Revision Figure 8.1) but have not performed statistical tests on this plot as it would give false impressions. The plot represents the amount of OPTN/NAP1 that is bound to TBK1 relative to their maximum amount. As such, the relative amounts would false-positively evoke high significance scores. Hence, we have refrained from adding this statistical analysis to this plot. Moreover, we have opted for displaying one of three replicates, representative for the others as can be seen below.

However, we feel that plotting a single replicate is clearer and makes the interpretation easier than plotting all three datapoints.

Revision Figure 8.1

9. In addition to immunoblot analysis shown in Figure 5F, the authors should perform mtKeima assay.

We have now performed the suggested experiment. This confirmed the results we obtained earlier by western blotting, revealing that overexpression of wild type NAP1, but not the TBK1-binding mutant, results in slower mitophagy. We have added the figure to the manuscript as Figure S7.

10. Figure 6D, quantification is required.

We thank the reviewer for pointing this out and agree that the manuscript would benefit from a more quantitative assessment. We therefore quantified the number of TBK1 foci colocalizing with the mitochondrial marker HSP60 and can now confirm that OPTN recruits more TBK1 to the mitochondrial surface compared to NDP52. This quantification has been included in Figure 6D (as shown below).

Cropped from Figure 6

Reviewer #2:

Remarks to the Author:

In this manuscript, the authors studied the role of the TBK1 adaptors, NAP1 and SINTBAD, in mitophagy. They first found that NAP1/SINTBAD has an inhibitory role in PINK1/Parkin-dependent mitophagy. Interestingly, in NDP52-mediated mitophagy, NAP1/SINTBAD acts as a cargo co-receptor by recruiting TBK1 to NDP52, stabilizing its interaction with FIP200. Conversely, NAP1/SINTBAD inhibits OPTN-mediated mitophagy by competing with OPTN for TBK1 binding. Finally, the authors analyzed the crosstalk of OPTN-mediated and NDP52-mediated mitophagy regulation by NAP1/SINTBAD and found that NAP1/SINTBAD acts as cargo receptor rheostats and elevates the threshold for mitophagy initiation by OPTN while promoting the progression of the pathway once set in motion by supporting NDP52.

Overall, the experiments are well designed and the data support most of the conclusions and are potentially important. However, there are still several major concerns that need to be addressed by the authors.

We would like to thank the reviewer for taking the time to carefully read our manuscript and we appreciate the kind words about our work. As outlined below, we have addressed the points raised by the reviewer with additional experiments.

1. In many experiments other than in vitro assays, the expression levels of NDP52 and OPTN have not been examined. Their expression levels may affect TBK1 activity and mitophagy. It is particularly important to show NDP52 and OPTN expressed against PentaKO cells in comparison to endogenous expression levels in control HeLa cells.

This is a valuable comment, and we agree that this is an important point in the interpretation of our experiments. To this end, we undertook the following steps to ensure as low as possible overexpression levels:

- (i) OPTN and NDP52 were cloned in retroviral vectors, which do not contain CMV promoters to avoid too high overexpression levels upon viral integration into the host genome.
- (ii) Cells were FACS sorted for equal expression levels of OPTN and NDP52 within experiments but also across experiments, to ensure continuity across the different cell lines.
- (iii) We gated for cells in the lower half of GFP-positive quadrant, to avoid enriching for highly overexpressing clones. Our cells were thus selected for as lowly overexpressing as possible.
- (iv) We performed western blot experiments to confirm equal expressions between OPTN and NDP52. This confirmed that we successfully sorted for equal expression levels of OPTN versus NDP52 and that our subsequent experiments allow the direct comparison of OPTN- versus NDP52-driven mitophagy.

- (v) We then also compared endogenous versus exogenous expression levels. The result of which can be seen below:

Note that, while these blots show an overexpression of OPTN and NDP52 relative to their endogenous expression levels, which could affect overall mitophagy rates and TBK1 activation levels as the reviewer correctly states, we confirmed several of our key findings in single or double knockout cell lines (for example the single OPTN or NDP52 knockout lines in Fig. 6C). Since those results were consistent with our observations in overexpressing cell lines, we feel confident that our findings with rescue (overexpressing) lines are biologically relevant and are unlikely to result from systematic artefacts created from the overexpression system.

We supported with three additional experiments. First, we evaluated the overexpression of GFP-NDP52 in OPTN-KO cells (to exclude the contribution of OPTN to TBK1 activation) and assessed TBK1 activation (p-TBK1). This showed that increased expression levels of NDP52 did not change the outcome of the experiment (Revision Figure 1.1).

Revision Figure 1.1

Secondly, we compared p-TBK1 levels in WT cells (with endogenous expression levels of OPTN and NDP52) versus pentaKO cells rescued with either GFP-OPTN or GFP-NDP52. This showed that pentaKO cells overexpressing GFP-OPTN had roughly the same level of TBK1 activation as read out by p-TBK1 levels (Revision Figure 1.2).

Revision Figure 1.2

Thirdly, we measured mitophagy flux in wild-type HeLa cells, using cleavage of the pSu9-GFP-HaLo reporter as a read out, and compared the rate of mitophagy flux with those of the pentaKO (5KO) + GFP-OPTN or GFP-NDP52 cells. This further confirmed that our overexpression levels do not cause an excessive deviation from the endogenous mitophagy flux (Revision Figure 1.3).

Revision Figure 1.3

Together, and in addition to the complementary knockout experiments we did in the manuscript (such as Fig. 6C), we are confident that our observations are biologically relevant.

2. In this manuscript, NAP1 and SINTBAD are analyzed as having the same function because of their structural similarity, and most experiments use only NAP1. However, it is possible that NAP1 and SINTBAD have different functions, but this is rarely mentioned; SINTBAD should be used in at least some key experiments to see if SINTBAD yields the same results as NAP1.

We thank the reviewer for this comment. The reviewer correctly states that NAP1 and SINTBAD share structural similarity, with the exception that SINTBAD contains an additional C-terminal domain that is not present in NAP1. This led us to believe that working with NAP1 would yield insights that would be likely also true for SINTBAD as the protein domains of NAP1 are also present in SINTBAD (note: the opposite is not true as SINTBAD contains a C-terminal region that is missing in NAP1). In addition, we observed that both NAP1 and SINTBAD were recruited to the mitochondrial surface upon mitophagy induction, and their combined loss led to statistically significant acceleration of mitophagy (whereas the individual knockouts only provided non-significant trends toward acceleration), which would be consistent with a role that is shared between NAP1 and SINTBAD. Also, the DNA sequence of SINTBAD contains a very difficult GC-rich region which makes it very hard to work with and we frequently observed loss of fragments upon cloning or amplification in bacterial cells. To avoid the unintended introduction of SINTBAD deletion variants into our experiments, we felt NAP1 would be the preferred protein to work with in our experiments for all three points discussed above.

However, the reviewer is correct in stating that the manuscript would benefit from verifying SINTBAD's response in some key experiments to see if SINTBAD yields the same results as NAP1. To this end, we repeated several key experiments with SINTBAD to confirm that the role of NAP1 and SINTBAD during mitophagy, as discussed in this manuscript, can be considered the same. In detail:

- (i) Figure 1: These experiments were performed with both NAP1 and SINTBAD, confirming that both proteins are recruited to the mitochondrial surface upon mitophagy induction.
- (ii) Figure 2: Here we show that it requires the combined deletion of NAP1 and SINTBAD to see statistically significant acceleration of mitophagy, as the expression of either protein alone is sufficient to reduce mitophagy flux.

- (iii) Figure 3: Here we show that NAP1/SINTBAD support NDP52-driven mitophagy and the key experiments were either performed in cells expressing or double-knockout for both NAP1 and SINTBAD (e.g. panel A-D). The biochemical reconstitution experiment in panel D was also performed with recombinant SINTBAD, consistent with our findings from cells.
- (iv) Figure 4: The original figure showed an important role for NAP1 in supporting mitophagy as a cargo co-receptor. As we had not performed these experiments with SINTBAD, we now added these as a new supplementary figure to the manuscript. This confirmed that SINTBAD can also potently induce mitophagy when artificially tethered to the mitochondrial surface. As we showed for NAP1, also SINTBAD can induce mitophagy in this assay in the pentaKO background, confirming this mitophagy induction is mediated by SINTBAD itself and not through indirect recruitment of NDP52. Moreover, inhibition of TBK1's kinase activity with a small molecule impairs SINTBAD's capacity to induce mitophagy in this tethering assay, consistent with what we found for NAP1. Together, this shows that we can reproduce our NAP1 findings with SINTBAD, further suggesting that both proteins have the same function in mitophagy. The data have been added as a new supplementary Figure S3.

C

D

- (v) Figure 5: This figure shows that NAP1 and SINTBAD compete with OPTN for TBK1-binding. As one of the key experiments (panel A-B) was only performed with NAP1, we repeated this experiment with SINTBAD and observed that SINTBAD also competed with OPTN for TBK1 recruitment to the GST-4xUb coated beads. Note that we were unable to purify sufficiently high amounts of unlabeled SINTBAD, required to reach the higher concentrations in this experimental set-up, and we therefore rearranged the tags so that OPTN became unlabeled and SINTBAD became mCherry-tagged, as it aided its purification.

(vi) Figure 7: This figure describes the crosstalk between OPTN and NDP52. The figure contains only two panels with NAP1 and we have repeated this experiment with SINTBAD-mCherry (for the same reasons as mentioned above). This once more confirmed that also SINTBAD competes with OPTN for TBK1 binding and shows further that addition of NDP52 restores TBK1 signal on the GST-4xUb coated beads. We have added this figure as a supplementary Figure S9B.

We are therefore grateful for the excellent suggestion from the reviewer to provide additional data on SINTBAD and this has strengthened our conclusions that NAP1 and SINTBAD play a similar role during mitophagy. We therefore feel these additional data allow us to be more confident about a redundant role of NAP1 and SINTBAD during mitophagy.

3. In Figure 6D, the authors conclude that OPTN is able to accumulate more TBK1 on the mitochondrial surface than NDP52. However, it seems possible that NDP52 is less accumulated on mitochondria or less expressed than OPTN and therefore can bind less TBK1. A more quantitative interpretation of Figure 6D is needed to show whether OPTN or NDP52 accumulates more TBK1.

We thank the reviewer for this comment and agree that the manuscript would benefit from a more quantitative assessment. We would therefore like to refer to our expression blots shown in response to point 1 (point iv), which show that OPTN and NDP52 are expressed at similar levels. Moreover, the imaging settings were kept identical for both cell lines, with equal laser

power and gain intensity, so that their signal intensities can be directly compared. The discrepancy that may have been observed in the images submitted in the initial manuscript may have therefore stemmed from the stochastic variation between cells. We therefore replaced images with cells that share more similar OPTN and NDP52 expression levels.

In addition, we also quantified the number of TBK1 foci colocalizing with the mitochondrial marker HSP60 and can now confirm that OPTN recruits more TBK1 to the mitochondrial surface compared to NDP52. This quantification has been included in Figure 6D (as shown below).

4. The crosstalk between OPTN-axis and NDP52-axis is not fully elucidated.

4.1. In Figure 7C, OA+Rapalog induces mitophagy slightly more strongly than OA alone at 2 hours mitophagy induction, but no difference is observed at other induction times. This result alone is not sufficient to conclude that the recruitment of TBK1 by OPTN enhances mitophagy via NDP52.

4.2. The recovery of TBK1 on the beads by adding NDP52 in Figure 7D may simply be a result of NDP52 on the beads accumulating TBK1 via NAP1. TBK1 accumulation on beads with and without OPTN is needed to be compared to confirm that NDP52 supports OPTN-driven mitophagy.

4.3. If the recruitment of TBK1 by NDP52 supports OPTN-driven mitophagy, the binding between TBK1 and OPTN should be stronger in the experiment in lane 10 of Figure 7E with NDP52 than without NDP52, so this need to be confirmed.

The reviewer is correct in stating that only at 2 hours of mitophagy induction there was a significant difference between the different cell lines in Figure 7C. However, while our results showed a clear trend, we suspected that our statistical power suffered from the experimental variation for the limited number of independent experiments (N=3). We therefore tested whether increasing the number of replicates would increase the statistical power. We are happy to present that, after repeating the experiment a couple more times, we can now share an updated graph which confirms that recruitment of TBK1 boosts NDP52-driven mitophagy, not only at 2 h of O/A but also the other early time points prior to reaching the maximum plateau from 4 h onwards. We have now replaced the graph in Figure 7C with this updated version below (Revision Figure 4.1).

Revision Figure 4.1

We also agree with the reviewer's statement that if there would be crosstalk between OPTN and NDP52, one expects an enhanced effect in some of our assays. However, the assay in Figure 7E would not allow such comparison because there we captured GFP-TBK1 onto GFP-trap beads and evaluated how much OPTN versus NDP52/NAP1 is bound to TBK1. This complements the microscopy-based bead assay from Figure 7D, which tests how much TBK1 is recruited to the ubiquitin-coated beads, without discriminating which cargo receptor TBK1 would be bound to. This GFP-TBK1 pull-down experiment fills this gap and gives insight into which cargo receptor TBK1 is bound to under the different stoichiometric ratios. However, due to the competition between OPTN and NAP1, TBK1 would be bound to either OPTN or NDP52/NAP1 and would thus not allow to visualize a cumulative or enhanced effect from NDP52 to OPTN.

However, the reviewer raises an important point with this question, and we therefore designed additional experiments to answer it in detail. We started from an interesting observation that we had made earlier: when we performed microscopy-based bead assays we reproducibly noticed that the amount of TBK1 bound to OPTN-GST coated beads decreased after addition of ATP/MgCl₂ (Revision Figure 4.2). We confirmed that this decrease was dependent on the kinase activity (and not simply the addition of ATP/MgCl₂) by using the kinase-dead variant of TBK1.

Revision Figure 4.2

This effect enabled us to assess the contribution of having NDP52 in addition to OPTN in our microscopy-based bead assay. Indeed, we observed that TBK1 is released from OPTN bound to the GST-4xUb coated beads in the presence of ATP/MgCl₂ (Revision Figure 4.3). However, upon addition of NDP52 and NAP1 (but not NDP52 alone as it cannot interact with TBK1 directly) was able to retain the released TBK1. In other words, NDP52/NAP1 might be required to cooperate with OPTN to allow OPTN to recruit, cluster, and activate TBK1, after which TBK1 is partially released. The NDP52/NAP1-axis can then bind activated TBK1 and retain it on the mitochondrial surface, allowing TBK1 to phosphorylate other substrates (such as newly incoming OPTN and NDP52 molecules whose ubiquitin-binding affinities are increased by TBK1 phosphorylation, which then in return can recruit more TBK1— creating an effective feedforward loop). As such, NDP52 could contribute to OPTN-mediated mitophagy, and this could explain the overall boost we observe when both cargo receptors are present. The new data have been added as Fig. 7F.

F

Revision Figure 4.3

We have also carefully revised our phrasing in this section of the manuscript as our data suggest that OPTN boosts NDP52-mediated mitophagy (Fig. 7C). However, as the reviewer correctly

states that we do not have direct evidence for the opposite direction (in which we would have tethered FKBP-SKICH domain of NDP52 during OPTN-driven mitophagy), we have referred to this situation as ‘cooperation’ as NDP52 aids by sequestering NAP1 away and by capturing the activated TBK1 that is released from OPTN. We hope this helps the readers in understanding the difference between both receptors, based on the evidence we have been able to collect.

5. Figure 5A, 6E, and 7D show the diagram of the experiment, but Figure 3D does not, making it difficult to understand.

We thank the reviewer for this comment and are grateful to hear that these schematics are helpful to our readers. We do, however, apologize for having it missed in Figure 3D and are therefore thankful that the reviewer pointed this mistake out. We have now added the missing diagram to Figure 3D.

6. Figure 8, panel 4: Is the pale orange circle with LC3 bound a phagophore or an autophagosome? If it is a phagophore, it should be flat or cup-shaped, not circular, and should be a single membrane; if it is an autophagosome, it should be described as enclosing the mitochondria.

We thank the reviewer for this question and are happy to clarify that this orange circle represents an ATG9-vesicle, which will form the seed for the expanding phagophore. For this reason, we displayed the vesicle as a circle and not yet flat or cup-shaped phagophore. It would need the recruitment of ATG2 to the vesicle and potentially other membrane precursors for the influx of lipids from the ER to occur in order to expand and to form the typical cup-shape phagophore. To remove this confusion from our manuscript, we have now added this information to the figure legend and thank the reviewer for pointing out this was insufficiently clear in the submitted version.

Decision Letter, first revision:

Message: 21st Feb 2024

Dear Professor Martens,

Thank you again for submitting your manuscript "Control of mitophagy initiation and progression by the TBK1 adaptors NAP1 and SINTBAD". We now have comments (below) from the 2 reviewers who evaluated your paper. In light of these reports, we remain very interested in your study and would like to see your response to the last comments of the referees, in the form of a revised manuscript.

You will see that though reviewer #2 is convinced that the manuscript is now ready for publication, reviewer #2 opines that assessing the cellular phenotypes and significance of NAP1 phosphorylation in O/A-dependent PARKIN-mediated mitophagy by using the phosphorylation mutants is necessary for the work to be accepted. We kindly ask you to experimentally address this concern and to please respond to all concerns of the referees in full in a point-by-point response and highlight all changes in the revised manuscript text file. If you have comments that are intended for editors only, please include those in a separate cover letter.

We expect to see your revised manuscript within 6 weeks. If you cannot send it within this time, please contact us to discuss an extension; we would still consider your revision, provided that no similar work has been accepted for publication at NSMB or published elsewhere.

Reporting Summary:

When submitting the revised version of your manuscript, please pay close attention to our [href="https://www.nature.com/nature-portfolio/editorial-policies/image-integrity">Digital Image Integrity Guidelines](https://www.nature.com/nature-portfolio/editorial-policies/image-integrity). and to the following points below:

-- that unprocessed scans are clearly labelled and match the gels and western blots

presented in figures.

-- that control panels for gels and western blots are appropriately described as loading on sample processing controls

-- all images in the paper are checked for duplication of panels and for splicing of gel lanes.

Data availability: this journal strongly supports public availability of data. All data used in accepted papers should be available via a public data repository, or alternatively, as Supplementary Information. If data can only be shared on request, please explain why in your Data Availability Statement, and also in the correspondence with your editor. Please note that for some data types, deposition in a public repository is mandatory - more information on our data deposition policies and available repositories can be found below: <https://www.nature.com/nature-research/editorial-policies/reporting-standards#availability-of-data>

Nature Structural & Molecular Biology is committed to improving transparency in authorship. As part of our efforts in this direction, we are now requesting that all authors identified as 'corresponding author' on published papers create and link their Open Researcher and Contributor Identifier (ORCID) with their account on the Manuscript Tracking System (MTS), prior to acceptance. This applies to primary research papers only. ORCID helps the scientific community achieve unambiguous attribution of all scholarly contributions. You can create and link your ORCID from the home page of the MTS by

clicking on 'Modify my Springer Nature account'. For more information please visit please visit www.springernature.com/orcid.

[Redacted]

Sincerely,

Dimitris Typas
Associate Editor
Nature Structural & Molecular Biology
ORCID: 0000-0002-8737-1319

Reviewers' Comments:

Reviewer #1:

Remarks to the Author:

In the revision process, the authors performed a large number of experiments, provided a novelty (i.e., phosphorylation of NAP1 by TBK1 and its enhanced binding to LC3) and also did statistical analysis to ensure the quality of the data. This reviewer thinks that the presented data are sufficient to address the concerns raised in the first round. Critically, however, the significance of NAP1 phosphorylation in O/A-dependent PARKIN-mediated mitophagy should be analyzed using the phosphorylation mimics and defective mutants. Clarification of this point would make this paper worthy of acceptance.

Reviewer #2:

Remarks to the Author:

The authors have addressed all comments made by this reviewer. I believe that the current manuscript becomes suitable for publication.

Author Rebuttal, first revision:

Reviewers' Comments:

Reviewer #1:

Remarks to the Author:

In the revision process, the authors performed a large number of experiments, provided a novelty (i.e., phosphorylation of NAP1 by TBK1 and its enhanced binding to LC3) and also did statistical analysis to ensure the quality of the data. This reviewer thinks that the presented data are sufficient to address the concerns raised in the first round. Critically, however, the significance of NAP1 phosphorylation in O/A-dependent PARKIN-mediated mitophagy should be analyzed using the phosphorylation mimics and defective mutants. Clarification of this point would make this paper worthy of acceptance.

We thank the reviewer for taking the time to carefully read our revised manuscript and are grateful to hear the reviewer believes that the presented data are sufficient to address the concerns raised in the first round.

The reviewer, however, also raises an additional point regarding the significance of NAP1 phosphorylation in O/A-dependent PARKIN-mediated mitophagy. We have therefore tested this experimentally by mutating all six serine residues in NAP1, identified as the TBK1 phosphorylation residues by mass spectrometry, and assessed the impact on PINK1/Parkin mitophagy.

First, we assessed whether phospho-deficient NAP1 (6xAla) would still be able to induce mitophagy when artificially tethered to the mitochondrial surface. Our previous experiments with wild-type NAP1 had shown that inhibition of TBK1 blocked NAP1-induced mitophagy. We therefore hypothesized that direct NAP1 phosphorylation by TBK1 could be the cause of this. To test this directly, we tethered wild-type and 6xAla-mutant NAP1 and measured mitophagy flux. This revealed that mitophagy flux was not affected. This suggests that NAP1 phosphorylation is not critical under these conditions, suggesting that the critical activity of the TBK1 kinase is directed at another autophagy factor and not NAP1.

Revision Figure 1.1

Next, we tested the role of NAP1 phosphorylation during NDP52-mediated mitophagy. To this end, we used the 5KO HeLa cells in which also ULK1/2 and NAP1/SINTBAD were knocked out, and rescued with GFP-NDP52. Our earlier results (Fig 3E) had shown that under these conditions, NAP1 and SINTBAD are critical as NDP52-driven mitophagy uses TBK1 in the absence of ULK1/2. To utilise TBK1, NDP52 needs NAP1/SINTBAD to recruit TBK1 to sites of autophagosome biogenesis. To evaluate the impact of NAP1 phosphorylation under these conditions of NDP52-NAP1/SINTBAD-TBK1 dependent mitophagy, we rescued the cells with wild-type NAP1 or 6xAla-mutant NAP1. This showed that the 6xAla-mutant could also rescue the mitophagy defect, suggesting that NAP1 phosphorylation is not critical but, as we showed *in vitro*, may also further boost mitophagy by facilitating the interaction with LC3s/GBRPs (Fig 4I). This mimicks the role of TBK1 phosphorylation on cargo receptors (like OPTN or NDP52), where TBK1 phosphorylation further activates these receptors and broadens their affinity towards different ubiquitin-species, but is not critical on its own (Richter et al (2016) PNAS). Consistently, NAP1 binds directly to both the FIP200-ULK1 complex and to TBK1 (this work, but also; Nguyen et al (2023) Mol Cell; Ravenhill et al (2019) Mol Cell; Fu et al (2021) Sci Adv), either of which are sufficient to initiate autophagosome biogenesis (FIP200-ULK1 > ATG9 + PI3KC3-CI; TBK1 > PI3KC3-CI + FIP200-ULK1). This web of interactions therefore provides resilience and redundancy to the pathway, and prevents the possibility that failure of a single interaction would stall the pathway completely. Along those lines, NAP1 phosphorylation may form another layer of redundancy in promoting mitophagy flux without being critical on its own. However, to fully disentangle this possibility would require inhibiting all functionally redundant interactions, which is a very complex and challenging task to achieve. For example, it took our team over 5 years to disentangle the redundant and overlapping activities of ULK1/2 and TBK1 during PINK1/Parkin mitophagy (e.g. Nguyen et al (2023) Mol Cell).

Revision Figure 1.2

Reviewer #2:

Remarks to the Author:

The authors have addressed all comments made by this reviewer. I believe that the current manuscript becomes suitable for publication.

We thank the reviewer for taking the time to carefully read our revised manuscript and are grateful to hear the reviewer judges the manuscript suitable for publication.

Decision Letter, second revision:

Message: Our ref: NSMB-A48279B

8th Apr 2024

Dear Professor Martens,

Thank you for submitting your revised manuscript "Control of mitophagy initiation and progression by the TBK1 adaptors NAP1 and SINTBAD" (NSMB-A48279B). Like Reviewer #2 in the previous round, Reviewer #1 now also signs off, finding that the paper has improved in revision. Therefore we are happy to accept it in principle in Nature Structural & Molecular Biology, pending minor revisions to satisfy the referees' final requests (such as including the results of these final experiments in the manuscript) and to comply with our editorial and formatting guidelines.

We are now performing detailed checks on your paper and will send you a checklist detailing our editorial and formatting requirements in about two weeks. Please do not upload the final materials and make any revisions until you receive this additional information from us.

To facilitate our work at this stage, it is important that we have a copy of the main text as a word file. If you could please send along a word version of this file as soon as possible, we would greatly appreciate it; please make sure to copy the NSMB account (cc'ed above).

Sincerely,

Dimitris Typas
Associate Editor
Nature Structural & Molecular Biology
ORCID: 0000-0002-8737-1319

Reviewer #1 (Remarks to the Author):

Although the significance of phosphorylation of NAP1 by TBK1 has not been clearly demonstrated, careful experiments were performed to address the concerns raised in the second review round, and the reasons are explained based on the results in the rebuttal. This reviewer is concerned that this explanation is rarely provided in the revised paper, but I believe the revised manuscript is worthy of acceptance.

Author Rebuttal, second revision:

Reviewers' Comments:

Reviewer #1:

Remarks to the Author:

Although the significance of phosphorylation of NAP1 by TBK1 has not been clearly demonstrated, careful experiments were performed to address the concerns raised in the second review round, and the reasons are explained based on the results in the rebuttal. This reviewer is concerned that this explanation is rarely provided in the revised paper, but I believe the revised manuscript is worthy of acceptance.

We thank the reviewer and have included the data in the manuscript (Figure S4F-G), including a statement in the discussion which highlights that this warrants further investigation.

Final Decision Letter:**Message:** 22nd May 2024

Dear Professor Martens,

We are now happy to accept your revised paper "Control of mitophagy initiation and progression by the TBK1 adaptors NAP1 and SINTBAD" for publication as an Article in Nature Structural & Molecular Biology.

Your paper will be published online soon after we receive proof corrections and will appear in print in the next available issue. You can find out your date of online publication by

contacting the production team shortly after sending your proof corrections.

Please note that *Nature Structural & Molecular Biology* is a Transformative Journal (TJ). Authors may publish their research with us through the traditional subscription access route or make their paper immediately open access through payment of an article-processing charge (APC). Authors will not be required to make a final decision about access to their article until it has been accepted. Find out more about Transformative Journals

Sincerely,

Dimitris Typas
Senior Editor
Nature Structural & Molecular Biology
ORCID: 0000-0002-8737-1319